# ANALYSIS OF APPROXIMATE LINEAR PROGRAMMING SOLUTION TO MARKOV DECISION PROBLEM WITH LOG BARRIER FUNCTION

**Donghwan Lee, Hyukjun Yang, & Bumgeun Park**
Department of Electrical Engineering
Korea Advanced Institute of Science and Technology
Daejeon, 34141, South Korea
{donghwan,jundol32,j4t123}@kaist.ac.kr

## ABSTRACT

There are two primary approaches to solving Markov decision problems (MDPs): dynamic programming based on the Bellman equation and linear programming (LP). Dynamic programming methods are the most widely used and form the foundation of both classical and modern reinforcement learning (RL). By contrast, LP-based methods have been less commonly employed, although they have recently gained attention in contexts such as offline RL. The relative underuse of the LP-based methods stems from the fact that it leads to an inequality-constrained optimization problem, which is generally more challenging to solve effectively compared with Bellman-equation-based methods. The purpose of this paper is to establish a theoretical foundation for solving LP-based MDPs in a more effective and practical manner. Our key idea is to leverage the log-barrier function, widely used in inequality-constrained optimization, to transform the LP formulation of the MDP into an unconstrained optimization problem. This reformulation enables approximate solutions to be obtained easily via gradient descent. While the method may appear simple, to the best of our knowledge, a thorough theoretical interpretation of this approach has not yet been developed. This paper aims to bridge this gap.

## 1 INTRODUCTION

There are two primary approaches to solving Markov decision problems (MDPs): dynamic programming methods (Bertsekas and Tsitsiklis, 1996; Puterman, 2014) based on the Bellman equation and linear programming (LP) methods (Puterman, 2014; De Farias and Van Roy, 2003; Ghate and Smith, 2013; Ying and Zhu, 2020). Dynamic programming is by far the most widely used approach and constitutes the foundation of both classical and modern reinforcement learning (RL) (Sutton and Barto, 1998). In contrast, LP-based methods have been employed less frequently (Wang and Chen, 2016; Chen and Wang, 2016; Lee and He, 2019; 2018; Chen et al., 2018; Serrano and Neu, 2020; Neu and Okolo, 2023; Nachum and Dai, 2020; Bas-Serrano et al., 2021; Lu et al., 2021; 2022) in RL, though they have recently gained traction in contexts such as offline reinforcement learning (Ozdaglar et al., 2023; Zhan et al., 2022; Gabbianelli et al., 2024; Kamoutsi et al., 2021; Sikchi et al., 2024). The relative underuse of LP formulations can be attributed to the fact that they result in inequality-constrained optimization problems, which are generally more difficult to solve effectively compared to Bellman-equation-based methods.

Recently, the LP formulation of MDPs has recently received renewed interest, especially in the offline-RL literature (Ozdaglar et al., 2023; Zhan et al., 2022; Gabbianelli et al., 2024; Kamoutsi et al., 2021; Sikchi et al., 2024), because it offers several advantages over Bellman-equation–based approaches. Unfortunately, LP-based RL methods typically rely on primal-dual schemes (Wang and Chen, 2016; Chen and Wang, 2016; Lee and He, 2019; 2018; Chen et al., 2018; Serrano and Neu, 2020; Neu and Okolo, 2023; Nachum and Dai, 2020; Bas-Serrano et al., 2021; Lu et al., 2021; 2022), which are known to often exhibit relatively slow convergence and higher computational costs

in practice compared to standard RL approaches. For these reasons, we believe there is a clear need to develop effective alternatives for solving the LP form.

The purpose of this paper is to establish a theoretical foundation for addressing LP-based MDPs in a more effective and practical manner. The key idea is to employ the log-barrier function (Boyd and Vandenberghe, 2004), widely used in inequality-constrained optimization, to reformulate the LP representation of the MDP into an unconstrained optimization problem. This reformulation allows approximate solutions to be obtained efficiently using gradient descent (Nesterov, 2018; Bertsekas, 1999). While this approach may appear simple, to the best of our knowledge, a comprehensive theoretical interpretation has not yet been developed. This paper aims to bridge this gap.

More specifically, we investigate the single-objective function $f_\eta$ induced by the log-barrier formulation with the barrier parameter $\eta > 0$, whose minimizer yields an approximate solution, $\tilde{Q}_\eta$, to the original MDP. This approximate solution, $\tilde{Q}_\eta$, corresponds to an approximate optimal Q-function. We first conduct an error analysis, deriving not only an upper bound but also a lower bound on the error norm, $\|\tilde{Q}_\eta - Q^*\|_\infty$, between the approximate solution, $\tilde{Q}_\eta$, and the true optimal Q-function $Q^*$. These bounds depend linearly on the log-barrier parameter $\eta$, which implies that both the upper and lower bounds decrease linearly as $\eta$ becomes smaller. Beyond the error norm, we also establish error bounds for the MDP objective function $J^\pi$ itself. In particular, our framework yields both a primal approximate solution $\tilde{Q}_\eta$ and a dual approximate solution $\tilde{\lambda}_\eta$, from which the corresponding primal policy and dual policy are derived. For each case, we derive bounds on the deviation of their objective values from the optimal objective value, and these bounds likewise diminish linearly with $\eta$.

In addition, we establish several properties of the objective function $f_\eta$, including its convexity, properties of its domain, and properties of its sublevel set. As mentioned earlier, the approximate LP solution can be obtained via gradient descent, and we provide an analysis of the convergence behavior of this gradient descent method. Lastly, we explore the applicability and extension of the proposed theoretical foundation to deep RL. Specifically, we introduce a novel loss function, derived from the log-barrier formulation, which serves as an alternative to the conventional mean-squared-error Bellman loss in deep Q-network (DQN) (Mnih et al., 2015) and deep deterministic policy gradient (DDPG) (Lillicrap et al., 2015). This yields a new deep RL algorithm within the DQN and DDPG frameworks. The effectiveness of the proposed method is demonstrated through comparative evaluations with standard DQN and DDPG across multiple OpenAI Gym benchmark tasks. The experimental results demonstrate that the proposed method performs on par with conventional DQN across the evaluated environments, and achieves markedly superior performance in specific tasks. In addition, experimental results show that incorporating the proposed method into DDPG yields markedly improved learning performance compared to the conventional DDPG algorithm in a wide range of tasks. Finally, the main contributions are briefly summarized as follows: (1) Log-barrier LP for MDPs & error bounds: we introduce a novel log-barrier formulation of the MDP LP and derive rigorous error bounds for the approximate solution, explicitly quantifying how the approximation error scales with the barrier weight $\eta$. (2) Analytic properties & convergence: We prove structural properties of the objective (convexity, local strong convexity, local Lipschitzness, convex feasible domain) and show exponential convergence of deterministic gradient descent in the tabular setting. (3) Preliminary deep-RL evaluation: we propose a deep-RL variant (log-barrier loss) and provide empirical results showing stable training and, in several environments, superior performance to standard deep-RL baselines.

## 2  RELATED WORKS

Research on MDPs has traditionally been dominated by dynamic programming (DP) methods based on the Bellman equation (Bertsekas and Tsitsiklis, 1996; Puterman, 2014; Sutton and Barto, 1998), which underpin classical RL algorithms and modern deep RL methods such as DQN (Mnih et al., 2015), but these approaches can become less flexible in large-scale or constrained settings. As an alternative, linear programming (LP) formulations of MDPs (Puterman, 2014) have been studied extensively: De Farias and Van Roy (2003) introduced approximate linear programming (ALP), which reduces the number of decision variables by using linear function approximations; Malek et al. (2014) further exploited the dual LP with stochastic convex optimization methods in the average-cost setting, showing performance guarantees relative to a restricted policy class; Lakshmi-

narayanan et al. (2017) proposed the linearly relaxed ALP (LRALP), which alleviates computational load by projecting constraints into a lower-dimensional subspace while controlling approximation error. More recent developments have advanced convex formulations such as convex Q-learning (Lu et al., 2021; 2022), logistic Q-learning (Bas-Serrano et al., 2021), and primal-dual algorithms (Wang and Chen, 2016; Lee and He, 2018; Serrano and Neu, 2020; Neu and Okolo, 2023) with growing attention in offline RL where environment interaction is limited (Nachum and Dai, 2020; Zhan et al., 2022; Ozdaglar et al., 2023; Gabbianelli et al., 2024). In parallel, optimization and RL communities have explored barrier-based techniques: the log-barrier method is a classical tool in convex optimization (Boyd and Vandenberghe, 2004; Nesterov, 2018; Bertsekas, 1999), and has recently been adapted for safe RL, e.g., Zhang et al. (2024a;b) introduced a constrained soft actor-critic variant using a smoothed log-barrier for stable constraint handling in continuous control tasks. Despite these advances, existing LP-based approaches have not leveraged barrier functions to resolve inequality-constrained formulations, and existing barrier-based RL methods have not been applied to the LP representation of MDPs. Our work closes this gap by introducing a log-barrier reformulation of the LP approach to MDPs, yielding an unconstrained objective amenable to gradient-based optimization while retaining the structural advantages of LP formulations.

## 3 PRELIMINARIES

### 3.1 MARKOV DECISION PROBLEM

We consider the infinite-horizon discounted Markov decision problem (Puterman, 2014) and Markov decision process, where the agent sequentially takes actions to maximize cumulative discounted rewards. In a Markov decision process with the state-space $\mathcal{S} := \{1, 2, \ldots, |\mathcal{S}|\}$ and action-space $\mathcal{A} := \{1, 2, \ldots, |\mathcal{A}|\}$, where $|\mathcal{S}|$ and $|\mathcal{A}|$ denote cardinalities of each set, the decision maker selects an action $a \in \mathcal{A}$ at the current state $s \in \mathcal{S}$, then the state transits to the next state $s' \in \mathcal{S}$ with probability $P(s'|s, a)$, and the transition incurs a reward $r(s, a, s') \in \mathbb{R}$, where $P(s'|s, a)$ is the state transition probability from the current state $s \in \mathcal{S}$ to the next state $s' \in \mathcal{S}$ under action $a \in \mathcal{A}$, and $r(s, a, s')$ is the reward function. For convenience, we consider a deterministic reward function and simply write $r(s_k, a_k, s_{k+1}) =: r_{k+1}, k \in \{0, 1, \ldots\}$. A deterministic policy, $\pi : \mathcal{S} \to \mathcal{A}$, maps a state $s \in \mathcal{S}$ to an action $\pi(s) \in \mathcal{A}$. The objective of the Markov decision problem is to find an optimal policy, $\pi^*$, such that the cumulative discounted rewards over infinite time horizons is maximized, i.e., $\pi^* := \arg\max_{\pi \in \Theta} \mathbb{E}\left[\sum_{k=0}^{\infty} \gamma^k r_{k+1} \,\middle|\, \pi\right]$, where $\gamma \in [0, 1)$ is the discount factor, $\Theta$ is the set of all deterministic policies, $(s_0, a_0, s_1, a_1, \ldots)$ is a state-action trajectory generated by the Markov chain under policy $\pi$, and $\mathbb{E}[\cdot|\pi]$ is an expectation conditioned on the policy $\pi$. Moreover, Q-function under policy $\pi$ is defined as $Q^\pi(s, a) = \mathbb{E}\left[\sum_{k=0}^{\infty} \gamma^k r_{k+1} \,\middle|\, s_0 = s, a_0 = a, \pi\right], (s, a) \in \mathcal{S} \times \mathcal{A}$, and the optimal Q-function is defined as $Q^*(s, a) = Q^{\pi^*}(s, a)$ for all $(s, a) \in \mathcal{S} \times \mathcal{A}$. Once $Q^*$ is known, then an optimal policy can be retrieved by the greedy policy $\pi^*(s) = \arg\max_{a \in \mathcal{A}} Q^*(s, a)$. Throughout, we assume that the Markov decision process is ergodic so that the stationary state distribution exists. In this paper, we define an upper bound of the reward function as $|r(s, a, s')| \leq r_{\max}, (s, a, s') \in \mathcal{S} \times \mathcal{A} \times \mathcal{S}$.

### 3.2 LP FORMULATION OF MDP BASED ON Q-FUNCTION

In this paper, for the sake of clarity and brevity, the majority of technical proofs are presented in Appendix. It is well known that a Markov decision problem (MDP) can be formulated as a linear program (LP) (Luenberger et al., 1984; Puterman, 2014). While the LP formulation is typically expressed in terms of the value function (Puterman, 2014), one can also consider an LP formulation based on the Q-function. To this end, let us consider the following LP:

$$\min_{Q \in \mathbb{R}^{|\mathcal{S}||\mathcal{A}|}} \sum_{(s,a) \in \mathcal{S} \times \mathcal{A}} \rho(s, a) Q(s, a) \tag{1}$$

subject to

$$R(s, a) + \gamma \sum_{s' \in \mathcal{S}} P(s'|s, a) Q(s', a') \leq Q(s, a), \quad (s, a, a') \in \mathcal{S} \times \mathcal{A} \times \mathcal{A},$$

where $R(s, a)$ is the expected reward conditionend on $(s, a) \in \mathcal{S} \times \mathcal{A}$, $\rho$ denotes any probability distribution over $\mathcal{S} \times \mathcal{A}$ with strictly positive support. For convenience, in this paper, we define the

following Bellman operators

$$(TQ)(s,a) := R(s,a) + \gamma \sum_{s' \in \mathcal{S}} P(s'|s,a) \max_{a' \in \mathcal{A}} Q(s',a')$$

$$(FQ)(s,a,a') := R(s,a) + \gamma \sum_{s' \in \mathcal{S}} P(s'|s,a) Q(s',a').$$

Note that we can always find a strictly feasible solution of the above LP. For instance, if we choose $Q(s,a) = \frac{r_{\max} + \varepsilon}{1 - \gamma} > 0, (s,a) \in \mathcal{S} \times \mathcal{A}$ with any $\varepsilon > 0$, then $R(s,a) + \gamma \sum_{s' \in \mathcal{S}} P(s'|s,a) Q(s',a') - Q(s,a) = R(s,a) - r_{\max} - \varepsilon < 0$. The LP formulation in Equation (1) constructed on the basis of the Q-function (Lee and He, 2019; 2018) has not been extensively studied compared to the LP formulation based on the value function. Although the LP formulation involving the Q-function was considered in Lee and He (2019; 2018), it differs significantly from the LP discussed above. In particular, the LP in Lee and He (2019; 2018) involves not only the Q-function but also an additional value function, thereby employing a somewhat more indirect approach compared to the above formulation. Accordingly, we begin by briefly introducing several theoretical properties and interpretations of this formulation, prior to presenting our main results. As a preliminary but fundamental result, it is straightforward to show that the solution of the above LP is unique and corresponds to the optimal Q-function, $Q^*$ (Section A). In addition, the dual (Boyd and Vandenberghe, 2004, Chapter 5) of the above LP can be derived in the following form (Section B).

**Lemma 1.** *The dual problem of the LP in Equation* (1) *is given by*

$$\max_{\lambda \geq 0} \sum_{(s,a,a') \in \mathcal{S} \times \mathcal{A} \times \mathcal{A}} \lambda(s,a,a') R(s,a) \tag{2}$$

subject to

$$\sum_{i \in \mathcal{A}} \lambda(s,a,i) - \gamma \sum_{(i,j) \in \mathcal{S} \times \mathcal{A}} P(s|i,j) \lambda(i,j,a) = \rho(s,a), \quad \forall (s,a) \in \mathcal{S} \times \mathcal{A}. \tag{3}$$

The original LP in Equation (1) is referred to as the primal LP (or primal problem), while the above LP in Equation (2) is called the dual LP (or dual problem). The variable $Q$ in the primal LP is referred to as the primal variable, while the variable $\lambda$ in the dual LP is called the dual variable. We can examine several important properties and interpretations of the dual LP. For instance, the optimal dual variable $\lambda^*$ corresponds to a probability distribution, which represents the stationary state–action–next-action distribution under the optimal policy $\pi^*$ constructed from the dual variable as follows:

$$\pi^*(\cdot|s) := \left[ \begin{array}{cccc} \frac{\lambda^*(s,1)}{\sum_{a \in \mathcal{A}} \lambda^*(s,a)} & \frac{\lambda^*(s,2)}{\sum_{a \in \mathcal{A}} \lambda^*(s,a)} & \cdots & \frac{\lambda^*(s,|\mathcal{A}|)}{\sum_{a \in \mathcal{A}} \lambda^*(s,a)} \end{array} \right],$$

where $\lambda^*(s,a) := \sum_{a' \in \mathcal{A}} \lambda^*(s,a,a')$. We note that when the optimal policy is deterministic, then the above policy becomes a one-hot vector indicating the optimal action (Chen and Wang, 2016). Similarly, if $Q^*$ is the primal optimal solution (the solution of Equation (1)), then $\beta^*(s) := \arg \max_{a \in \mathcal{A}} Q^*(s,a)$ likewise induces an optimal policy. Additional details can be found in Appendix (Sections C and D).

## 4 LOG-BARRIER FUNCTION APPROACH

In the previous section, we provided a brief discussion of the LP formulation in Equation (1) based on the Q-function. We now turn to the main results of this paper. First of all, note that Equation (1) involves inequality constraints. A common approach to handling such constraints is to introduce the Lagrangian together with Lagrange multipliers, or dual variables (Bertsekas, 1999; Boyd and Vandenberghe, 2004). One then seeks the primal and dual variables through first-order primal–dual iterations (Kojima et al., 1989). However, this method typically suffers from slow convergence and does not, in general, guarantee convergence in many practical settings (Applegate et al., 2021). Another practical and widely used approach to handling inequality constraints is the use of barrier functions, most notably the log-barrier function (Boyd and Vandenberghe, 2004). This method imposes a large (or even infinite) penalty on variables that violate the inequality constraints, thereby forcing the iterates to remain within the feasible region while enabling the optimization problem to

be solved. In this paper, we apply the log-barrier function to the LP formulation in Equation (1), and we undertake an in-depth study and interpretation of this barrier-based approach to solving MDPs. This line of research represents an approach that has not yet been addressed in the existing literature.

The log-barrier function is a classical tool in constrained optimization used to handle inequality constraints. For a constraint of the form $g(x) \leq 0$, the log-barrier introduces a penalty term $\eta\varphi(x) := -\eta \ln(-g(x))$, where $\eta > 0$ is a barrier parameter. This function approaches infinity as $g(x)$ gets close to zero, and thus, it prevents the iterates from leaving the feasible region. As $\eta$ decreases, the solution of the barrier-augmented problem converges to the solution of the original constrained optimization problem. Moreover, one can prove that the log-barrier function $\varphi(x) := -\ln(-x)$ is strictly convex in its domain $\{x \in \mathbb{R} : x < 0\}$. Using the log-barrier function, the inequality constraints can be integrated into a single objective function as follows:

$$f_\eta(Q) := \sum_{(s,a)\in\mathcal{S}\times\mathcal{A}} Q(s,a)\rho(s,a) + \eta \sum_{(s,a,a')\in\mathcal{S}\times\mathcal{A}\times\mathcal{A}} w(s,a,a')\varphi\left((FQ)(s,a,a') - Q(s,a)\right),$$

(4)

where $\eta > 0$ is the barrier parameter (weight) and $w(s,a,a') > 0, (s,a,a') \in \mathcal{S} \times \mathcal{A} \times \mathcal{A}$ are weight parameters of the inequality constraints, which are introduced in order to consider random sampling approaches in stochastic implementations later. For instance, in the deep-RL variant later, $w$ should be interpreted as the empirical distribution of state–action pairs induced by the replay buffer or by mini-batch sampling. Moreover, we can also set $w(s,a,a') = 1$ for all $(s,a,a') \in \mathcal{S} \times \mathcal{A} \times \mathcal{A}$. The objective function $f_\eta$ has the following domain:

$$\mathcal{D} := \left\{Q \in \mathbb{R}^{|\mathcal{S}||\mathcal{A}|} : (FQ)(s,a,a') - Q(s,a) < 0, (s,a,s') \in \mathcal{S} \times \mathcal{A} \times \mathcal{A}\right\},$$

which can be also seen as the strictly feasible set. Moreover, it can be shown that $\mathcal{D}$ is convex, open, bounded below, and unbounded above. Moreover, the objective function $f_\eta$ is strictly convex in $\mathcal{D}$. To proceed, let us define the level set of the objective function for $c > 0$

$$\mathcal{L}_c := \{Q \in \mathcal{D} : f_\eta(Q) \leq c\}.$$

We can also establish that $\mathcal{L}_c$ is convex, closed, bounded, and $f_\eta$ is strongly convex and has Lipschitz continuous gradient in $\mathcal{L}_c$ with any $c > 0$. The detailed theoretical analysis is given in Sections E to G.

The introduction of the log-barrier function enables us to reformulate the MDP as an unconstrained optimization problem. Consequently, a natural approach to address this problem is to employ gradient descent. For this purpose, we first establish the closed-form expression of the gradient presented in the following lemma, which can be proved via direct calculations.

**Lemma 2.** *The gradient of $f_\eta(Q)$ for $Q \in \mathcal{D}$ is given by*

$$(\nabla_Q f_\eta(Q))(s,a) = \rho(s,a) + \gamma \sum_{(s',a')\in\mathcal{S}\times\mathcal{A}} P(s|s',a')\lambda_\eta(s',a',a) - \sum_{a'\in\mathcal{A}} \lambda_\eta(s,a,a')$$

*for $(s,a) \in \mathcal{S} \times \mathcal{A}$, where $\lambda_\eta(s,a,a') := \frac{\eta w(s,a,a')}{Q(s,a)-(FQ)(s,a,a')}$.*

Using the closed-form gradient obtained above, we can gain insight into the solution of the optimization problem. To this end, let us assume that $\tilde{Q}_\eta$ is a minimizer of $f_\eta(Q)$

$$\tilde{Q}_\eta := \arg\min_{Q\in\mathcal{D}} f_\eta(Q).$$

The corresponding first-order optimality condition, $\nabla_Q f(Q)|_{Q=\tilde{Q}_\eta} = 0$, is given by

$$\left(\nabla_Q f(Q)|_{Q=\tilde{Q}_\eta}\right)(s,a) = \rho(s,a) + \gamma \sum_{(s',a')\in\mathcal{S}\times\mathcal{A}} \tilde{\lambda}_\eta(s',a',a)P(s|s',a') - \sum_{a'\in S} \tilde{\lambda}_\eta(s,a,a')$$

$$= 0, \quad (s,a) \in \mathcal{S} \times \mathcal{A}.$$

where $\tilde{\lambda}_\eta(s,a,a') := \frac{\eta w(s,a,a')}{\tilde{Q}(s,a)-(F\tilde{Q}_\eta)(s,a,a')}$.

As mentioned earlier, since $f_\eta$ is strictly convex over the domain, this first-order condition constitutes the necessary and sufficient condition for the optimal solution. We can observe that the above

equation is exactly identical to the equality constraints in the dual problem with $\tilde{\lambda}_\eta$ as the dual variables. In other words, $\tilde{\lambda}_\eta$ approximates the true optimal dual variable $\lambda^*$. Therefore, from the above solution $\tilde{Q}_\eta$, we can consider two types of policies. Interpreting the solution as an approximate solution to the LP formulation of the MDP, we may derive a greedy policy based on $\tilde{Q}_\eta$, as well as a policy constructed from the approximate dual variable $\tilde{\lambda}_\eta$ which also depends on $\tilde{Q}_\eta$. Hereafter, we refer to the policy induced by the primal optimal solution $\tilde{Q}_\eta$ as the primal $\eta$-policy, and the policy induced by the dual optimal solution $\tilde{\lambda}_\eta$ as the dual $\eta$-policy. In particular, the primal $\eta$-policy, which is deterministic, can be written as

$$\tilde{\beta}_\eta(s) := \arg\max_{a\in\mathcal{A}} \tilde{Q}_\eta(s,a),$$

and the dual $\eta$-policy, which is stochastic, can be written as

$$\tilde{\pi}_\eta(\cdot|s) := \left[ \begin{array}{cccc} \frac{\tilde{\lambda}_\eta(s,1)}{\sum_{a\in\mathcal{A}}\tilde{\lambda}_\eta(s,a)} & \frac{\tilde{\lambda}_\eta(s,2)}{\sum_{a\in\mathcal{A}}\tilde{\lambda}_\eta(s,a)} & \cdots & \frac{\tilde{\lambda}_\eta(s,|\mathcal{A}|)}{\sum_{a\in\mathcal{A}}\tilde{\lambda}_\eta(s,a)} \end{array} \right],$$

where $\tilde{\lambda}_\eta(s,a) := \sum_{a'\in\mathcal{A}}\tilde{\lambda}_\eta(s,a,a')$. By minimizing the above objective function, we can obtain an approximate solution of the primal LP. Since the objective function is strictly convex, the approximate solution can be efficiently found with a gradient descent algorithm. We can prove that, under certain mild conditions, this gradient descent with a constant step-size converges exponentially to $\tilde{Q}_\eta$ (Sections H and I).

Next, note that the minimizer of the log-barrier-based objective function provides only an approximate solution to the LP formulation of the MDP, rather than an exact one. Nevertheless, by decreasing the barrier parameter $\eta$, the solution is permitted to approach the boundary of the inequality constraints, i.e., the equality constraints, and thus progressively converges to the exact LP solution. In the limit as $\eta \to 0$, the solution converges to the true solution of the MDP. Building on this insight, we can express the error between $\tilde{Q}_\eta$ and $Q^*$ as a function of $\eta$. The following theorem establishes such an error bound between $\tilde{Q}_\eta$ and $Q^*$, and, in addition, presents a bound on the Bellman error corresponding to $\tilde{Q}_\eta$ (Section J).

**Theorem 1.** *We have*

*1.* $\eta \min_{(s,a,a')\in\mathcal{S}\times\mathcal{A}\times\mathcal{A}} w(s,a,a') < \left\| \tilde{Q}_\eta - Q^* \right\|_\infty \leq \frac{\eta \sum_{(s,a,a')\in\mathcal{S}\times\mathcal{A}\times\mathcal{A}} w(s,a,a')}{\min_{(s,a)\in\mathcal{S}\times\mathcal{A}}\rho(s,a)}$

*2.* $\eta(1-\gamma) \min_{(s,a,a')\in\mathcal{S}\times\mathcal{A}\times\mathcal{A}} w(s,a,a') < \left\| \tilde{Q}_\eta - T\tilde{Q}_\eta \right\|_\infty \leq \frac{(1+\gamma)\eta \sum_{(s,a,a')\in\mathcal{S}\times\mathcal{A}\times\mathcal{A}} w(s,a,a')}{\min_{(s,a)\in\mathcal{S}\times\mathcal{A}}\rho(s,a)}$

From the above result, we can establish both upper and lower bounds on the $l_\infty$-norm of the optimality error and the Bellman error. Both bounds depend on $\eta$ and are shown to be linear functions of $\eta$. Hence, as $\eta \to 0$, the upper and lower bounds of the error norms decrease linearly, i.e., $\tilde{Q}_\eta \to Q^*$. Moreover, these bounds also reveal the interplay between the bounds and other hyperparameters. In the above result, we established bounds on the norm of the error. In addition, we can also derive upper and lower bounds on the MDP objective function itself. The following theorem presents such bounds for the objective functions corresponding to the primal $\eta$-policy and the dual $\eta$-policy, expressed relative to the optimal objective value $J^{\pi^*}$ (Section K).

**Theorem 2.** *We have*

*1.* $J^{\pi^*} - \eta \sum_{(s,a,a')\in\mathcal{S}\times\mathcal{A}\times\mathcal{A}} w(s,a,a') \leq J^{\tilde{\pi}_\eta} \leq J^{\pi^*}$

*2.* $J^{\pi^*} - \frac{\eta(1+\gamma) \sum_{(s,a,a')\in\mathcal{S}\times\mathcal{A}\times\mathcal{A}} w(s,a,a')}{(1-\gamma)\min_{(s,a)\in\mathcal{S}\times\mathcal{A}}\rho(s,a)} \leq J^{\tilde{\beta}_\eta} \leq J^{\pi^*}$

*3.* $-\frac{\eta(1+\gamma) \sum_{(s,a,a')\in\mathcal{S}\times\mathcal{A}\times\mathcal{A}} w(s,a,a')}{(1-\gamma)\min_{(s,a)\in\mathcal{S}\times\mathcal{A}}\rho(s,a)} \leq J^{\tilde{\beta}_\eta} - J^{\tilde{\pi}_\eta} \leq \eta \sum_{(s,a,a')\in\mathcal{S}\times\mathcal{A}\times\mathcal{A}} w(s,a,a')$

The above theorem shows that the upper and lower bounds of the objective functions corresponding to the primal $\eta$-policy and the dual $\eta$-policy also depend linearly on $\eta$. Hence, as $\eta \to 0$, the objective functions associated with both the primal and dual $\eta$-policies converge to the optimal objective value $J^*$.

## 5 POLICY EVALUATION

Although the primary focus of this paper is on computing the optimal Q-function, the proposed framework is equally applicable to policy evaluation. In this section, we therefore provide a brief account of its application to the policy evaluation setting. To this end, let us assume a given policy $\pi$, and consider the problem of finding its corresponding Q-function $Q^\pi$. This problem can be solved through the following LP:

$$\min_{Q \in \mathbb{R}^{|\mathcal{S}||\mathcal{A}|}} \sum_{(s,a) \in \mathcal{S} \times \mathcal{A}} \rho(s,a) Q(s,a)$$

$$\text{subject to} \quad (T^\pi Q)(s,a) \leq Q(s,a), \quad (s,a) \in \mathcal{S} \times \mathcal{A},$$

where $R(s,a)$ is the expected reward conditioned on $(s,a) \in \mathcal{S} \times \mathcal{A}$, $\rho$ denotes any probability distribution over $\mathcal{S} \times \mathcal{A}$ with strictly positive support, and

$$(T^\pi Q)(s,a) := R(s,a) + \gamma \sum_{(s',a') \in \mathcal{S} \times \mathcal{A}} P(s'|s,a)\pi(a'|s')Q(s',a'), \quad (s,a) \in \mathcal{S} \times \mathcal{A}.$$

Then, analogous to the case of finding the optimal Q-function, we can derive the following results through a similar theoretical analysis. In particular, we can prove that the unique optimal solution of the above LP is $Q^\pi$. Moreover, let us consider the objective function with log-barrier function

$$f_\eta^\pi(Q) := \sum_{(s,a) \in \mathcal{S} \times \mathcal{A}} Q(s,a)\rho(s,a) + \eta \sum_{(s,a) \in \mathcal{S} \times \mathcal{A}} w(s,a)\varphi\left((T^\pi Q)(s,a) - Q(s,a)\right)$$

where $\eta > 0$ is a weight parameter and $w(s,a) > 0, (s,a) \in \mathcal{S} \times \mathcal{A}$ are weight parameters of the inequality constraints. Then, we can prove that the corresponding optimal solution $\tilde{Q}_\eta^\pi := \arg\min_{Q \in \mathcal{D}} f_\eta^\pi(Q)$ approximates $Q^\pi$. Many of the analytical results established in the previous section can be applied in a similar manner. For brevity, all related details are provided in Section L.

## 6 DEEP RL VARIANTS

Thus far, we have provided a theoretical analysis of the approximate solution to the LP formulation of MDPs using the log-barrier function. Although the primary focus of this paper is on theoretical analysis with tabular setting, in this section we further explore the potential extension of the proposed framework to deep RL. In particular, we introduce a novel DQN algorithm inspired by the idea of standard DQN (Mnih et al., 2015). Note that when a deep neural network is used, the model becomes a nonlinear function of the parameters $\theta$, and the precise theoretical results derived for the tabular setting no longer apply directly. Nevertheless, the tabular analysis in the previous sections provides useful intuition and insights on deep RL extensions. Similar to the conventional DQN framework, we employ an experience replay buffer $D$ and mini-batch sampling $B$. Furthermore, in the definition of $f_\eta$, the probability density is replaced with samples from the mini-batch, which leads to the following loss function:

$$L(\theta) := \frac{1}{|B|} \sum_{(s,a,r,s') \in B, a' \in \mathcal{A}} [Q_\theta(s,a) + \eta\varphi(r + \gamma Q_\theta(s',a') - Q_\theta(s,a))],$$

where $B$ is a mini-batch uniformly sampled from the experience replay buffer $D$, $|B|$ is the size of the mini-batch, $Q_\theta$ is a deep neural network approximation of Q-function, $\theta \in \mathbb{R}^m$ is the parameter to be determined, and $(s,a,r,s')$ is the transition sample of the state-action-reward-next state. The loss function can be seen as a stochastic approximation of $f_\eta$, where $\rho$ and $w$ can be set to probability distributions corresponding to the replay buffer. However, this stochastic approximation is generally biased because it approximates a function in which the state transition probabilities appear outside the log-barrier function. In fact, by applying Jensen's inequality, we can show that the above loss function is essentially an unbiased stochastic approximation of an upper bounding surrogate funciton of $f_\eta$ (Section M). However, note that in the

deterministic case, the upper surrogate function coincides with the true objective function with zero Jenson gap. In this setting, $L(\theta)$ becomes an unbiased stochastic approximation of $f_\eta$. For example, a dynamical system expressed as $s_{k+1} = f(s_k, a_k)$ is deterministic, and hence the upper bound coincides with $f_\eta$. Therefore, the loss function $L$ can be regarded as an unbiased stochastic approximation of $f_\eta$ as follows: $f_\eta(Q_\theta) = \sum_{(s,a)\in\mathcal{S}\times\mathcal{A}} Q_\theta(s,a)\rho(s,a) + \eta\sum_{(s,a,a')\in\mathcal{S}\times\mathcal{A}\times\mathcal{A}} w(s,a,a')\varphi\left(r(s,a,f(s,a)) + \gamma Q_\theta(f(s,a),a') - Q_\theta(s,a)\right) \cong L(\theta)$. Another difference from the conventional DQN is that the algorithm proposed in this paper does not employ target variables, and hence the update of target variables is also omitted. Apart from this distinction, the remaining components are similar to those of standard DQN. The overall pseudocode of the algorithm and implementation details for stabilization of the algorithm are summarized in Section N.

Next, we briefly discuss the extension of policy evaluation, previously introduced, to the deep RL setting. In particular, we consider its potential application to DDPG (Lillicrap et al., 2015). Here, the policy is deterministic, and the following loss function can be formulated:

$$L_{\text{critic}}(\theta; \pi_\chi) := \frac{1}{|B|} \sum_{(s,a,r,s')\in B} [Q_\theta(s,a) + \eta\varphi(r + \gamma Q_\theta(s', \pi_\chi(s')) - Q_\theta(s,a))],$$

where $\pi_\chi$ denotes the deterministic policy being learned, and $\chi$ is its parameter vector. A discussion similar to that of the preceding loss function applies here, and the corresponding modified DDPG algorithm along with its implementation details is provided in Section O. Finally, we note that we can heuristically apply several alternative choices (e.g., SoftPlus) instead of the log-barrier function. While these alternatives often yield reasonable performance, the proposed method typically performed better in our experiments.

## 7 EXPERIMENTS

To validate the performance of the proposed method, we conduct experiments in both discrete and continuous control environments. We implement our deep RL variant as described in Section 6, naming our algorithms **Log-barrier DQN** and **Log-barrier DDPG**, respectively. For discrete control tasks, we compare our algorithm with the standard DQN (Mnih et al., 2015), and for continuous control tasks, we benchmark it against the standard DDPG (Lillicrap et al., 2015). Additional algorithmic details are provided in Section N and Section O, with detailed hyperparameters listed in Section P.

**Log-Barrier DQN** For our DQN experiments, we chose five environments form the Gymnasium library (Acrobot-v1, CartPole-v1, LunarLander-v3, MountainCar-v0, and Pendulum-v1). For MountainCar-v0, we replace the original sparse reward with a dense shaping reward, and for Pendulum-v1 we discretize the continuous action space to enable DQN training. As shown in Figure 1, the results reveal a remarkable point that the Log-barrier DQN demonstrates rapid adaptation and significant stability in the CartPole environment, where the agent's survival is threatened by a critical angle criterion. We hypothesize that this is due to the sharp decision boundary between states where the agent survives and states where the episode is terminated. Standard DQN, which relies on Bellman updates with an mean square error (MSE) loss, can suffer from error propagation across this boundary. In contrast, the proposed approach, which uses the LP form, can globally mitigate this hazard. Instead of estimating value from neighboring states, our method directly minimizes its objective while satisfying the LP constraints.

**Log-Barrier DDPG** In the continuous control experiments, we applied the proposed method for the policy evaluation task in DDPG. The resulting DDPG variant demonstrated superior performance on four complex MuJoCo environments (Ant, Walker2d, HalfCheetah, Humanoid), while showing no significant advantage on the simpler Hopper task, as shown in Figure 2. We attribute this success to a fundamental property of critic update mechanism of the proposed algorithm. We conjecture that the core of the advantage arises from the fact that the LP form is inherently a minimization objective, which naturally counteracts the Q-value overestimation bias prevalent in actor-critic methods. Standard DDPG critics learn by minimizing the MSE to a target value, a process that simply follows the target without regard for its potential bias. If the target is inflated, the critic learns an inflated value, leading to a feedback loop of compounding overestimation. In contrast, the proposed approach minimizes the value function itself, subject to the constraints of Bellman consistency. This systematic

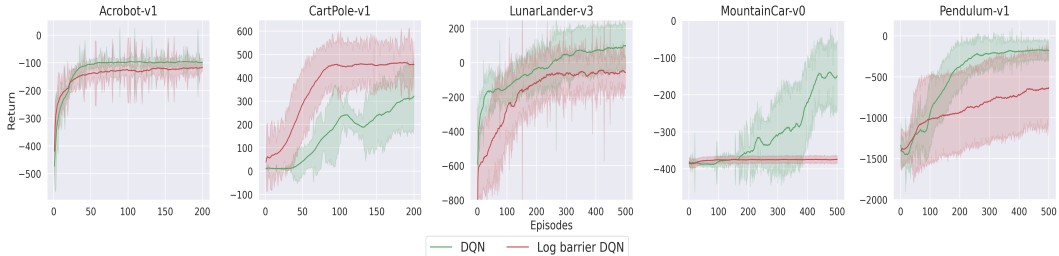

Figure 1: Learning curves comparing the Log-barrier DQN and standard DQN on the Gymnasium control environments. Each curve represents the average return over 10 random seeds, with the shaded area indicating one standard deviation from the mean.

search for the tightest and lowest possible Q-values that still satisfy the dynamics acts as a powerful, implicit regularizer against optimistic value estimates. This results in a more conservative and stable critic, which provides a more reliable gradient to the actor, leading to superior final performance. Consequently, this allows the proposed method to overcome the limitations of standard DDPG, and to successfully solve environments such as Ant and Humanoid that were previously thought to be beyond its capabilities.

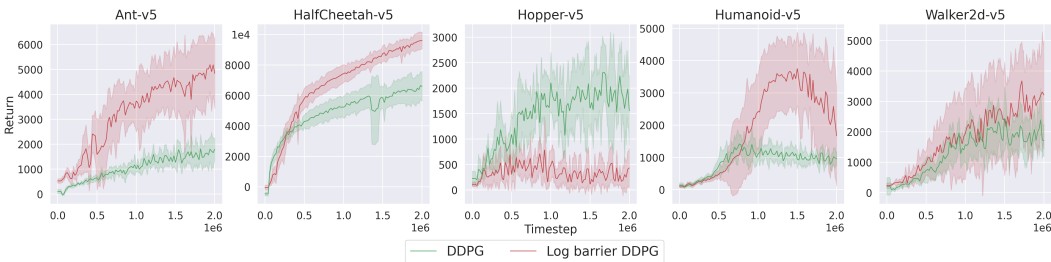

Figure 2: Learning curves comparing the Log-barrier DDPG and standard DDPG on the Mujoco continuous control environments. Each curve represents the average return over 8 random seeds, with the shaded area indicating one standard deviation from the mean.

## 8 CONCLUSION

In this paper, we have developed a theoretical framework for solving MDPs via their LP formulation using a log-barrier term. Reformulating the LP into a single objective $f_\eta$, we have showed that gradient descent efficiently produces approximate solutions $\tilde{Q}_\eta$ and prove error bounds (including $\|\tilde{Q}_\eta - Q^*\|_\infty$) that scale linearly with the barrier parameter $\eta$. We have also characterized primal and dual approximate solutions, their induced policies, and proved structural properties of $f_\eta$ (e.g., convexity, local strong convexity/Lipschitzness) together with exponential convergence of gradient descent in the tabular setting. Practically, we have derived a novel log-barrier loss for deep RL and evaluate it in DQN and DDPG: the method matches standard DQN in most cases and outperforms conventional DDPG in several tasks. While experiments are limited in scope and the approach requires careful hyperparameter tuning, the empirical results support the promise of the log-barrier formulation; fuller large-scale validation and robustness improvements remain important directions for future work.

