## ACKNOWLEDGMENTS

The work was supported by the Institute of Information Communications Technology Planning Evaluation (IITP) funded by the Korea government under Grant 2022-0-00469, and the BK21 FOUR from the Ministry of Education (Republic of Korea).

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

# Appendix

## A  APPENDIX: LEMMA 3

In the next lemma, we establish that the optimal solution to the LP formulation of the MDP Equation (1), as presented in this paper, is precisely the optimal Q-function, $Q^*$.

**Lemma 3.** *The optimal solution of the LP Equation* (1) *is unique and given by* $Q^*$

*Proof.* Let us consider the following LP defined also in Equation (1):

$$\min_{Q \in \mathbb{R}^{|\mathcal{S}||\mathcal{A}|}} \sum_{(s,a) \in \mathcal{S} \times \mathcal{A}} \rho(s,a) Q(s,a)$$

subject to

$$R(s,a) + \gamma \sum_{s' \in \mathcal{S}} P(s'|s,a) Q(s',a') \leq Q(s,a), \quad (s,a,a') \in \mathcal{S} \times \mathcal{A} \times \mathcal{A},$$

where $\rho$ denotes any probability distribution over $\mathcal{S} \times \mathcal{A}$ with strictly positive support. Let us assume that the optimal solution of the above LP is $\hat{Q}$. We want to prove that $\hat{Q} = Q^*$. First of all, since $\hat{Q}$ is feasible, it satisfies

$$(F\hat{Q})(s,a,a') \leq \hat{Q}(s,a), \quad (s,a,a') \in \mathcal{S} \times \mathcal{A} \times \mathcal{A}.$$

This implies

$$(T\hat{Q})(s,a) \leq \hat{Q}(s,a), \quad (s,a) \in \mathcal{S} \times \mathcal{A},$$

and vice versa. Equivalently, it can be written as $T\hat{Q} \leq \hat{Q}$. Because $T$ is a monotone operator (Bertsekas and Tsitsiklis, 1996), we have

$$\lim_{k \to \infty} T^k \hat{Q} \leq \hat{Q} \Rightarrow Q^* \leq \hat{Q}.$$

Moreover, since $TQ^* = Q^*$, $Q^*$ is a feasible solution, and hence, $\hat{Q} \leq Q^*$. Therefore, $Q^* \leq \hat{Q} \leq Q^*$, which implies that $\hat{Q} = Q^*$.

$\square$

## B  APPENDIX: PROOF OF LEMMA 1

Let us consider the Lagrangian function (Boyd and Vandenberghe, 2004)

$$L(Q, \lambda) = \sum_{(s,a) \in \mathcal{S} \times \mathcal{A}} \rho(s,a) Q(s,a)$$
$$+ \sum_{(s,a,a') \in \mathcal{S} \times \mathcal{A} \times \mathcal{A}} \lambda(s,a,a')(R(s,a) + \gamma \sum_{s' \in \mathcal{S}} P(s'|s,a) Q(s',a') - Q(s,a)),$$

where $\lambda$ is the Lagrangian multiplier vector (or dual variable). Next, the function can be written by

$$L(Q, \lambda) = \sum_{(s,a,a') \in \mathcal{S} \times \mathcal{A} \times \mathcal{A}} \lambda(s,a,a') R(s,a)$$
$$+ \sum_{(s,a) \in \mathcal{S} \times \mathcal{A}} Q(s,a) \left\{ \sum_{i \in \mathcal{A}} \lambda(s,a,i) - \gamma \sum_{(i,j) \in \mathcal{S} \times \mathcal{A}} P(s|i,j) \lambda(i,j,a) - \rho(s,a) \right\}$$

One can observe that the optimal value of the dual problem $\max_{\lambda \geq 0} \min_{Q \in \mathbb{R}^{|\mathcal{S}||\mathcal{A}|}} L(Q, \lambda)$ is finite only when

$$\sum_{i \in \mathcal{A}} \lambda(s,a,i) - \gamma \sum_{(i,j) \in \mathcal{S} \times \mathcal{A}} P(s|i,j) \lambda(i,j,a) - \rho(s,a) = 0, \quad \forall (s,a) \in \mathcal{S} \times \mathcal{A}.$$

Therefore, one gets the following dual problem:

$$\max_{\lambda \geq 0} \sum_{(s,a,a') \in \mathcal{S} \times \mathcal{A} \times \mathcal{A}} \lambda(s,a,a')R(s,a)$$

subject to

$$\sum_{i \in \mathcal{A}} \lambda(s,a,i) - \gamma \sum_{(i,j) \in \mathcal{S} \times \mathcal{A}} P(s|i,j)\lambda(i,j,a) = \rho(s,a), \quad \forall(s,a) \in \mathcal{S} \times \mathcal{A}.$$

This completes the proof.

## C  APPENDIX: THEOREM 3

We examine several important properties and interpretations of the dual LP.

**Theorem 3.** *Suppose $\lambda(s,a,a'), \forall(s,a,a') \in \mathcal{S} \times \mathcal{A} \times \mathcal{A}$ is any dual feasible solution (any solutions satisfying the dual constraints in Equation (3)). Then, we obtain the following results:*

1. *$(1-\gamma)\lambda$ is a probability distribution, i.e.,*

$$(1-\gamma)\lambda(s,a,a') \geq 0, \quad \forall(s,a,a') \in \mathcal{S} \times \mathcal{A} \times \mathcal{A}, \qquad \sum_{(s,a,a') \in \mathcal{S} \times \mathcal{A} \times \mathcal{A}} (1-\gamma)\lambda(s,a,a') = 1$$

2. *Moreover, let us define the marginal distributions*

$$\lambda(s,a) := \sum_{a' \in \mathcal{A}} \lambda(s,a,a'), \quad \lambda(s) := \sum_{a \in \mathcal{A}} \lambda(s,a), \quad \rho(s) := \sum_{a' \in \mathcal{A}} \rho(s,a). \qquad (5)$$

*Then, we have*

$$\lambda(s') - \gamma \sum_{s \in \mathcal{S}} P^\pi(s'|s)\lambda(s) = \rho(s'), \quad \forall s' \in \mathcal{S}, \qquad (6)$$

*where*

$$\pi(\cdot|s) := \begin{bmatrix} \frac{\lambda(s,1)}{\sum_{a \in \mathcal{A}} \lambda(s,a)} & \frac{\lambda(s,2)}{\sum_{a \in \mathcal{A}} \lambda(s,a)} & \cdots & \frac{\lambda(s,|\mathcal{A}|)}{\sum_{a \in \mathcal{A}} \lambda(s,a)} \end{bmatrix} \qquad (7)$$

*and*

$$P^\pi(s'|s) := \sum_{a \in \mathcal{A}} P(s'|s,a)\pi(a|s)$$

*is the probability of the transition from $s$ to $s'$ under the policy $\pi$.*

3. *If the marginal $\rho(s) := \sum_{a \in \mathcal{A}} \rho(s,a), s \in \mathcal{S}$ represents the initial state distribution, then $(1-\gamma)\lambda(\cdot)$ is the discounted state occupation probability distribution defined as*

$$(1-\gamma)\lambda(s) = \mu^\pi(s) = (1-\gamma)\sum_{k=0}^{\infty} \gamma^k \mathbb{P}[s_k = s|\pi, \rho]$$

4. *If the marginal $\rho(s) := \sum_{a \in \mathcal{A}} \rho(s,a), s \in \mathcal{S}$ represents the initial state distribution, then the dual objective function satisfies*

$$\sum_{(s,a,a') \in \mathcal{S} \times \mathcal{A} \times \mathcal{A}} \lambda(s,a,a')R(s,a) = \sum_{s \in \mathcal{S}} V^\pi(s)\rho(s) = J^\pi$$

Theorem 3 provides an interpretation of any feasible dual solution. In particular, analogous to the LP formulation (Puterman, 2014) based on the value function, the dual variable in the Q-function-based LP formulation corresponds to a probability distribution. It represents the stationary state–action–next-action distribution under the policy $\pi$ in Equation (7) constructed from the dual variable. Moreover, we can prove that the marginal distribution $\lambda(s)$ in Equation (5) corresponds to the discounted state-occupation probability distribution. Building on this observation, it can be shown that the dual objective function coincides with the objective value attained by the policy

$\pi$ in Equation (7). The preceding discussions are valid for any dual feasible variable $\lambda$, and, of course, the same properties extend to the dual optimal solution $\lambda^*$. In particular, if $\lambda^*$ is the dual optimal solution, then the associated policy Equation (7) with $\lambda$ replaced by $\lambda^*$ coincides with an optimal policy. Similarly, if $Q^*$ is the primal optimal solution (the solution of Equation (1)), then $\beta^*(s) := \arg\max_{a \in \mathcal{A}} Q^*(s, a)$ likewise induces an optimal policy.

*Proof.* For the first statement, summing the dual constraints in Equation (3) over $(s, a) \in \mathcal{S} \times \mathcal{A}$, we get

$$\sum_{(s,a,i) \in \mathcal{S} \times \mathcal{A} \times \mathcal{A}} \lambda(s, a, i) - \gamma \sum_{(s,a) \in \mathcal{S} \times \mathcal{A}, (i,j) \in \mathcal{S} \times \mathcal{A}} P(s|i, j) \lambda(i, j, a) = \sum_{(s,a) \in \mathcal{S} \times \mathcal{A}} \rho(s, a),$$

which leads to

$$(1 - \gamma) \sum_{a \in \mathcal{A}, (i,j) \in \mathcal{S} \times \mathcal{A}} \lambda(i, j, a) = 1.$$

It proves the first statement. For the second statement, summing the dual constraints in Equation (3) over $a' \in \mathcal{A}$ leads to

$$\lambda(s, a) - \gamma \sum_{(i,j) \in \mathcal{S} \times \mathcal{A}} P(s|i, j) \lambda(i, j, a) = \rho(s, a).$$

Moreover, summing the above equation over $a \in \mathcal{A}$ leads to

$$\sum_{a \in \mathcal{A}} \lambda(s', a) - \gamma \sum_{(s,a) \in \mathcal{S} \times \mathcal{A}} P(s'|s, a) \lambda(s, a) = \rho(s'),$$

which can be further written as

$$\sum_{a \in A} \lambda(s', a) - \gamma \sum_{(s,a) \in \mathcal{S} \times \mathcal{A}} P(s'|s, a) \frac{\lambda(s, a)}{\sum_{i \in \mathcal{A}} \lambda(s, i)} \sum_{j \in \mathcal{A}} \lambda(s, j)$$

$$= \sum_{a \in \mathcal{A}} \lambda(s', a) - \gamma \sum_{(s,a) \in \mathcal{S} \times \mathcal{A}} P(s'|s, a) \pi(a|s) \sum_{i \in \mathcal{A}} \lambda(s, i)$$

$$= \sum_{a \in \mathcal{A}} \lambda(s', a) - \gamma \sum_{s \in \mathcal{S}} P^\pi(s'|s) \sum_{a \in \mathcal{A}} \lambda(s, a)$$

$$= \lambda(s') - \gamma \sum_{s \in \mathcal{S}} P^\pi(s'|s) \lambda(s)$$

$$= \rho(s'),$$

and this proves the second statement.

To prove the third statement, we can first consider the vectorized form of Equation (6)

$$\lambda - \gamma (P^\pi)^T \lambda = \rho,$$

where

$$\lambda = \begin{bmatrix} \lambda(1) \\ \vdots \\ \lambda(|\mathcal{S}|) \end{bmatrix} \in \mathbb{R}^{|\mathcal{S}|}, \rho = \begin{bmatrix} \rho(1) \\ \vdots \\ \rho(|S|) \end{bmatrix} \in \mathbb{R}^{|\mathcal{S}|}, P^\pi = \begin{bmatrix} P^\pi(1|1) & \cdots & P^\pi(|\mathcal{S}||1) \\ \vdots & \ddots & \vdots \\ P^\pi(1||\mathcal{S}|) & \cdots & P^\pi(|\mathcal{S}|||\mathcal{S}|) \end{bmatrix} \in \mathbb{R}^{|\mathcal{S}| \times |\mathcal{S}|},$$

which leads to

$$\lambda^T = \rho^T + \gamma \rho^T P_\pi + \gamma^2 \rho^T P_\pi^2 + \cdots,$$

which can be written by

$$\lambda(s) = \sum_{k=0}^{\infty} \gamma^k \mathbb{P}[s_k = s|\pi, \rho] = \frac{1}{1 - \gamma} \mu^\pi(s), \quad s \in \mathcal{S}.$$

This completes the proof of the third statement.

The last statement can be proved through the following identities:

$$
\begin{aligned}
\sum_{(s,a,a')\in\mathcal{S}\times\mathcal{A}\times\mathcal{A}} \lambda(s,a,a')R(s,a) &= \sum_{(s,a)\in\mathcal{S}\times\mathcal{A}} \lambda(s,a)R(s,a) \\
&= \sum_{(s,a)\in\mathcal{S}\times\mathcal{A}} \pi(a|s)R(s,a)\lambda(s) \\
&= \sum_{s\in\mathcal{S}} R^\pi(s)\lambda(s) \\
&= \sum_{s\in\mathcal{S}} V^\pi(s)\rho(s) \\
&= J^\pi,
\end{aligned}
$$

where $R^\pi$ denotes the expected reward under the policy $\pi$, and $V^\pi$ denotes the value function under $\pi$. $\qquad\square$

## D  APPENDIX: COROLLARY 1

Based on Theorem 3, we can readily obtain the following corollary.

**Corollary 1.** *Suppose that $Q^*$ is the primal optimal solution, and $\lambda^*$ is the dual optimal solution.*

*Then, we obtain the following results:*

1. *$\beta^*$ defined in the following is an optimal policy:*
$$
\beta^*(s) = \arg\max_{a\in\mathcal{A}} Q^*(s,a).
$$

2. *Suppose that the marginal $\rho(s) = \sum_{a\in\mathcal{A}} \rho(s,a)$ represents the initial state distribution and $\varphi^*(a|s) := \frac{\rho(s,a)}{\sum_{a\in\mathcal{A}} \rho(s,a)}$ represents an optimal policy, i.e., $\beta^*$. Then, $\pi^*$ defined in the following is an optimal policy:*
$$
\pi^*(\cdot|s) := \begin{bmatrix} \frac{\lambda^*(s,1)}{\sum_{a\in\mathcal{A}}\lambda^*(s,a)} & \frac{\lambda^*(s,2)}{\sum_{a\in\mathcal{A}}\lambda^*(s,a)} & \cdots & \frac{\lambda^*(s,|\mathcal{A}|)}{\sum_{a\in\mathcal{A}}\lambda^*(s,a)} \end{bmatrix}
$$

*Proof.* The first statement can be directly obtained from Lemma 3 (the primal optimal solution is the optimal Q-function $Q^*$). For the second statement, we first note that the optimal primal solution $Q^*$ does not depend on $\rho$, while the optimal dual solution $\lambda^*$ depends on $\rho$. Suppose that the marginal

$$
\rho(s) = \sum_{a\in A} \rho(s,a)
$$

represents the initial state distribution and

$$
\varphi^*(a|s) := \frac{\rho(s,a)}{\sum_{a\in\mathcal{A}} \rho(s,a)}
$$

represents an optimal policy. using Theorem 3 and strong duality, one can derive

$$
\begin{aligned}
J^{\varphi^*} &:= \sum_{(s,a)\in\mathcal{S}\times\mathcal{A}} \rho(s)\varphi^*(a|s)Q^*(s,a) \\
&= \sum_{(s,a)\in\mathcal{S}\times\mathcal{A}} \rho(s,a)Q^*(s,a) \\
&= \sum_{(s,a,a')\in\mathcal{S}\times\mathcal{A}\times\mathcal{A}} \lambda^*(s,a,a')R(s,a) \\
&= \sum_{(s,a)\in\mathcal{S}\times\mathcal{A}} \lambda^*(s,a)R(s,a)
\end{aligned}
$$

$$= \sum_{(s,a)\in\mathcal{S}\times\mathcal{A}} \pi^*(a|s)R(s,a)\lambda^*(s)$$

$$= \sum_{s\in\mathcal{S}} R^{\pi^*}(s)\lambda^*(s)$$

$$= \sum_{s\in\mathcal{S}} V^{\pi^*}(s)\rho(s)$$

$$= J^{\pi^*}$$

This implies that $\pi^*$ is an optimal policy as well.

$\square$

## E   APPENDIX: LEMMA 4

In this paper, using the log-barrier function, the inequality constraints of the LP form of MDPs is integrated into a single objective function as follows:

$$f_\eta(Q) := \sum_{(s,a)\in\mathcal{S}\times\mathcal{A}} Q(s,a)\rho(s,a) + \eta \sum_{(s,a,a')\in\mathcal{S}\times\mathcal{A}\times\mathcal{A}} w(s,a,a')\varphi\left((FQ)(s,a,a') - Q(s,a)\right),$$

where $\eta > 0$ is the barrier parameter (weight) and $w(s,a,a') > 0, (s,a,a') \in \mathcal{S}\times\mathcal{A}\times\mathcal{A}$ are weight parameters of the inequality constraints. Several useful properties of the domain $\mathcal{D}$ of $f_\eta$

$$\mathcal{D} := \left\{ Q \in \mathbb{R}^{|\mathcal{S}||\mathcal{A}|} : (FQ)(s,a,a') - Q(s,a) < 0, (s,a,s') \in \mathcal{S}\times\mathcal{A}\times\mathcal{A} \right\},$$

are summarized in the following lemma.

**Lemma 4.** *The following statements hold true:*

1. *$\mathcal{D}$ is convex and open*

2. *$\mathcal{D}$ is bounded below and unbounded above.*

*Proof.* For the first statement, let us note that $\mathcal{D}$ is the set of points which satisfy the linear inequalities

$$(FQ)(s,a,a') - Q(s,a) < 0, \quad \forall(s,a,a') \in \mathcal{S}\times\mathcal{A}\times\mathcal{A},$$

which means that it is an intersections of affine sets, and hence, it is a convex set. Moreover, we can prove that it is open because the inequalities are strict.

For the second statement, one can observe that every $Q \in \mathcal{D}$ satisfies

$$(FQ)(s,a,a') - Q(s,a) < 0, \quad \forall(s,a,a') \in \mathcal{S}\times\mathcal{A}\times\mathcal{A},$$

which implies

$$(TQ)(s,a) - Q(s,a) < 0, \quad \forall(s,a) \in \mathcal{S}\times\mathcal{A}$$

and vice versa. Next, because the Bellman operator is monotone (Bertsekas, 2012, Lemma 1.1.1), i.e., for $Q' \geq Q$, $TQ' \geq TQ$, we have $T^kQ \leq Q$. Taking the limit $\lim_{k\to\infty} T^kQ = Q^* \leq Q$ leads to the conclusion that $\mathcal{D}$ is bounded below. Moreover, let

$$Q(s,a) = \frac{r_{\max} + \varepsilon}{1 - \gamma} > 0, \quad (s,a) \in \mathcal{S}\times\mathcal{A}$$

with any $\varepsilon > 0$. Then, one has

$$R(s,a) + \gamma \sum_{s'\in S} P(s'|s,a)Q(s',a') - Q(s,a)$$

$$= R(s,a) + \gamma \sum_{s'\in S} P(s'|s,a)\frac{r_{\max} + \varepsilon}{1 - \gamma} - \frac{r_{\max} + \varepsilon}{1 - \gamma}$$

$$=R(s,a) + (\gamma - 1)\frac{r_{\max} + \varepsilon}{1 - \gamma}$$
$$=R(s,a) - r_{\max} - \varepsilon < 0$$

Therefore, $Q$ is strictly feasible. Because $\varepsilon > 0$ is arbitrary, $\mathcal{D}$ is unbounded above, and we can conclude the proof. $\qquad\square$

The properties of the domain $\mathcal{D}$ established in the preceding lemma provide not only conceptual insight but also play a crucial role in the proofs of several subsequent results.

## F APPENDIX: LEMMA 5

Several useful properties of the level set $\mathcal{L}_c$ defined as

$$\mathcal{L}_c := \{Q \in \mathcal{D} : f_\eta(Q) \leq c\}$$

are summarized in the following lemma.

**Lemma 5.** *For any given $c > 0$, the following statements hold true:*

1. *$\mathcal{L}_c$ is convex and closed.*

2. *$\mathcal{L}_c$ is bounded.*

3. *there exists a constant $m_c > 0$ such that*

$$Q(s,a) - (FQ)(s,a,a') \geq m_c, \quad \forall (s,a,a') \in \mathcal{S} \times \mathcal{A} \times \mathcal{A}, \quad \forall Q \in \mathcal{L}_c$$

*Proof.* First of all, it is known that the log-barrier function is convex (Boyd and Vandenberghe, 2004, Chapter 11.2.1), and the composition of a convex function and affine function is convex (Boyd and Vandenberghe, 2004, Chapter 3.2.2). Since a sublevel set of a convex function is convex (Boyd and Vandenberghe, 2004, Chapter 3.1.6), $\mathcal{L}_c$ is convex.

Moreover, let us consider any sequence $Q_k \in \mathcal{L}_c$ such that $\lim_{x \to \infty} Q_k = \bar{Q} \in \mathcal{D}$. We want to prove that $\bar{Q} \in \mathcal{L}_c$ so that $\mathcal{L}_c$ is closed. This is true because $f_\eta$ is continuous in its domain (because it is convex in the domain), i.e.,

$$\lim_{x \to \infty} f_\eta(Q_k) = f_\eta(\lim_{x \to \infty} Q_k) = f_\eta(\bar{Q}) \leq c.$$

This completes the proof of the first statement.

For the second statement, we note that by Lemma 4, every $Q \in \mathcal{D}$ is bounded below. Therefore, $\mathcal{L}_c \subset \mathcal{D}$ is bounded below as well. To prove the upper bound, let us assume that $\mathcal{L}_c$ is unbounded from above. In this case, we can find a sequence $(Q_k)_{k=0}^\infty$ with $Q_k \in \mathcal{L}_c$ such that

$$\lim_{k \to \infty} \max_{(s,a) \in \mathcal{S} \times \mathcal{A}} Q_k(s,a) = \infty$$

To proceed, let us define

$$(s_k, a_k) := \arg\max_{(s,a) \in \mathcal{S} \times \mathcal{A}} Q_k(s,a).$$

Then, we have

$$0 < Q_k(s,a) - (FQ_k)(s,a,a')$$
$$\leq Q_k(s_k,a_k) + \max_{(s,a) \in \mathcal{S} \times \mathcal{A}} |R(s,a)| + \gamma \sum_{s' \in \mathcal{S}} P(s'|s,a) Q_k(s_k,a_k)$$
$$\leq (1 + \gamma) Q_k(s_k,a_k) + r_{\max}, \quad \forall (s,a,a') \in \mathcal{S} \times \mathcal{A} \times \mathcal{A}.$$

Because $-\ln$ is a nonincreasing function, it follows that

$$-\ln\{Q_k(s,a) - (FQ_k)(s,a,a')\} \geq -\ln\{(1+\gamma)Q_k(s_k,a_k) + r_{\max}\}, \quad \forall (s,a,a') \in \mathcal{S} \times \mathcal{A} \times \mathcal{A}.$$

Using this inequality, the objective function is bounded as

$$
\begin{aligned}
f_\eta(Q_k) &= \sum_{(s,a)\in\mathcal{S}\times\mathcal{A}} Q_k(s,a)\rho(s,a) - \eta \sum_{(s,a,a')\in\mathcal{S}\times\mathcal{A}\times\mathcal{A}} w(s,a,a')\ln\left[(FQ_k)(s,a,a') - Q_k(s,a)\right] \\
&\geq \sum_{(s,a)\in\mathcal{S}\times\mathcal{A}} Q_k(s,a)\rho(s,a) - \eta \sum_{(s,a,a')\in\mathcal{S}\times\mathcal{A}\times\mathcal{A}} w(s,a,a')\ln\left[(1+\gamma)Q_k(s_k,a_k) + r_{\max}\right] \\
&\geq Q_k(s_k,a_k)\rho(s_k,a_k) + \sum_{(s,a)\in(\mathcal{S}\times\mathcal{A})\backslash(s_k,a_k)} Q^*(s,a)\rho(s,a) \\
&\quad - \eta \sum_{(s,a,a')\in\mathcal{S}\times\mathcal{A}\times\mathcal{A}} w(s,a,a')\ln\left[(1+\gamma)Q_k(s_k,a_k) + r_{\max}\right] \\
&\geq Q_k(s_k,a_k)\min_{(s,a)\in\mathcal{S}\times\mathcal{A}}\rho(s,a) \\
&\quad + (|\mathcal{S}\times\mathcal{A}| - 1)\min_{(s,a)\in\mathcal{S}\times\mathcal{A}}Q^*(s,a)\rho(s,a) \\
&\quad - \eta \sum_{(s,a,a')\in\mathcal{S}\times\mathcal{A}\times\mathcal{A}} w(s,a,a')\ln\left[(1+\gamma)Q_k(s_k,a_k) + r_{\max}\right].
\end{aligned}
$$

By taking the limit, we have

$$
\begin{aligned}
\lim_{k\to\infty} f_\eta(Q_k) &\geq \lim_{k\to\infty} Q_k(s_k,a_k)\min_{(s,a)\in\mathcal{S}\times\mathcal{A}}\rho(s,a) \\
&\quad + (|\mathcal{S}\times\mathcal{A}| - 1)\min_{(s,a)\in\mathcal{S}\times\mathcal{A}}Q^*(s,a)\rho(s,a) \\
&\quad - \eta \sum_{(s,a,a')\in\mathcal{S}\times\mathcal{A}\times\mathcal{A}} w(s,a,a')\lim_{k\to\infty}\ln\left[(1+\gamma)Q_k(s_k,a_k) + r_{\max}\right] \\
&= \infty
\end{aligned}
$$

which is a contradiction. Therefore, $\mathcal{L}_c$ is bounded.

Next, we prove the last statement. Since $\mathcal{L}_c$ is bounded, for any $Q \in \mathcal{L}_c$, the first term in $f_\eta$ is bounded as follows:

$$
\left| \sum_{(s,a)\in\mathcal{S}\times\mathcal{A}} Q(s,a)\rho(s,a) \right| \leq B_1, \quad \forall Q \in \mathcal{L}_c,
$$

where $B_1 > 0$ is some constant. By contradiction, let us assume that

$$
\inf_{Q\in\mathcal{L}_c}\min_{(s,a,a')\in\mathcal{S}\times\mathcal{A}\times\mathcal{A}}\{Q(s,a) - (FQ)(s,a,a')\} = 0
$$

Then, there exists a sequence $Q_k \in \mathcal{L}_c$ such that

$$
\lim_{k\to\infty}\min_{(s,a,a')\in\mathcal{S}\times\mathcal{A}\times\mathcal{A}}\{Q_k(s,a) - (FQ_k)(s,a,a')\} = 0.
$$

Therefore, one gets

$$
\begin{aligned}
f_\eta(Q_k) &= \sum_{(s,a)\in\mathcal{S}\times\mathcal{A}} Q_k(s,a)\rho(s,a) - \eta \sum_{(s,a,a')\in\mathcal{S}\times\mathcal{A}\times\mathcal{A}} w(s,a,a')\ln\{Q_k(s,a) - (FQ_k)(s,a,a')\} \\
&\geq -B_1 - \eta \sum_{(s,a,a')\in\mathcal{S}\times\mathcal{A}\times\mathcal{A}} w(s,a,a')\ln\{Q_k(s,a) - (FQ_k)(s,a,a')\}.
\end{aligned}
$$

By taking the limit

$$
\begin{aligned}
c &\geq \lim_{k\to\infty} f_\eta(Q_k) \\
&\geq -B_1 - \eta \max_{(s,a,a')\in\mathcal{S}\times\mathcal{A}\times\mathcal{A}} w(s,a,a')|\mathcal{S}||\mathcal{A}|^2 \lim_{k\to\infty}\ln\left[\min_{(s,a,a')\in\mathcal{S}\times\mathcal{A}\times\mathcal{A}}\{Q_k(s,a) - (FQ_k)(s,a,a')\}\right] \\
&= \infty.
\end{aligned}
$$

which is a contradiction. Therefore, there should exist $m_c > 0$ such that

$$
\inf_{Q\in\mathcal{L}_c}\min_{(s,a,a')\in\mathcal{S}\times\mathcal{A}\times\mathcal{A}}\{Q(s,a) - (FQ)(s,a,a')\} \geq m_c.
$$

This completes the overall proof. $\qquad\square$

## G   APPENDIX: LEMMA 6

The following summarizes several properties of the objective function $f_\eta$ related to its convexity and smoothness.

**Lemma 6.** *For any given $c > 0$, we have*

1. $\nabla_Q f_\eta(Q)$ is bounded in $\mathcal{L}_c$

2. $f_\eta$ is $\mu$-strongly convex and $L$-smooth in $\mathcal{L}_c$

3. Moreover, $f_\eta$ is strictly convex in the domain $\mathcal{D}$.

*Proof of the first statement.* By Lemma 5, there exists a constant $m_c > 0$ such that

$$Q(s,a) - (FQ)(s,a,a') \geq m_c, \quad \forall (s,a,a') \in \mathcal{S} \times \mathcal{A} \times \mathcal{A}$$

for any $Q \in \mathcal{L}_c$. Therefore, we can derive the following bounds

$$|(\nabla_Q f(Q))(s,a)| \leq 1 + \gamma \sum_{(s',a') \in \mathcal{S} \times \mathcal{A}} P(s|s',a')|\lambda_\eta(s',a',a)| + \sum_{a' \in \mathcal{A}} |\lambda_\eta(s,a,a')|$$

$$\leq 1 + \gamma \sum_{(s',a') \in \mathcal{S} \times \mathcal{A}} P(s|s',a')|\lambda_\eta(s',a',a)| + \sum_{a' \in \mathcal{A}} |\lambda_\eta(s,a,a')|$$

$$\leq 1 + \gamma \sum_{(s',a') \in \mathcal{S} \times \mathcal{A}} P(s|s',a') \frac{\eta w(s',a',a)}{Q(s',a') - (FQ)(s',a',a)}$$

$$+ \sum_{a' \in \mathcal{A}} \frac{\eta w(s,a,a')}{Q(s,a) - (FQ)(s,a,a')}$$

$$\leq 1 + \gamma \max_{(s,a,a') \in \mathcal{S} \times \mathcal{A} \times \mathcal{A}} w(s,a,a') \frac{\eta|\mathcal{S}||\mathcal{A}|}{m_c} + \max_{(s,a,a') \in \mathcal{S} \times \mathcal{A} \times \mathcal{A}} w(s,a,a') \frac{\eta|\mathcal{A}|}{m_c}$$

which completes the proof of the first statement. $\square$

*Proof of the second statement.* For the second statement, since the log-barrier function is strongly convex in a convex compact set, $f_\eta$ is also strongly convex in the set because it is a composition of a linear function and strongly convex function. To be more precise, for all $(s,a,a') \in \mathcal{S} \times \mathcal{A} \times \mathcal{A}$, let us define

$$v_{s,a,a'} := \nabla_Q(Q(s,a) - (FQ)(s,a,a')) = e_{s,a} - \gamma \sum_{s' \in \mathcal{S}} P(s'|s,a) e_{s',a'}$$

where $e_{s,a} = e_s \otimes e_a \in \mathbb{R}^{|\mathcal{S}||\mathcal{A}|}$, and $e_s \in \mathbb{R}^{|\mathcal{S}|}$ and $e_a \in \mathbb{R}^{|\mathcal{A}|}$ are $s$th basis vector (all components are $0$ except for the $s$th component which is 1) and $a$th basis vector, respectively. Then, the gradient can be written as

$$\nabla_Q f_\eta(Q) = \rho - \eta \sum_{(s,a,a') \in \mathcal{S} \times \mathcal{A} \times \mathcal{A}} \frac{w(s,a,a')}{Q(s,a) - (FQ)(s,a,a')} v_{s,a,a'} \in \mathbb{R}^{|\mathcal{S}||\mathcal{A}|}$$

and the Hessian is

$$\nabla_Q^2 f_\eta(Q) = \eta \sum_{(s,a,a') \in \mathcal{S} \times \mathcal{A} \times \mathcal{A}} \frac{w(s,a,a')}{\{Q(s,a) - (FQ)(s,a,a')\}^2} v_{s,a,a'} v_{s,a,a'}^T \in \mathbb{R}^{|\mathcal{S}||\mathcal{A}| \times |\mathcal{S}||\mathcal{A}|}$$

By Lemma 5, we have

$$m_c \leq Q(s,a) - (FQ)(s,a,a') \leq M_c, \quad \forall Q \in \mathcal{L}_c, (s,a,a') \in \mathcal{S} \times \mathcal{A} \times \mathcal{A}$$

for some $m_c, M_c > 0$, where the bounds come from the fact that $\mathcal{L}_c$ is bounded in Lemma 5. Then, one has

$$x^T \nabla_Q^2 f_\eta(Q) x = \eta \sum_{(s,a,a') \in \mathcal{S} \times \mathcal{A} \times \mathcal{A}} \frac{w(s,a,a')(v_{s,a,a'}^T x)^2}{\{Q(s,a) - (FQ)(s,a,a')\}^2}$$

$$\geq \frac{\eta \min\limits_{(s,a,a')\in\mathcal{S}\times\mathcal{A}\times\mathcal{A}} w(s,a,a')}{M_c^2} \sum_{(s,a,a')\in\mathcal{S}\times\mathcal{A}\times\mathcal{A}} \left(v_{s,a,a'}^T x\right)^2$$

$$= \frac{\eta \min\limits_{(s,a,a')\in\mathcal{S}\times\mathcal{A}\times\mathcal{A}} w(s,a,a')}{M_c^2} x^T S x,$$

where

$$S := \sum_{(s,a,a')\in\mathcal{S}\times\mathcal{A}\times\mathcal{A}} v_{s,a,a'} v_{s,a,a'}^T \succeq 0.$$

Next, we prove that the above $S$ is strictly positive definite. By contradiction, let us suppose that $S$ is not strictly positive definite. Then, there exists a nonzero $x$ such that

$$x^T \sum_{(s,a,a')\in\mathcal{S}\times\mathcal{A}\times\mathcal{A}} v_{s,a,a'} v_{s,a,a'}^T x = \sum_{(s,a,a')\in\mathcal{S}\times\mathcal{A}\times\mathcal{A}} \left(v_{s,a,a'}^T x\right)^2 = 0.$$

This implies

$$v_{s,a,a'}^T x = 0, \quad \forall (s,a,a') \in \mathcal{S} \times \mathcal{A} \times \mathcal{A},$$

which can be equivalently written as

$$e_{s,a}^T x - \gamma \sum_{s'\in\mathcal{S}} P(s'|s,a) e_{s',a'}^T x = x(s,a) - \gamma \sum_{s'\in\mathcal{S}} P(s'|s,a) x(s',a') = 0, \quad \forall (s,a,a') \in \mathcal{S}\times\mathcal{A}\times\mathcal{A}.$$

Now, let us fix $a = a' \in \mathcal{A}$ and

$$(I - \gamma P_a)^T x_a = 0$$

where

$$x_a = \begin{bmatrix} x(1,a) \\ \vdots \\ x(|S|,a) \end{bmatrix} \in \mathbb{R}^{|\mathcal{S}|}, \quad P_a = \begin{bmatrix} P(1|1,a) & \cdots & P(|\mathcal{S}||1,a) \\ \vdots & \ddots & \vdots \\ P(1||S|,a) & \cdots & P(|\mathcal{S}|||S|,a) \end{bmatrix} \in \mathbb{R}^{|\mathcal{S}|\times|\mathcal{S}|}$$

Since $I - \gamma P_a$ is nonsingular for any $a \in \mathcal{A}$, $x_a = 0$ for all $a \in \mathcal{A}$. This is a contradiction. Therefore, $S \succ 0$ and

$$\nabla_Q^2 f_\eta(Q) \succeq \frac{\eta \min\limits_{(s,a,a')\in\mathcal{S}\times\mathcal{A}\times\mathcal{A}} w(s,a,a')\lambda_{\min}(S)}{M_c^2} I,$$

where $\lambda_{\min}(\cdot)$ denotes the minimum eigenvalue. This means that in $\mathcal{L}_c$, $f_\eta$ is $\mu$-strongly convex (Nesterov, 2018, Theorem 2.1.11) with

$$\mu = \frac{\eta \min\limits_{(s,a,a')\in\mathcal{S}\times\mathcal{A}\times\mathcal{A}} w(s,a,a')\lambda_{\min}(S)}{M_c^2}.$$

Similarly, we can easily show that in $\mathcal{L}_c$

$$x^T \nabla_Q^2 f_\eta(Q) x = \eta \sum_{(s,a,a')\in\mathcal{S}\times\mathcal{A}\times\mathcal{A}} \frac{w(s,a,a')(v_{s,a,a'}^T x)^2}{\{Q(s,a) - (FQ)(s,a,a')\}^2}$$

$$\leq \frac{\eta \max\limits_{(s,a,a')\in\mathcal{S}\times\mathcal{A}\times\mathcal{A}} w(s,a,a')}{m_c^2} \sum_{(s,a,a')\in\mathcal{S}\times\mathcal{A}\times\mathcal{A}} \left(v_{s,a,a'}^T x\right)^2$$

$$= \eta \frac{1}{m_c^2} x^T S x$$

$$\leq \frac{\eta \max\limits_{(s,a,a')\in\mathcal{S}\times\mathcal{A}\times\mathcal{A}} w(s,a,a')\lambda_{\max}(S)}{m_c^2} \|x\|_2^2,$$

where $\lambda_{\max}(\cdot)$ denotes the maximum eigenvalue. This implies

$$\|\nabla_Q f_\eta(Q) - \nabla_Q f_\eta(Q')\|_2 \leq L\|Q - Q'\|_2$$

with

$$L = \frac{\eta \max\limits_{(s,a,a')\in\mathcal{S}\times\mathcal{A}\times\mathcal{A}} w(s,a,a')\lambda_{\max}(S)}{m_c^2}.$$

by the sufficient condition in Nesterov (2018, Lemma 1.2.2). Therefore, $f_\eta$ is $L$-smooth in $\mathcal{L}_c$. $\quad\square$

*Proof of the third statement.* In the domain $Q \in \mathcal{D}$ which is unbounded above, by Lemma 5, we have

$$m_c \leq Q(s,a) - (FQ)(s,a,a'), \quad \forall Q \in \mathcal{D}, (s,a,a') \in \mathcal{S}\times\mathcal{A}\times\mathcal{A}$$

Therefore, the Hessian is bounded as

$$x^T\nabla_Q^2 f_\eta(Q)x \geq \frac{\eta \min\limits_{(s,a,a')\in\mathcal{S}\times\mathcal{A}\times\mathcal{A}} w(s,a,a')}{M(Q)^2}x^T Sx,$$

where

$$\max_{(s,a,a')\in\mathcal{S}\times\mathcal{A}\times\mathcal{A}}\{Q(s,a) - (FQ)(s,a,a')\} = M(Q) > 0, \quad \forall Q \in \mathcal{D}.$$

Since $S$ is strictly positive definite, the above result implies that the Hessian is strictly positive definite

$$\nabla_Q^2 f_\eta(Q) \succ 0, \quad \forall Q \in \mathcal{D}.$$

Therefore, by the second-order condition in Boyd and Vandenberghe (2004, Chapter 3.1.4), $f_\eta$ is strictly convex in $\mathcal{D}$. $\quad\square$

## H   APPENDIX: THEOREM 4

When gradient descent is applied to the objective function $f_\eta$, we obtain the following results on its convergence rate. An interesting observation is that, although $f_\eta$ is strictly convex within the domain $\mathcal{D}$ but not strongly convex, the convergence result coincides with that of a globally strongly convex function defined on the domain. The reason is that, while $f_\eta$ is only strictly convex over $\mathcal{D}$, it is both strongly convex and $L$-smooth within its sublevel sets $\mathcal{L}_c$. Exploiting this property, we can prove that the convergence rate for the gradient descent iterates of $f_\eta$ matches that of a strongly convex function on the domain. This constitutes one of the key results of this paper.

**Theorem 4.** *Let us consider the gradient descent iterates:*

$$Q_{k+1} = Q_k - \alpha\nabla_Q f_\eta(Q)|_{Q=Q_k},$$

*for $k = 0, 1, \ldots$ with a strictly feasible initial point $Q_0 \in \mathcal{D}$. With the step-size $0 < \alpha \leq \frac{2}{L_c+\mu_c}$, the iterates satisfies*

$$\left\|Q_k - \tilde{Q}_\eta\right\|_2^2 \leq \left(1 - \frac{2\alpha\mu_c L_c}{\mu_c + L_c}\right)^k \left\|Q_0 - \tilde{Q}_\eta\right\|_2^2$$

*where $\mu_c > 0$ is the coefficient of the strong convexity and $L_c > 0$ is the Lipschitz constant of the gradient of $f_\eta$ within the level set $\mathcal{L}_c$ where $c > f_\eta(Q_0)$,*

To prove the convergence, we first need the following basic lemma.

**Lemma 7** (Theorem 2.1.5 in Nesterov (2018) ). *If $f : \mathbb{R}^n \to \mathbb{R}$ is $L$-smooth, then*

$$f(y) \leq f(x) + \langle\nabla f(x), y - x\rangle + \frac{L}{2}\|y - x\|_2^2, \quad \forall x, y \in \mathbb{R}^n.$$

By inserting $y = x - \alpha \nabla f(x)$ in Lemma 7, we have

$$f(x - \alpha \nabla f(x)) \leq f(x) - \alpha \langle \nabla f(x), \nabla f(x) \rangle + \frac{L}{2} \|\alpha \nabla f(x)\|_2^2$$

$$= f(x) - \alpha \left( 1 - \frac{\alpha L}{2} \right) \|\nabla f(x)\|_2^2, \quad \forall x \in \mathbb{R}^n.$$

Based on this observation, we now prove Theorem 4 as follows.

*Proof of Theorem 4.* First of all, let us choose any $Q \in \mathcal{D}$ such that $\nabla_Q f_\eta(Q) \neq 0$. Then, $Q \in \mathcal{L}_c$ for some $c > 0$, which is any number such that $c > f_\eta(Q)$. Therefore, by Lemma 6, $f_\eta$ is $L_c$-smooth and $\mu_c$-strongly convex in $\mathcal{L}_c$ for some real numbers $L_c, \mu_c > 0$. We can choose a ball $B_r(Q)$ centered at $Q$ with radius $r > 0$ such that $B_r(Q) \subset \mathcal{L}_c$. Let us assume that

$$0 < \alpha \leq \frac{2}{L_c + \mu_c}$$

and define the next iterate

$$Q_\alpha^+ := Q - \alpha \nabla f_\eta(Q).$$

Because $B_r(Q) \subset \mathcal{L}_c$ we can choose a sufficiently small $\alpha > 0$ such that $Q_\alpha^+ \in \mathcal{L}_c$. Because $Q, Q_\alpha^+ \in \mathcal{L}_c$, we can apply Lemma 7 so that

$$f_\eta(Q_\alpha^+) = f_\eta(Q - \alpha \nabla_Q f_\eta(Q))$$

$$\leq f_\eta(Q) - \alpha \langle \nabla_Q f_\eta(Q), \nabla_Q f_\eta(Q) \rangle + \frac{L_c}{2} \|\alpha \nabla_Q f_\eta(Q)\|_2^2$$

$$= f_\eta(Q) - \alpha \left( 1 - \frac{\alpha L_c}{2} \right) \|\nabla_Q f_\eta(Q)\|_2^2$$

which implies $Q_\alpha^+ \in \mathcal{L}_c$. Now, we can increase $\alpha$ so that $\alpha = \frac{2}{L_c + \mu_c}$ or $Q_\alpha^+ \in \partial \mathcal{L}_c$, where $\partial \mathcal{L}_c$ is the boundary of $\mathcal{L}_c$ noting that $\mathcal{L}_c$ is a closed set (Lemma 5). Next, we want to prove that $\alpha$ reaches $\alpha = \frac{2}{L_c + \mu_c}$ until $Q_\alpha^+ \in \partial \mathcal{L}_c$. By contradiction, let us assume that $Q_\alpha^+ \in \partial \mathcal{L}_c$ until $\alpha$ reaches $\alpha = \frac{2}{\mathcal{L}_c + \mu_c}$. For any $\varepsilon > 0$, one can show that

$$Q_{\alpha+\varepsilon}^+ \notin \mathcal{L}_c$$

and

$$f_\eta(Q_{\alpha+\varepsilon}^+) > c$$

Howver, we know that

$$f_\eta(Q_\alpha^+) \leq f_\eta(Q) - \alpha \left( 1 - \frac{\alpha L_c}{2} \right) \|\nabla_Q f_\eta(Q)\|_2^2$$

$$\leq c - \alpha \left( 1 - \frac{\alpha L_c}{2} \right) \|\nabla_Q f_\eta(Q)\|_2^2$$

$$\leq f_\eta(Q_{\alpha+\varepsilon}^+) - \alpha \left( 1 - \frac{\alpha L_c}{2} \right) \|\nabla_Q f_\eta(Q)\|_2^2$$

Taking $\varepsilon \to 0$ leads to

$$f_\eta(Q_\alpha^+) < f_\eta(Q_\alpha^+) + \alpha \left( 1 - \frac{\alpha L_c}{2} \right) \|\nabla_Q f_\eta(Q)\|_2^2 \leq \lim_{\varepsilon \to 0} f_\eta(Q_{\alpha+\varepsilon}^+)$$

which contradicts with the fact that $f_\eta$ is continuous in $\mathcal{D}$. In summary, for any $0 < \alpha \leq \frac{2}{L_c + \mu_c}$ and $Q \in \mathcal{L}_c$, we have

$$f_\eta(Q_\alpha^+) \leq f_\eta(Q) - \alpha \left( 1 - \frac{\alpha L_c}{2} \right) \|\nabla_Q f_\eta(Q)\|_2^2$$

and $Q_\alpha^+ \in \mathcal{L}_c$. Therefore, if $Q_0 \in \mathcal{D}$, then

$$Q_k \in \mathcal{L}_c, \quad k = 0, 1, 2, \ldots$$

for some $c > 0$. Next, standard results in convex optimization (Theorem 2.1.15 in Nesterov (2018)) can be applied to conclude the proof. $\qquad \square$

**Theorem 5.** *Let us consider the gradient descent iterates:*

$$Q_{k+1} = Q_k - \alpha \, \nabla_Q f_\eta(Q)|_{Q=Q_k} \, ,$$

*for $k = 0, 1, \dots$ with a strictly feasible initial point $Q_0 \in \mathcal{D}$ and*

$$\|Q_0\|_\infty \leq \frac{\eta \sum\limits_{(s,a,a') \in \mathcal{S} \times \mathcal{A} \times \mathcal{A}} w(s,a,a')}{\min_{(s,a) \in \mathcal{S} \times \mathcal{A}} \rho(s,a)} + \frac{r_{\max}}{1 - \gamma}.$$

*Then, with the step-size $0 < \alpha \leq \frac{2}{L_c + \mu_c}$, the iterates satisfies*

$$\left\| Q_k - \tilde{Q}_\eta \right\|_\infty \leq \left( \sqrt{1 - \frac{2\alpha\mu_c L_c}{\mu_c + L_c}} \right)^k 2\sqrt{|\mathcal{S} \times \mathcal{A}|} \left( \frac{\eta \sum\limits_{(s,a,a') \in \mathcal{S} \times \mathcal{A} \times \mathcal{A}} w(s,a,a')}{\min_{(s,a) \in \mathcal{S} \times \mathcal{A}} \rho(s,a)} + \frac{r_{\max}}{1 - \gamma} \right),$$

*where $\mu_c > 0$ is the coefficient of the strong convexity and $L_c > 0$ is the Lipschitz constant of the gradient of $f_\eta$ within the level set $\mathcal{L}_c$ where $c > f_\eta(Q_0)$.*

*Moreover, to achieve $\left\| Q_k - \tilde{Q}_\eta \right\|_\infty \leq \varepsilon$, we need at least the following number of iterations:*

$$O \left( \ln \left( \frac{\sqrt{|\mathcal{S} \times \mathcal{A}|}}{\varepsilon} \left\{ \frac{\eta \sum\limits_{(s,a,a') \in \mathcal{S} \times \mathcal{A} \times \mathcal{A}} w(s,a,a')}{\min_{(s,a) \in \mathcal{S} \times \mathcal{A}} \rho(s,a)} + \frac{r_{\max}}{1 - \gamma} \right\} \right) \right).$$

*Proof.* By Theorem 4 and Lemma 9, we can obtain

$$
\begin{aligned}
\left\| Q_k - \tilde{Q}_\eta \right\|_2 &\leq \left( \sqrt{1 - \frac{2\alpha\mu_c L_c}{\mu_c + L_c}} \right)^k \left\| Q_0 - \tilde{Q}_\eta \right\|_2 \\
&\leq \left( \sqrt{1 - \frac{2\alpha\mu_c L_c}{\mu_c + L_c}} \right)^k \sqrt{|\mathcal{S} \times \mathcal{A}|} \left\| Q_0 - \tilde{Q}_\eta \right\|_\infty \\
&\leq \left( \sqrt{1 - \frac{2\alpha\mu_c L_c}{\mu_c + L_c}} \right)^k 2\sqrt{|\mathcal{S} \times \mathcal{A}|} \left( \frac{\eta \sum\limits_{(s,a,a') \in \mathcal{S} \times \mathcal{A} \times \mathcal{A}} w(s,a,a')}{\min_{(s,a) \in \mathcal{S} \times \mathcal{A}} \rho(s,a)} + \frac{r_{\max}}{1 - \gamma} \right),
\end{aligned}
$$

where the second inequality comes from the inequality $\| \cdot \|_2 \leq \sqrt{|\mathcal{S} \times \mathcal{A}|} \| \cdot \|_\infty$, the last inequality is due to Lemma 9 and the bounded assumption on $Q_0$. Rearranging terms and taking log function on both sides lead to

$$k \ln \left( \sqrt{1 - \frac{2\alpha\mu_c L_c}{\mu_c + L_c}} \right) \leq \ln \left( \varepsilon \frac{1}{2\sqrt{|\mathcal{S} \times \mathcal{A}|}} \frac{1}{\frac{\eta \sum\limits_{(s,a,a') \in \mathcal{S} \times \mathcal{A} \times \mathcal{A}} w(s,a,a')}{\min_{(s,a) \in \mathcal{S} \times \mathcal{A}} \rho(s,a)} + \frac{r_{\max}}{1 - \gamma}} \right).$$

Rearranging and simplifying terms again lead to

$$k \geq \frac{\ln \left( \frac{2\sqrt{|\mathcal{S} \times \mathcal{A}|}}{\varepsilon} \left\{ \frac{\eta \sum\limits_{(s,a,a') \in \mathcal{S} \times \mathcal{A} \times \mathcal{A}} w(s,a,a')}{\min_{(s,a) \in \mathcal{S} \times \mathcal{A}} \rho(s,a)} + \frac{r_{\max}}{1 - \gamma} \right\} \right)}{-\ln \left( \sqrt{1 - \frac{2\alpha\mu_c L_c}{\mu_c + L_c}} \right)}.$$

$\square$

We note that if $w$ is a probability distribution, then the sum in the numerator becomes one. On the other hand, the term $\min_{(s,a) \in \mathcal{S} \times \mathcal{A}} \rho(s,a)$ depends on the initial state distribution $\rho$, and this

term can be made arbitrarily small. For instance, when the initial state distribution is uniform, i.e., $\rho(s,a) = \frac{1}{|\mathcal{S}||\mathcal{A}|}$ and $w$ is a probability distribution, then

$$O\left(\ln\left(\frac{\eta|\mathcal{S}\times\mathcal{A}|^{3/2}}{\varepsilon} + \frac{|\mathcal{S}\times\mathcal{A}|^{1/2}}{\varepsilon}\frac{r_{\max}}{1-\gamma}\right)\right)$$

For computational cose of the gradient computation, in tabular log-barrier method, each gradient update scales with the number of state–action pairs (i.e., linear in $|\mathcal{S}\times\mathcal{A}|$) because the objective and its gradient must be evaluated over the relevant domain.

## I APPENDIX: GRADIENT DESCENT EXAMPLE

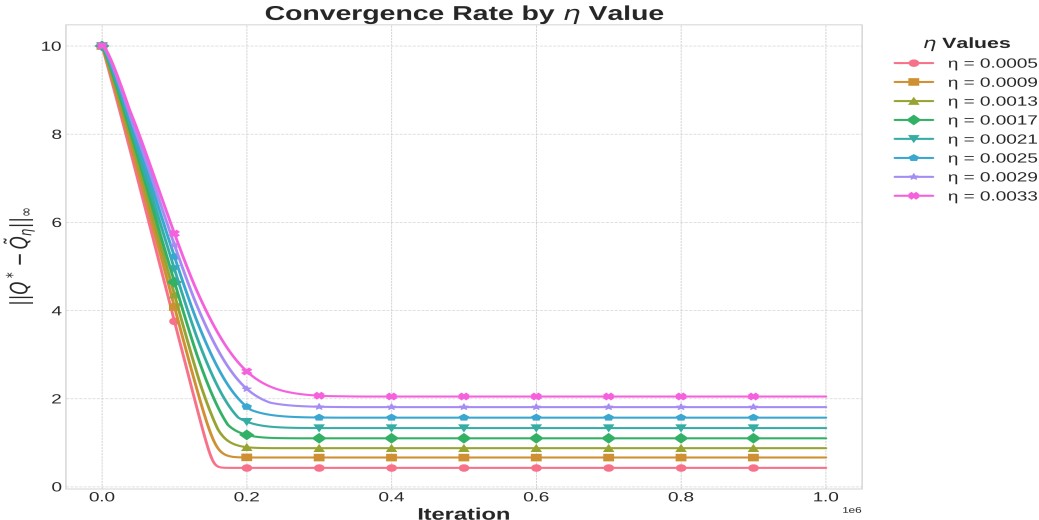

Figure 3: Comparison of the log-barrier coefficient $\eta$ on the convergence rate. The plot shows the max-norm error, $\|Q^* - \tilde{Q}_\eta\|_\infty$, between the learned and optimal Q-functions versus the number of training iterations. These results were obtained from experiments on a 6×6 FrozenLake-v1 environment, where the ground-truth $Q^*$ was pre-computed using value iteration.

We conducted an experiment with two primary objectives: first, to demonstrate that the gradient expression proposed in Lemma 2 can serve as a valid gradient within a gradient descent framework, and second, to analyze how the log-barrier coefficient $\eta$ affects the performance. For this purpose, we used the FrozenLake-v1 environment from Gymnasium with a custom 6×6 grid in a tabular setting. We represented the Q-function in a tabular form, for which the optimal Q-function, $Q^*$, can be pre-computed with guaranteed convergence using value iteration (Puterman, 2014). Then, we utilized the gradient descent method with the learning rate $\alpha = 0.01$ for all experiments and measured the max-norm error between $Q^*$ and $\tilde{Q}_\eta$. The overall convergence of the algorithm, as shown in Figure 3, validates our first objective by confirming that the expression from Lemma 2 serves as a functional gradient.

Our second objective is to analyze the impact of the coefficient $\eta$, which serves as an approximation factor in the log-barrier method. A higher value of $\eta$ corresponds to a relaxed approximation of the feasible set's boundary, leading to a smoother optimization landscape. Conversely, a lower value of $\eta$ yields a more precise approximation but creates a steeper barrier near the boundary. This theoretical property is clearly demonstrated in Figure 3, where the different error floors for each curve confirm the role of $\eta$ as an approximation factor.

Figure 4 and Figure 5 show the error evolutions for different learning rates while keeping the barrier weight $\eta$ fixed. Figure 4 corresponds to $\eta = 0.001$ and Figure 5 to $\eta = 0.005$. As the plots show, larger learning rates lead to faster convergence in these experiments.

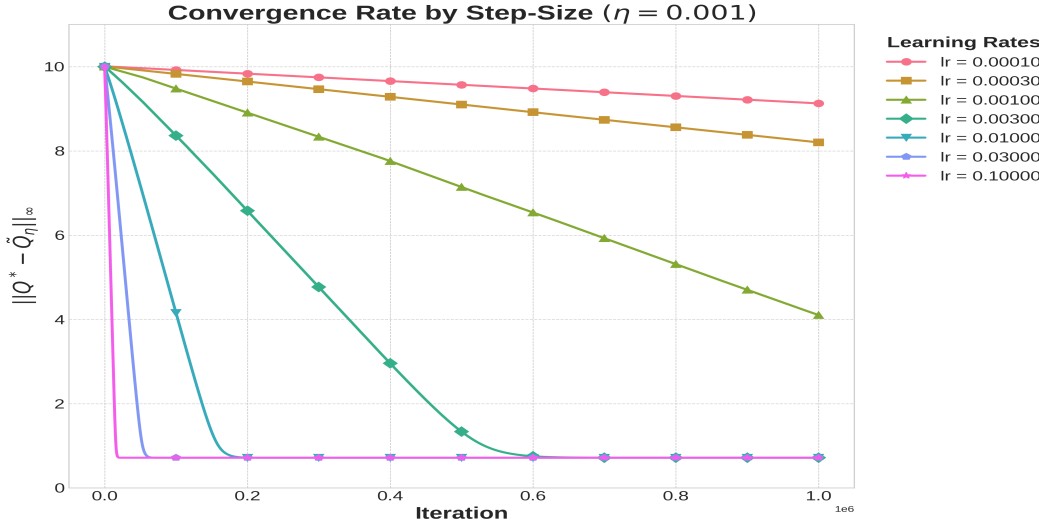

Figure 4: Comparison of the error evolutions with $\eta = 0.001$ for different learning rates. The plot shows the max-norm error, $\|Q^* - \tilde{Q}_\eta\|_\infty$, between the learned and optimal Q-functions versus the number of training iterations. These results were obtained from experiments on a 6×6 FrozenLake-v1 environment, where the ground-truth $Q^*$ was pre-computed using value iteration.

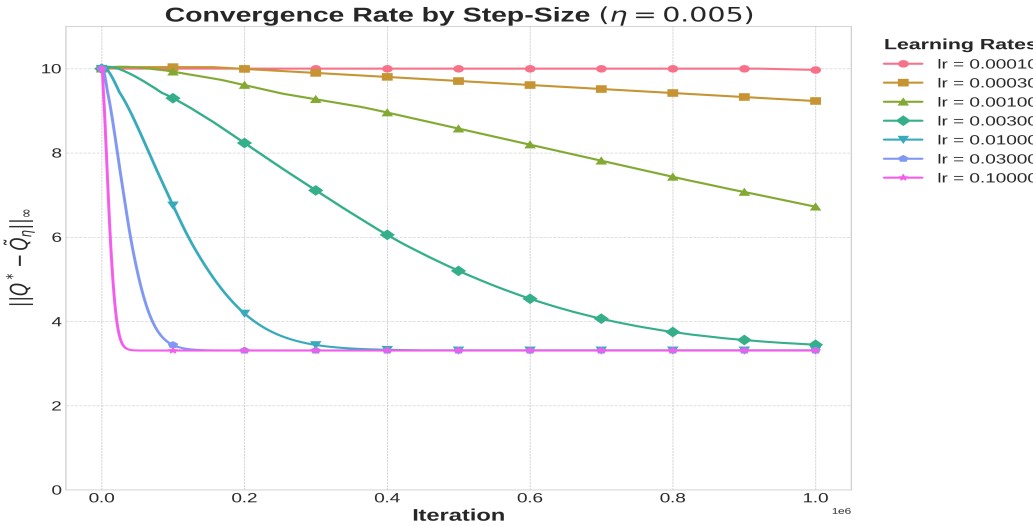

Figure 5: Comparison of the error evolutions with $\eta = 0.005$ for different learning rates. The plot shows the max-norm error, $\|Q^* - \tilde{Q}_\eta\|_\infty$, between the learned and optimal Q-functions versus the number of training iterations. These results were obtained from experiments on a 6×6 FrozenLake-v1 environment, where the ground-truth $Q^*$ was pre-computed using value iteration.

Figure 6 compares the convergence of classical dynamic programming (specifically, value iteration) with gradient descent on the log-barrier objective. For the log-barrier method we used a fixed learning rate and fixed $\eta$; the learning rate was tuned to produce the best possible performance for that method. As Figure 6 shows, value iteration converges faster than the gradient-descent log-barrier approach in this experiment.

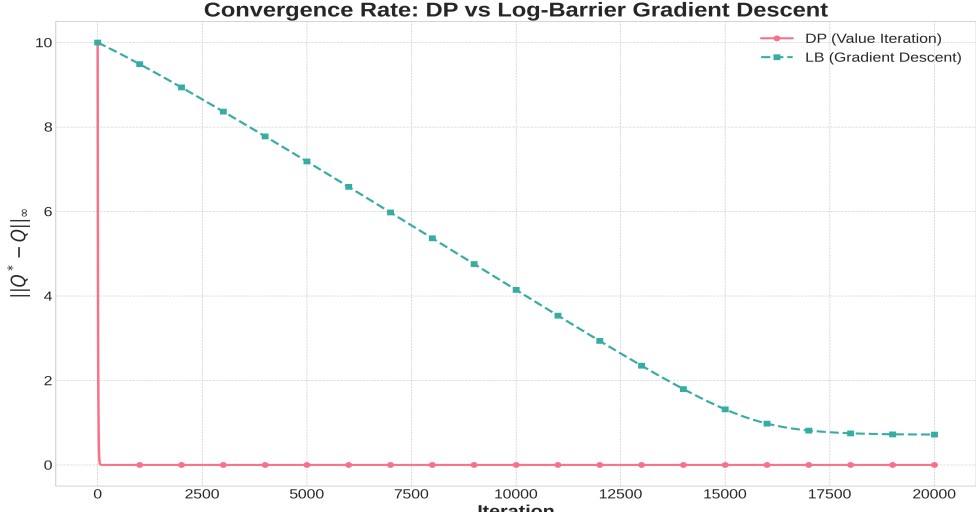

Figure 6: Comparison of the error evolutions of the classical dynamic programming and the gradient descent of the log-barrier objective with fixed $\eta$ and learning rate. The plot shows the max-norm error, $\|Q^* - \tilde{Q}_\eta\|_\infty$, between the learned and optimal Q-functions versus the number of training iterations. These results were obtained from experiments on a 6×6 FrozenLake-v1 environment, where the ground-truth $Q^*$ was pre-computed using value iteration.

Remark on the choice of $\eta$: Theoretically, as $\eta$ goes to zero, the log-barrier solution $\tilde{Q}_\eta$ approaches the true $Q^*$. This relationship is established in Theorem 1. In practice, however, taking $\eta$ too small can cause numerical issues. For example, with very small $\eta$ the gradient-based estimate $\tilde{Q}_\eta$ may step outside the domain that satisfies the inequality constraints due to finite-precision and optimization error. The $\eta$ values reported in Table 1 and Table 2 were obtained by empirical tuning; in our experiments, we typically used moderate values of $\eta$ (see Table 1 and Table 2 for the exact settings). At present, a principled criterion for selecting $\eta$ has not been established, and we view the development of such a selection rule as important future work.

## J    APPENDIX: PROOF OF THEOREM 1

For convenience of the reader, the statements of Theorem 1 are summarized as follows:

1. $\eta \min\limits_{(s,a,a')\in\mathcal{S}\times\mathcal{A}\times\mathcal{A}} w(s,a,a') < \left\|\tilde{Q}_\eta - Q^*\right\|_\infty \leq \dfrac{\eta \sum\limits_{(s,a,a')\in\mathcal{S}\times\mathcal{A}\times\mathcal{A}} w(s,a,a')}{\min_{(s,a)\in\mathcal{S}\times\mathcal{A}}\rho(s,a)}$

2. $\eta(1-\gamma) \min\limits_{(s,a,a')\in\mathcal{S}\times\mathcal{A}\times\mathcal{A}} w(s,a,a') < \left\|\tilde{Q}_\eta - T\tilde{Q}_\eta\right\|_\infty \leq \dfrac{(1+\gamma)\eta \sum\limits_{(s,a,a')\in\mathcal{S}\times\mathcal{A}\times\mathcal{A}} w(s,a,a')}{\min_{(s,a)\in\mathcal{S}\times\mathcal{A}}\rho(s,a)}$

To prove Theorem 1, we first need to prove the following lemma.

**Lemma 8.** *Let us define*

$$\tilde{\lambda}_\eta(s,a,a') := -\frac{\eta w(s,a,a')}{(F\tilde{Q}_\eta)(s,a,a') - \tilde{Q}_\eta(s,a)}, \quad (s,a,a') \in \mathcal{S} \times \mathcal{A} \times \mathcal{A}.$$

*Then, we have the following results:*

*1.* $\eta(1-\gamma) \min\limits_{(s,a,a')\in\mathcal{S}\times\mathcal{A}\times\mathcal{A}} w(s,a,a') < \left\|\tilde{Q}_\eta - T\tilde{Q}_\eta\right\|_\infty$

*2.* $\eta \min\limits_{(s,a,a')\in\mathcal{S}\times\mathcal{A}\times\mathcal{A}} w(s,a,a') < \tilde{Q}_\eta(s,a) - Q^*(s,a), \quad \forall(s,a) \in \mathcal{S} \times \mathcal{A}.$

3. $\eta \min_{(s,a,a') \in \mathcal{S} \times \mathcal{A} \times \mathcal{A}} w(s,a,a') < \left\| \tilde{Q}_\eta - Q^* \right\|_\infty$

*Proof.* By Theorem 3, we have

$$0 < \frac{\eta w(s,a,a')(1-\gamma)}{\tilde{Q}_\eta(s,a) - (F\tilde{Q}_\eta)(s,a,a')} < 1, \quad \forall (s,a,a') \in \mathcal{S} \times \mathcal{A} \times \mathcal{A},$$

which implies

$$\eta \min_{(s,a,a') \in \mathcal{S} \times \mathcal{A} \times \mathcal{A}} w(s,a,a')(1-\gamma) < \tilde{Q}_\eta(s,a) - (F\tilde{Q}_\eta)(s,a,a'), \quad \forall (s,a,a') \in \mathcal{S} \times \mathcal{A} \times \mathcal{A},$$

and hence,

$$\eta(1-\gamma) \min_{(s,a,a') \in \mathcal{S} \times \mathcal{A} \times \mathcal{A}} w(s,a,a') < \tilde{Q}_\eta(s,a) - (T\tilde{Q}_\eta)(s,a), \quad \forall (s,a) \in \mathcal{S} \times \mathcal{A}.$$

By taking the absolute value on the right-hand side and taking the maximum over $(s,a) \in \mathcal{S} \times \mathcal{A}$, we can completes the proof of the first statement. Next, for the second statement, we first note the following bounds:

$$\eta(1-\gamma) \min_{(s,a,a') \in \mathcal{S} \times \mathcal{A} \times \mathcal{A}} w(s,a,a')$$

$$< \tilde{Q}_\eta(s,a) - (T\tilde{Q}_\eta)(s,a)$$

$$= \tilde{Q}_\eta(s,a) - (T\tilde{Q}_\eta)(s,a) - Q^*(s,a) + (TQ^*)(s,a)$$

$$= \tilde{Q}_\eta(s,a) - Q^*(s,a) - \gamma \sum_{s' \in \mathcal{S}} P(s'|s,a) \{ \max_{a' \in \mathcal{A}} \tilde{Q}_\eta(s',a') - \max_{a' \in \mathcal{A}} Q^*(s',a') \}$$

$$\leq \tilde{Q}_\eta(s,a) - Q^*(s,a) - \gamma \sum_{s' \in \mathcal{S}} P(s'|s,a) \{ \tilde{Q}_\eta(s',b) - Q^*(s',b) \}$$

$$\leq \tilde{Q}_\eta(s,a) - Q^*(s,a) - \gamma \sum_{s' \in \mathcal{S}} P(s'|s,a) \min_{(s',a') \in \mathcal{S} \times \mathcal{A}} \{ \tilde{Q}_\eta(s',a') - Q^*(s',a') \}$$

$$\leq \tilde{Q}_\eta(s,a) - Q^*(s,a) - \gamma \min_{(s',a') \in \mathcal{S} \times \mathcal{A}} \{ \tilde{Q}_\eta(s',a') - Q^*(s',a') \}, \quad \forall (s,a) \in \mathcal{S} \times \mathcal{A},$$

where in the fourth line, $b = \arg\max_{a' \in \mathcal{A}} Q^*(s',a')$. Taking the minimum over $(s,a) \in \mathcal{S} \times \mathcal{A}$ on both sides yields

$$(1-\gamma)\eta \min_{(s,a,a') \in \mathcal{S} \times \mathcal{A} \times \mathcal{A}} w(s,a,a') < (1-\gamma) \min_{(s',a') \in \mathcal{S} \times \mathcal{A}} \{ \tilde{Q}_\eta(s',a') - Q^*(s',a') \},$$

which is the second statement. Moreover, it also implies

$$\eta \min_{(s,a,a') \in \mathcal{S} \times \mathcal{A} \times \mathcal{A}} w(s,a,a') < \min_{(s',a') \in \mathcal{S} \times \mathcal{A}} \{ \tilde{Q}_\eta(s',a') - Q^*(s',a') \}$$

$$\leq \min_{(s',a') \in \mathcal{S} \times \mathcal{A}} |\tilde{Q}_\eta(s',a') - Q^*(s',a')|$$

$$\leq \max_{(s',a') \in \mathcal{S} \times \mathcal{A}} |\tilde{Q}_\eta(s',a') - Q^*(s',a')|$$

$$= \left\| \tilde{Q}_\eta - Q^* \right\|_\infty,$$

which proves the last statement. $\square$

Based on Lemma 8, we will now prove Theorem 1.

*Proof of the first statement of Theorem 1.* Because the optimal solution $\tilde{Q}_\eta$ should be in the domain $\mathcal{D}$ of the log-barrier function, it should be strictly feasible for the primal LP. Moreover, $\tilde{\lambda}_\eta$ is feasible for the dual LP. Using these results, we can obtain the following lower bounds:

$$\sum_{(s,a) \in \mathcal{S} \times \mathcal{A}} \rho(s,a) Q^*(s,a)$$

$$= \sum_{(s,a,a') \in \mathcal{S} \times \mathcal{A} \times \mathcal{A}} \lambda^*(s,a,a') R(s,a)$$

$$\geq \sum_{(s,a,a') \in \mathcal{S} \times \mathcal{A} \times \mathcal{A}} \tilde{\lambda}_\eta(s,a,a') R(s,a)$$

$$= \sum_{(s,a,a') \in \mathcal{S} \times \mathcal{A} \times \mathcal{A}} \tilde{\lambda}_\eta(s,a,a') R(s,a)$$

$$+ \sum_{(s,a) \in \mathcal{S} \times \mathcal{A}} \tilde{Q}_\eta(s,a) \left\{ \rho(s,a) + \gamma \sum_{(s',a') \in \mathcal{S} \times \mathcal{A}} \tilde{\lambda}_\eta(s',a',a) P(s|s',a') - \sum_{a' \in \mathcal{A}} \tilde{\lambda}_\eta(s,a,a') \right\}$$

$$= L(\tilde{Q}_\eta, \tilde{\lambda}_\eta)$$

$$= \sum_{(s,a) \in \mathcal{S} \times \mathcal{A}} \rho(s,a) \tilde{Q}_\eta(s,a) + \sum_{(s,a,a') \in \mathcal{S} \times \mathcal{A} \times \mathcal{A}} \tilde{\lambda}_\eta(s,a,a') \{ (F\tilde{Q}_\eta)(s,a,a') - \tilde{Q}_\eta(s,a) \}$$

$$= \sum_{(s,a) \in \mathcal{S} \times \mathcal{A}} \rho(s,a) \tilde{Q}_\eta(s,a) + \sum_{(s,a,a') \in \mathcal{S} \times \mathcal{A} \times \mathcal{A}} -\eta \frac{w(s,a,a') \{ (F\tilde{Q}_\eta)(s,a,a') - \tilde{Q}_\eta(s,a) \}}{(F\tilde{Q}_\eta)(s,a,a') - \tilde{Q}_\eta(s,a)}$$

$$= \sum_{(s,a) \in \mathcal{S} \times \mathcal{A}} \rho(s,a) \tilde{Q}_\eta(s,a) - \eta \sum_{(s,a,a') \in \mathcal{S} \times \mathcal{A} \times \mathcal{A}} w(s,a,a'), \tag{8}$$

where the second line is due to the strong duality of the LP problem, the third and fourth lines are due to the fact that $\tilde{\lambda}_\eta$ is dual feasible, and the sixth line is obtained by rearranging terms. On the other hand, since $\tilde{Q}_\eta \in \mathcal{D}$ is primal feasible, we have

$$\sum_{(s,a) \in \mathcal{S} \times \mathcal{A}} \rho(s,a) Q^*(s,a) \leq \sum_{(s,a) \in \mathcal{S} \times \mathcal{A}} \rho(s,a) \tilde{Q}_\eta(s,a). \tag{9}$$

Combining Equation (8) and Equation (9) leads to

$$0 \leq \sum_{(s,a) \in \mathcal{S} \times \mathcal{A}} \rho(s,a) \{ \tilde{Q}_\eta(s,a) - Q^*(s,a) \} \leq \eta \sum_{(s,a,a') \in \mathcal{S} \times \mathcal{A} \times \mathcal{A}} w(s,a,a').$$

Using Lemma 8, we further derive

$$\sum_{(s,a) \in \mathcal{S} \times \mathcal{A}} \rho(s,a) \{ \tilde{Q}_\eta(s,a) - Q^*(s,a) \} \geq \min_{(s,a) \in \mathcal{S} \times \mathcal{A}} \rho(s,a) \sum_{(s,a) \in \mathcal{S} \times \mathcal{A}} (\tilde{Q}_\eta(s,a) - Q^*(s,a))$$

$$= \min_{(s,a) \in \mathcal{S} \times \mathcal{A}} \rho(s,a) \left\| \tilde{Q}_\eta - Q^* \right\|_1$$

$$\geq \min_{(s,a) \in \mathcal{S} \times \mathcal{A}} \rho(s,a) \left\| \tilde{Q}_\eta - Q^* \right\|_\infty,$$

where the last line comes from $\|\cdot\|_1 \geq \|\cdot\|_\infty$. By dividing both sides by $\min_{(s,a) \in \mathcal{S} \times \mathcal{A}} \rho(s,a) > 0$, one gets

$$\left\| \tilde{Q}_\eta - Q^* \right\|_\infty \leq \frac{\eta \sum_{(s,a,a') \in \mathcal{S} \times \mathcal{A} \times \mathcal{A}} w(s,a,a')}{\min_{(s,a) \in \mathcal{S} \times \mathcal{A}} \rho(s,a)}.$$

The lower bounds come from the second statement of Lemma 8. $\qquad \square$

*Proof of the second statement of Theorem 1.* Next, using the reverse triangle inequality, one gets

$$\left\| \tilde{Q}_\eta - Q^* \right\|_\infty = \left\| \tilde{Q}_\eta - T\tilde{Q}_\eta + T\tilde{Q}_\eta - Q^* \right\|_\infty$$

$$\geq \left\| \tilde{Q}_\eta - T\tilde{Q}_\eta \right\|_\infty - \left\| T\tilde{Q}_\eta - TQ^* \right\|_\infty$$

$$\geq \left\| \tilde{Q}_\eta - T\tilde{Q}_\eta \right\|_\infty - \gamma \left\| \tilde{Q}_\eta - Q^* \right\|_\infty,$$

where the last inequality is due to the contraction property of the Bellman operator with respect to $\infty$-norm.

Rearranging terms on both sides results in

$$\left\|\tilde{Q}_\eta - T\tilde{Q}_\eta\right\|_\infty \leq (1+\gamma)\left\|\tilde{Q}_\eta - Q^*\right\|_\infty \leq \frac{(1+\gamma)\eta \sum\limits_{(s,a,a')\in\mathcal{S}\times\mathcal{A}\times\mathcal{A}} w(s,a,a')}{\min_{(s,a)\in\mathcal{S}\times\mathcal{A}}\rho(s,a)}.$$

The lower bounds come from the first statement of Lemma 8, and this completes the proof. $\square$

Based on Theorem 1, we can derive the following bound on $\tilde{Q}_\eta$.

**Lemma 9** (Bound on $\tilde{Q}_\eta$). $\tilde{Q}_\eta$ *satisfies*

$$\left\|\tilde{Q}_\eta\right\|_\infty \leq \frac{\eta \sum\limits_{(s,a,a')\in\mathcal{S}\times\mathcal{A}\times\mathcal{A}} w(s,a,a')}{\min_{(s,a)\in\mathcal{S}\times\mathcal{A}}\rho(s,a)} + \frac{r_{\max}}{1-\gamma}.$$

*Proof.* Using Theorem 1, one has

$$\begin{aligned}
\left\|\tilde{Q}_\eta\right\|_\infty &= \left\|\tilde{Q}_\eta - Q^* + Q^*\right\|_\infty \\
&\leq \left\|\tilde{Q}_\eta - Q^*\right\|_\infty + \|Q^*\|_\infty \\
&\leq \frac{\eta \sum\limits_{(s,a,a')\in\mathcal{S}\times\mathcal{A}\times\mathcal{A}} w(s,a,a')}{\min_{(s,a)\in\mathcal{S}\times\mathcal{A}}\rho(s,a)} + \frac{r_{\max}}{1-\gamma},
\end{aligned}$$

where the last line is due to Theorem 1 and $\|Q^*\|_\infty \leq \frac{r_{\max}}{1-\gamma}$. This completes the proof.

$\square$

## K    APPENDIX: PROOF OF THEOREM 2

For convenience of the reader, the statements of Theorem 2 are summarized as follows:

1. $J^{\pi^*} - \eta \sum\limits_{(s,a,a')\in\mathcal{S}\times\mathcal{A}\times\mathcal{A}} w(s,a,a') \leq J^{\tilde{\pi}_\eta} \leq J^{\pi^*}$

2. $J^{\pi^*} - \frac{\eta(1+\gamma)\sum\limits_{(s,a,a')\in\mathcal{S}\times\mathcal{A}\times\mathcal{A}} w(s,a,a')}{(1-\gamma)\min_{(s,a)\in\mathcal{S}\times\mathcal{A}}\rho(s,a)} \leq J^{\tilde{\beta}_\eta} \leq J^{\pi^*}$

3. $-\frac{\eta(1+\gamma)\sum\limits_{(s,a,a')\in\mathcal{S}\times\mathcal{A}\times\mathcal{A}} w(s,a,a')}{(1-\gamma)\min_{(s,a)\in\mathcal{S}\times\mathcal{A}}\rho(s,a)} \leq J^{\tilde{\beta}_\eta} - J^{\tilde{\pi}_\eta} \leq \eta \sum\limits_{(s,a,a')\in\mathcal{S}\times\mathcal{A}\times\mathcal{A}} w(s,a,a')$

*Proof of the first statement.* Because the optimal solution $\tilde{Q}_\eta \in \mathcal{D}$ should be in the domain of the log-barrier function, it should be strictly feasible for the primal LP. Moreover, $\tilde{\lambda}_\eta$ is feasible for the dual LP. Using these results, we can derive the following inequalities:

$$\begin{aligned}
&\sum_{(s,a,a')\in\mathcal{S}\times\mathcal{A}\times\mathcal{A}} \lambda^*(s,a,a')R(s,a) \\
\geq\ &\sum_{(s,a,a')\in\mathcal{S}\times\mathcal{A}\times\mathcal{A}} \tilde{\lambda}(s,a,a')R(s,a) \\
=\ &\sum_{(s,a,a')\in\mathcal{S}\times\mathcal{A}\times\mathcal{A}} \tilde{\lambda}(s,a,a')R(s,a) \\
&+ \sum_{(s,a)\in\mathcal{S}\times\mathcal{A}} \tilde{Q}(s,a)\left\{\rho(s,a) + \gamma\sum_{(s',a')\in\mathcal{S}\times\mathcal{A}} \tilde{\lambda}(s',a',a)P(s|s',a') - \sum_{a'\in\mathcal{A}} \tilde{\lambda}(s,a,a')\right\}
\end{aligned}$$

$$=L(\tilde{Q}, \tilde{\lambda})$$

$$= \sum_{(s,a) \in \mathcal{S} \times \mathcal{A}} \rho(s,a) \tilde{Q}(s,a) - \eta \sum_{(s,a,a') \in \mathcal{S} \times \mathcal{A} \times \mathcal{A}} w(s,a,a')$$

$$\geq \sum_{(s,a) \in \mathcal{S} \times \mathcal{A}} \rho(s,a) Q^*(s,a) - \eta \sum_{(s,a,a') \in \mathcal{S} \times \mathcal{A} \times \mathcal{A}} w(s,a,a')$$

$$= \sum_{(s,a,a') \in \mathcal{S} \times \mathcal{A} \times \mathcal{A}} \lambda^*(s,a,a') R(s,a) - \eta \sum_{(s,a,a') \in \mathcal{S} \times \mathcal{A} \times \mathcal{A}} w(s,a,a'),$$

where the first inequality is due to the fact that $\tilde{\lambda}_\eta$ is dual feasible, the second inequality is due to the fact that $\tilde{Q}_\eta$ is primal feasible, and the last line is due to the strong duality of the LP problem. Summarizing the above results, one gets

$$\sum_{(s,a,a') \in \mathcal{S} \times \mathcal{A} \times \mathcal{A}} \lambda^*(s,a,a') R(s,a) \geq \sum_{(s,a,a') \in \mathcal{S} \times \mathcal{A} \times \mathcal{A}} \tilde{\lambda}(s,a,a') R(s,a)$$

$$\geq \sum_{(s,a,a') \in \mathcal{S} \times \mathcal{A} \times \mathcal{A}} \lambda^*(s,a,a') R(s,a)$$

$$- \eta \sum_{(s,a,a') \in \mathcal{S} \times \mathcal{A} \times \mathcal{A}} w(s,a,a').$$

The proof of the first statement is then completed using Theorem 3.

$\square$

*Proof of the second statement.* For the proof of the second statement, from Lemma 8 and Theorem 1, we have

$$\eta(1-\gamma) \min_{(s,a,a') \in \mathcal{S} \times \mathcal{A} \times \mathcal{A}} w(s,a,a')$$

$$\leq \tilde{Q}_\eta(s,a) - \left\{ R(s,a) + \gamma \max_{a' \in \mathcal{A}} \sum_{s' \in \mathcal{S}} P(s'|s,a) \tilde{Q}_\eta(s',a') \right\}$$

$$\leq \frac{\eta(1+\gamma) \sum_{(s,a,a') \in \mathcal{S} \times \mathcal{A} \times \mathcal{A}} w(s,a,a')}{\min_{(s,a) \in \mathcal{S} \times \mathcal{A}} \rho(s,a)}, \quad \forall (s,a,a') \in \mathcal{S} \times \mathcal{A} \times \mathcal{A}.$$

By plugging the primal $\eta$-policy $\tilde{\beta}_\eta$ into $a$ in the above inequality results in

$$\eta(1-\gamma) \min_{(s,a,a') \in \mathcal{S} \times \mathcal{A} \times \mathcal{A}} w(s,a,a')$$

$$\leq \tilde{Q}_\eta(s, \tilde{\beta}_\eta(s)) - R(s, \tilde{\beta}_\eta(s)) - \gamma \sum_{s' \in \mathcal{S}} P(s'|s, \tilde{\beta}_\eta(s)) \tilde{Q}_\eta(s', \tilde{\beta}_\eta(s'))$$

$$\leq \frac{\eta(1+\gamma) \sum_{(s,a,a') \in \mathcal{S} \times \mathcal{A} \times \mathcal{A}} w(s,a,a')}{\min_{(s,a) \in \mathcal{S} \times \mathcal{A}} \rho(s,a)}, \forall (s,a) \in \mathcal{S} \times \mathcal{A}.$$

Next, taking the expectation on both sides of the above inequality leads to

$$\eta(1-\gamma) \min_{(s,a,a') \in \mathcal{S} \times \mathcal{A} \times \mathcal{A}} w(s,a,a')$$

$$\leq \mathbb{E}[\tilde{Q}_\eta(s_k, \tilde{\beta}_\eta(s_k))|\tilde{\beta}_\eta, \rho] - \mathbb{E}[R(s_k, \tilde{\beta}_\eta(s_k))|\tilde{\beta}_\eta, \rho] - \gamma \mathbb{E}[\tilde{Q}_\eta(s_{k+1}, \tilde{\beta}_\eta(s_{k+1}))|\tilde{\beta}_\eta, \rho]$$

$$\leq \frac{\eta(1+\gamma) \sum_{(s,a,a') \in \mathcal{S} \times \mathcal{A} \times \mathcal{A}} w(s,a,a')}{\min_{(s,a) \in \mathcal{S} \times \mathcal{A}} \rho(s,a)}.$$

where the expectation is take with respect to the state $s_k$ and the next state $s_{k+1}$ generated under $\tilde{\beta}_\eta$ and the initial state distribution $\rho$. Multiplying both sides of the above inequality by $\gamma^k$ yields

$$\eta(1-\gamma)\gamma^k \min_{(s,a,a') \in \mathcal{S} \times \mathcal{A} \times \mathcal{A}} w(s,a,a')$$

$$\leq \gamma^k \mathbb{E}[\tilde{Q}_\eta(s_k, \tilde{\beta}_\eta(s_k))|\tilde{\beta}_\eta, \rho] - \gamma^k \mathbb{E}[R(s_k, \tilde{\beta}_\eta(s_k))|\tilde{\beta}_\eta, \rho] - \gamma^{k+1}\mathbb{E}[\tilde{Q}_\eta(s_{k+1}, \tilde{\beta}_\eta(s_{k+1}))|\tilde{\beta}_\eta, \rho]$$

$$\leq \frac{\eta(1+\gamma)\sum\limits_{(s,a,a')\in\mathcal{S}\times\mathcal{A}\times\mathcal{A}} w(s,a,a')}{\min_{(s,a)\in\mathcal{S}\times\mathcal{A}}\rho(s,a)}\gamma^k.$$

Now, summing them over all $k = 0, 1, 2, \ldots$ leads to

$$\eta \min_{(s,a,a')\in\mathcal{S}\times\mathcal{A}\times\mathcal{A}} w(s,a,a') \leq \mathbb{E}[\tilde{Q}_\eta(s_0, \tilde{\beta}_\eta(s_0))|\tilde{\beta}_\eta, \rho] - J^{\tilde{\beta}_\eta}$$

$$\leq s\frac{\eta(1+\gamma)\sum\limits_{(s,a,a')\in\mathcal{S}\times\mathcal{A}\times\mathcal{A}} w(s,a,a')}{(1-\gamma)\min_{(s,a)\in\mathcal{S}\times\mathcal{A}}\rho(s,a)},$$

where we use

$$J^{\tilde{\beta}_\eta} = \mathbb{E}\left[\sum_{k=0}^{\infty}\gamma^k R(s_k, \tilde{\beta}_\eta(s_k))\,\middle|\,\tilde{\beta}_\eta, \rho\right] = \sum_{k=0}^{\infty}\gamma^k\mathbb{E}[R(s_k, \tilde{\beta}_\eta(s_k))|\tilde{\beta}_\eta, \rho].$$

Rearranging terms leads to

$$\mathbb{E}[\tilde{Q}_\eta(s_0, \tilde{\beta}_\eta(s_0))] - \frac{\eta(1+\gamma)\sum\limits_{(s,a,a')\in\mathcal{S}\times\mathcal{A}\times\mathcal{A}} w(s,a,a')}{(1-\gamma)\min_{(s,a)\in\mathcal{S}\times\mathcal{A}}\rho(s,a)} \leq J^{\tilde{\beta}_\eta} \leq J^{\pi^*}. \tag{10}$$

Moreover, since $\tilde{Q}_\eta(s,a) \geq Q^*(s,a)$, we have

$$\sum_{s\in\mathcal{S}}\rho(s)\tilde{Q}_\eta(s, \tilde{\beta}_\eta(s)) \geq \sum_{s\in\mathcal{S}}\rho(s)\tilde{Q}_\eta(s, \pi^*(s)) \geq \sum_{s\in\mathcal{S}}\rho(s)Q^*(s, \pi^*(s)) = J^{\pi^*}. \tag{11}$$

Combining Equation (10) and Equation (11) leads to the second statement. $\square$

*Proof of the third statement.* The last statement can be easily obtained by combining the first and second statements. This completes the overall proof. $\square$

## L  APPENDIX: POLICY EVALUATION PROBLEM

In this section, we present a summary of several properties of the LP formulation associated with policy evaluation. Similar to the policy design problem discussed in the main text, a number of corresponding properties can be established through analogous arguments. For convenience of the reader, let us write the LP form of the policy evaluation problem again as follows:

$$\min_{Q\in\mathbb{R}^{|\mathcal{S}||\mathcal{A}|}} \sum_{(s,a)\in\mathcal{S}\times\mathcal{A}} \rho(s,a)Q(s,a) \tag{12}$$

subject to

$$R(s,a) + \gamma \sum_{(s',a')\in\mathcal{S}\times\mathcal{A}} P(s'|s,a)\pi(a'|s')Q(s',a') \leq Q(s,a), \quad (s,a)\in\mathcal{S}\times\mathcal{A},$$

where $R(s,a)$ is the expected reward conditioned on $(s,a)\in\mathcal{S}\times\mathcal{A}$, $\rho$ denotes any probability distribution over $\mathcal{S}\times\mathcal{A}$ with strictly positive support. For convenience, let us define the Bellman operator

$$(T^\pi Q)(s,a) := R(s,a) + \gamma \sum_{(s',a')\in\mathcal{S}\times\mathcal{A}} P(s'|s,a)\pi(a'|s')Q(s',a'), \quad (s,a)\in\mathcal{S}\times\mathcal{A}.$$

Analogous to the policy design problem, it can be established that the optimal solution of the above LP corresponds to the Q-function, $Q^\pi$.

**Lemma 10.** *The optimal solution of the LP in Equation* (12) *is unique and given by* $Q^\pi$

*Proof.* Let us assume that the optimal solution of the above LP in Equation (12) is $\hat{Q}$. Since $\hat{Q}$ is feasible, it satisfies

$$(T^\pi \hat{Q})(s,a) \leq \hat{Q}(s,a), \quad (s,a) \in \mathcal{S} \times \mathcal{A}.$$

Equivalently, it can be written as $T^\pi \hat{Q} \leq \hat{Q}$. Because $T$ is a monotone operator, we have

$$\lim_{k \to \infty} (T^\pi)^k \hat{Q} \leq \hat{Q} \Rightarrow Q^\pi \leq \hat{Q}.$$

Moreover, since $T^\pi Q^\pi = Q^\pi$, $Q^\pi$ is a feasible solution, and hence, $\hat{Q} \leq Q^\pi$. Therefore, $Q^\pi \leq \hat{Q} \leq Q^\pi$, which implies that $\hat{Q} = Q^\pi$. □

We next present the dual LP (Boyd and Vandenberghe, 2004, Chapter 5) corresponding to the policy evaluation LP.

**Lemma 11.** *The dual problem of the primal LP in Equation* (12) *is given by*

$$\max_{\lambda \geq 0} \sum_{(s,a) \in \mathcal{S} \times \mathcal{A}} \lambda(s,a) R(s,a)$$

subject to

$$\lambda(s,a) - \gamma \sum_{(i,j) \in S \times A} P(s|i,j)\pi(a|s)\lambda(i,j) - \rho(s,a) = 0, \quad \forall (s,a) \in \mathcal{S} \times \mathcal{A}.$$

*Proof.* Let us consider the Lagrangian function (Boyd and Vandenberghe, 2004, Chapter 5)

$$L(Q, \lambda) = \sum_{(s,a) \in \mathcal{S} \times \mathcal{A}} \rho(s,a) Q(s,a)$$

$$+ \sum_{(s,a) \in \mathcal{S} \times \mathcal{A}} \lambda(s,a)(R(s,a) + \gamma \sum_{(s',a') \in \mathcal{S} \times \mathcal{A}} P(s'|s,a)\pi(a'|s')Q(s',a') - Q(s,a)),$$

where $\lambda$ is the Lagrangian multiplier vector (or dual variable). Next, the function can be written by

$$L(Q, \lambda) = \sum_{(s,a) \in \mathcal{S} \times \mathcal{A}} \lambda(s,a) R(s,a)$$

$$+ \sum_{(s,a) \in \mathcal{S} \times \mathcal{A}} Q(s,a) \left\{ \lambda(s,a) - \gamma \sum_{(i,j) \in \mathcal{S} \times \mathcal{A}} P(s|i,j)\pi(a|s)\lambda(i,j) - \rho(s,a) \right\}.$$

One can observe that the dual problem $\max_{\lambda \geq 0} \min_{Q \in \mathbb{R}^{|\mathcal{S}||\mathcal{A}|}} L(Q, \lambda)$ is finite only when

$$\lambda(s,a) - \gamma \sum_{(i,j) \in \mathcal{S} \times \mathcal{A}} P(s|i,j)\pi(a|s)\lambda(i,j) - \rho(s,a) = 0, \quad \forall (s,a) \in \mathcal{S} \times \mathcal{A}.$$

Therefore, one gets the dual problem. □

In what follows, we investigate the properties of feasible solutions of the dual LP introduced above.

**Theorem 6.** *Suppose that the marginal*

$$\rho(s) = \sum_{a \in \mathcal{A}} \rho(s,a)$$

*represents the initial state distribution and* $\lambda(s,a), \forall (s,a) \in \mathcal{S} \times \mathcal{A}$ *is any dual feasible solution (all solutions satisfying the dual constraints). Then, we obtain the following results:*

1. *$(1 - \gamma)\lambda$ is a probability distribution, i.e.,*

$$(1 - \gamma)\lambda(s,a) \geq 0, \quad \forall (s,a) \in \mathcal{S} \times \mathcal{A}, \qquad \sum_{(s,a) \in \mathcal{S} \times \mathcal{A}} (1 - \gamma)\lambda(s,a) = 1.$$

2. *Moreover, let us define the marginal distributions*

$$\lambda(s) := \sum_{a \in \mathcal{A}} \lambda(s, a), \quad \rho(s) := \sum_{a \in \mathcal{A}} \rho(s, a).$$

*Then, we have*

$$\lambda(s') - \gamma \sum_{s \in \mathcal{S}} P^{\xi}(s'|s)\lambda(s) = \rho(s'),$$

*where*

$$\xi(\cdot|s) := \left[ \begin{array}{cccc} \frac{\lambda(s,1)}{\sum_{a \in \mathcal{A}} \lambda(s,a)} & \frac{\lambda(s,2)}{\sum_{a \in \mathcal{A}} \lambda(s,a)} & \cdots & \frac{\lambda(s,|\mathcal{A}|)}{\sum_{a \in \mathcal{A}} \lambda(s,a)} \end{array} \right]$$

*and*

$$P^{\xi}(s'|s) := \sum_{a \in \mathcal{A}} P(s'|s, a)\xi(a|s)$$

*is the probability of the transition from $s$ to $s'$ under the policy $\xi$.*

3. *$(1 - \gamma)\lambda(\cdot)$ is the discounted state occupation probability distribution defined as*

$$(1 - \gamma)\lambda(s) = \mu^{\xi}(s) = (1 - \gamma) \sum_{k=0}^{\infty} \gamma^k \mathbb{P}[s_k = s|\xi, \rho].$$

4. *The dual objective function satisfies*

$$\sum_{(s,a) \in \mathcal{S} \times \mathcal{A}} \lambda(s, a)R(s, a) = \frac{1}{1 - \gamma} \sum_{s \in \mathcal{S}} V^{\xi}(s)\rho(s) = \frac{1}{1 - \gamma} J^{\xi}$$

*Proof.* For the first statement, summing the dual constraints over $(s, a) \in \mathcal{S} \times \mathcal{A}$, we get

$$\Rightarrow \sum_{(s,a) \in \mathcal{S} \times \mathcal{A}} \lambda(s, a) - \gamma \sum_{(s,a) \in \mathcal{S} \times \mathcal{A}} \lambda(s, a) = 1,$$

which leads to

$$\sum_{(s,a) \in \mathcal{S} \times \mathcal{A}} \lambda(s, a) = 1.$$

It proves the first statement. For the second statement, summing the dual constraints in Equation (3) over $a \in \mathcal{A}$ leads to

$$\lambda(s, a) - \gamma \sum_{(i,j) \in \mathcal{S} \times \mathcal{A}} P(s|i, j)\lambda(i, j, a) = \rho(s, a).$$

Moreover, summing the above equation over $a \in \mathcal{A}$ leads to

$$\lambda(s) - \gamma \sum_{(i,j) \in \mathcal{S} \times \mathcal{A}} P(s|i, j)\lambda(i, j) = \rho(s),$$

which can be further written as

$$\lambda(s) - \gamma \sum_{(i,j) \in \mathcal{S} \times \mathcal{A}} P(s|i, j)\lambda(i, j)$$

$$= \lambda(s) - \gamma \sum_{(i,j) \in \mathcal{S} \times \mathcal{A}} P(s|i, j) \frac{\lambda(i, j)}{\sum_{a \in \mathcal{A}} \lambda(i, a)} \sum_{a \in \mathcal{A}} \lambda(i, a)$$

$$= \lambda(s) - \gamma \sum_{(i,j) \in \mathcal{S} \times \mathcal{A}} P(s|i, j)\xi(j|i)\lambda(i)$$

$$= \lambda(s) - \gamma \sum_{i \in \mathcal{S}} P^{\xi}(s|i)\lambda(i)$$

$$= \rho(s), \tag{13}$$

and this proves the second statement.

To prove the third statement, we can first consider the vectorized form of Equation (13)

$$\lambda - \gamma(P^\xi)^T \lambda = \rho,$$

where

$$\lambda = \begin{bmatrix} \lambda(1) \\ \vdots \\ \lambda(|\mathcal{S}|) \end{bmatrix} \in \mathbb{R}^{|\mathcal{S}|}, \quad \rho = \begin{bmatrix} \rho(1) \\ \vdots \\ \rho(|S|) \end{bmatrix} \in \mathbb{R}^{|\mathcal{S}|},$$

$$P^\xi = \begin{bmatrix} P^\xi(1|1) & \cdots & P^\pi(|\mathcal{S}||1) \\ \vdots & \ddots & \vdots \\ P^\xi(1||S|) & \cdots & P^\xi(|\mathcal{S}|||\mathcal{S}|) \end{bmatrix} \in \mathbb{R}^{|\mathcal{S}| \times |\mathcal{S}|},$$

which leads to

$$\lambda^T = \rho^T + \gamma \rho^T P_\xi + \gamma^2 \rho^T P_\xi^2 + \cdots.$$

The expression can be written as

$$\lambda(s) = \sum_{k=0}^\infty \gamma^k \mathbb{P}[s_k = s | \pi, \xi] = \frac{1}{1-\gamma} \mu^\xi(s).$$

This completes the proof of the third statement.

The last statement can be proved through the following identities:

$$\sum_{(s,a) \in \mathcal{S} \times \mathcal{A}} \lambda(s,a) R(s,a) = \sum_{(s,a) \in \mathcal{S} \times \mathcal{A}} \xi(a|s) R(s,a) \lambda(s)$$

$$= \sum_{s \in \mathcal{S}} V^\xi(s) \rho(s)$$

$$= J^\xi,$$

where $R^\xi$ denotes the expected reward under the policy $\xi$, and $V^\xi$ denotes the value function under $\xi$. $\qquad \square$

Based on Theorem 6, we can readily obtain the following corollary.

**Corollary 2.** *Suppose that the marginal* $\rho(s) := \sum_{a \in \mathcal{A}} \rho(s,a), s \in \mathcal{S}$ *represents the initial state distribution,*

$$\pi(a|s) := \frac{\rho(s,a)}{\sum\limits_{a \in \mathcal{A}} \rho(s,a)}$$

*represents* $\pi$, $Q^\pi$ *is the primal optimal solution, and* $\lambda^\pi$ *is the dual optimal solution. Then, the policy* $\xi$ *defined in the following is identical to* $\pi$ *in the sense that their corresponding MDP objective values are identical:*

$$\xi(\cdot|s) := \begin{bmatrix} \frac{\lambda^\pi(s,1)}{\sum_{a \in \mathcal{A}} \lambda^\pi(s,a)} & \frac{\lambda^\pi(s,2)}{\sum_{a \in \mathcal{A}} \lambda^\pi(s,a)} & \cdots & \frac{\lambda^\pi(s,|\mathcal{A}|)}{\sum_{a \in \mathcal{A}} \lambda^\pi(s,a)} \end{bmatrix}$$

*which is identical to the MDP objective function* $J^\pi$.

*Proof.* We first note that the optimal primal solution $Q^\pi$ does not depend on $\rho$, while the optimal dual solution $\lambda^\pi$ depends on $\rho$. Keeping this in mind, let us suppose that the marginal

$$\rho(s) = \sum_{a \in \mathcal{A}} \rho(s,a)$$

represents the initial state distribution and

$$\pi(a|s) := \frac{\rho(s,a)}{\sum\limits_{a \in \mathcal{A}} \rho(s,a)}$$

represents the policy $\pi$. The optimal primal objective function value is written as

$$\sum_{(s,a) \in \mathcal{S} \times \mathcal{A}} \rho(s,a) Q^\pi(s,a) = \sum_{(s,a) \in \mathcal{S} \times \mathcal{A}} \rho(s) \pi(a|s) Q^\pi(s,a)$$

$$= \sum_{s \in \mathcal{S}} \rho(s) V^\pi(s)$$

$$= J^\pi$$

On the other hand, the optimal dual objective function value is

$$\sum_{(s,a) \in \mathcal{S} \times \mathcal{A}} \lambda^\pi(s,a) R(s,a) = \sum_{(s,a) \in \mathcal{S} \times \mathcal{A}} \xi(a|s) R(s,a) \lambda^\pi(s)$$

$$= \sum_{s \in \mathcal{S}} R^\xi(s) \lambda^\pi(s)$$

$$= \sum_{s \in \mathcal{S}} V^\xi(s) \rho(s)$$

$$= J^\xi.$$

By the strong dualty, we have $J^\xi = J^\pi$, which means that the policy $\xi$ is identical to the policy $\pi$. $\square$

Let us consider the objective function with log-barrier function

$$f_\eta^\pi(Q) := \sum_{(s,a) \in \mathcal{S} \times \mathcal{A}} Q(s,a) \rho(s,a) + \eta \sum_{(s,a) \in \mathcal{S} \times \mathcal{A}} w(s,a) \varphi\left((T^\pi Q)(s,a) - Q(s,a)\right)$$

where $\eta > 0$ is a weight parameter and $w(s,a) > 0, (s,a) \in \mathcal{S} \times \mathcal{A}$ are weight parameters of the inequality constraints. Then, we can prove that the corresponding optimal solution $\tilde{Q}_\eta^\pi := \arg\min_{Q \in \mathcal{D}} f_\eta^\pi(Q)$ approximates $Q^\pi$. The following lemma establishes such an error bound between $\tilde{Q}_\eta$ and $Q^*$, and, in addition, presents a bound on the Bellman error corresponding to $\tilde{Q}_\eta$.

**Theorem 7.** *We have*

*1.* $\eta \min\limits_{(s,a) \in \mathcal{S} \times \mathcal{A}} w(s,a) < \left\| \tilde{Q}_\eta^\pi - Q^\pi \right\|_\infty \le \dfrac{\eta \sum\limits_{(s,a) \in \mathcal{S} \times \mathcal{A}} w(s,a)}{\min_{(s,a) \in \mathcal{S} \times \mathcal{A}} \rho(s,a)}$

*2.* $\eta(1-\gamma) \min\limits_{(s,a) \in \mathcal{S} \times \mathcal{A}} w(s,a) < \left\| \tilde{Q}_\eta^\pi - T^\pi \tilde{Q}_\eta^\pi \right\|_\infty \le \dfrac{(1+\gamma)\eta \sum\limits_{(s,a) \in \mathcal{S} \times \mathcal{A}} w(s,a)}{\min_{(s,a) \in \mathcal{S} \times \mathcal{A}} \rho(s,a)}$

To prove Theorem 7, we first need to prove the following lemma.

**Lemma 12.** *Let us define*

$$\tilde{\lambda}_\eta(s,a) := -\frac{\eta w(s,a)}{(T^\pi \tilde{Q}_\eta^\pi)(s,a) - \tilde{Q}_\eta^\pi(s,a)}, \quad (s,a) \in \mathcal{S} \times \mathcal{A}.$$

*Then, we have the following results:*

*1.* $\eta(1-\gamma) \min\limits_{(s,a) \in \mathcal{S} \times \mathcal{A}} w(s,a) < \left\| \tilde{Q}_\eta^\pi - T^\pi \tilde{Q}_\eta^\pi \right\|_\infty$

*2.* $\tilde{Q}_\eta^\pi(s,a) - Q^\pi(s,a) > \eta \min\limits_{(s,a) \in \mathcal{S} \times \mathcal{A}} w(s,a), \quad \forall (s,a) \in \mathcal{S} \times \mathcal{A}$

*3.* $\eta \min\limits_{(s,a) \in \mathcal{S} \times \mathcal{A}} w(s,a) < \left\| \tilde{Q}_\eta^\pi - Q^\pi \right\|_\infty$

*Proof.* Following similar lines as in the proof of Theorem 3, we can derive

$$0 < \frac{\eta w(s,a)(1-\gamma)}{\tilde{Q}_\eta^\pi(s,a) - (T^\pi \tilde{Q}_\eta^\pi)(s,a)} < 1, \quad \forall (s,a) \in \mathcal{S} \times \mathcal{A},$$

which implies

$$\eta \min_{(s,a)\in\mathcal{S}\times\mathcal{A}} w(s,a)(1-\gamma) < \tilde{Q}_\eta^\pi(s,a) - (T^\pi \tilde{Q}_\eta^\pi)(s,a), \quad \forall (s,a) \in \mathcal{S} \times \mathcal{A},$$

By taking the absolute value on the right-hand side and taking the maximum over $(s,a) \in \mathcal{S} \times \mathcal{A}$, we can obtain the first statement.

Next, for the second statement, we first note the following bounds:

$$\eta(1-\gamma) \min_{(s,a)\in\mathcal{S}\times\mathcal{A}} w(s,a)$$
$$< \tilde{Q}_\eta^\pi(s,a) - (T^\pi \tilde{Q}_\eta^\pi)(s,a)$$
$$= \tilde{Q}_\eta(s,a) - (T^\pi \tilde{Q}_\eta)(s,a) - Q^\pi(s,a) + (T^\pi Q^\pi)(s,a)$$
$$= \tilde{Q}_\eta^\pi(s,a) - Q^\pi(s,a) - \gamma \sum_{s'\in\mathcal{S}} P(s'|s,a) \sum_{a'\in\mathcal{A}} \pi(a'|s') \{ \tilde{Q}_\eta^\pi(s',a') - Q^\pi(s',a') \}$$
$$\leq \tilde{Q}_\eta(s,a) - Q^*(s,a) - \gamma \sum_{s'\in\mathcal{S}} P(s'|s,a) \min_{(s',a')\in\mathcal{S}\times\mathcal{A}} \{ \tilde{Q}_\eta^\pi(s',a') - Q^\pi(s',a') \}$$
$$\leq \tilde{Q}_\eta^\pi(s,a) - Q^\pi(s,a) - \gamma \min_{(s',a')\in\mathcal{S}\times\mathcal{A}} \{ \tilde{Q}_\eta^\pi(s',a') - Q^\pi(s',a') \}, \quad \forall (s,a) \in \mathcal{S} \times \mathcal{A},$$

where we use the Bellman equation $Q^\pi = T^\pi Q^\pi$ in the first equality.

Taking the minimum over $(s,a) \in \mathcal{S} \times \mathcal{A}$ on both sides yields

$$(1-\gamma)\eta \min_{(s,a)\in\mathcal{S}\times\mathcal{A}} w(s,a) < (1-\gamma) \min_{(s',a')\in\mathcal{S}\times\mathcal{A}} \{ \tilde{Q}_\eta^\pi(s',a') - Q^\pi(s',a') \},$$

which is the second statement. Moreover, it also implies

$$\eta \min_{(s,a)\in\mathcal{S}\times\mathcal{A}} w(s,a) < \min_{(s',a')\in\mathcal{S}\times\mathcal{A}} \{ \tilde{Q}_\eta^\pi(s',a') - Q^\pi(s',a') \}$$
$$\leq \min_{(s',a')\in\mathcal{S}\times\mathcal{A}} |\tilde{Q}_\eta^\pi(s',a') - Q^\pi(s',a')|$$
$$\leq \max_{(s',a')\in\mathcal{S}\times\mathcal{A}} |\tilde{Q}_\eta^\pi(s',a') - Q^\pi(s',a')|$$
$$= \left\| \tilde{Q}_\eta^\pi - Q^\pi \right\|_\infty,$$

which proves the last statement. $\qquad\square$

Based on Lemma 12, we will now prove Theorem 7.

*Proof of the first statement of Theorem 7.* Because the optimal solution $\tilde{Q}_\eta^\pi$ should be in the domain of the log-barrier function, it should be strictly feasible for the primal LP. Moreover, $\tilde{\lambda}_\eta^\pi$ is feasible for the dual LP. Using these results, we can obtain the following lower bounds:

$$\sum_{(s,a)\in\mathcal{S}\times\mathcal{A}} \rho(s,a)Q^\pi(s,a)$$
$$= \sum_{(s,a)\in\mathcal{S}\times\mathcal{A}} \lambda^\pi(s,a)R(s,a)$$
$$\geq \sum_{(s,a)\in\mathcal{S}\times\mathcal{A}} \tilde{\lambda}_\eta^\pi(s,a)R(s,a)$$
$$= \sum_{(s,a)\in\mathcal{S}\times\mathcal{A}} \tilde{\lambda}_\eta^\pi(s,a)R(s,a)$$

$$+ \sum_{(s,a)\in\mathcal{S}\times\mathcal{A}} \tilde{Q}_\eta^\pi(s,a)\left\{\rho(s,a) + \gamma \sum_{(s',a')\in\mathcal{S}\times\mathcal{A}} \tilde{\lambda}_\eta^\pi(s',a')P(s|s',a') - \tilde{\lambda}_\eta^\pi(s,a)\right\}$$

$$= L(\tilde{Q}_\eta^\pi, \tilde{\lambda}_\eta^\pi)$$

$$= \sum_{(s,a)\in\mathcal{S}\times\mathcal{A}} \rho(s,a)\tilde{Q}_\eta^\pi(s,a) + \sum_{(s,a)\in\mathcal{S}\times\mathcal{A}} \tilde{\lambda}_\eta^\pi(s,a)\{(T^\pi\tilde{Q}_\eta^\pi)(s,a) - \tilde{Q}_\eta^\pi(s,a)\}$$

$$= \sum_{(s,a)\in\mathcal{S}\times\mathcal{A}} \rho(s,a)\tilde{Q}_\eta^\pi(s,a) + \sum_{(s,a)\in\mathcal{S}\times\mathcal{A}} -\eta\frac{w(s,a)\{(T^\pi\tilde{Q}_\eta^\pi)(s,a) - \tilde{Q}_\eta^\pi(s,a)\}}{(T^\pi\tilde{Q}_\eta^\pi)(s,a,a') - \tilde{Q}_\eta^\pi(s,a)}$$

$$= \sum_{(s,a)\in\mathcal{S}\times\mathcal{A}} \rho(s,a)\tilde{Q}_\eta^\pi(s,a) - \eta\sum_{(s,a)\in\mathcal{S}\times\mathcal{A}} w(s,a), \tag{14}$$

where the second line is due to the strong duality of the LP problem, the third and fourth lines are due to the fact that $\tilde{\lambda}_\eta^\pi$ is dual feasible, and the sixth line is obtained by rearranging terms. On the other hand, since $\tilde{Q}_\eta^\pi \in \mathcal{D}$ is primal feasible, we have

$$\sum_{(s,a)\in\mathcal{S}\times\mathcal{A}} \rho(s,a)Q^*(s,a) \le \sum_{(s,a)\in\mathcal{S}\times\mathcal{A}} \rho(s,a)\tilde{Q}_\eta^\pi(s,a). \tag{15}$$

Combining Equation (14) and Equation (15) leads to

$$0 \le \sum_{(s,a)\in\mathcal{S}\times\mathcal{A}} \rho(s,a)\{\tilde{Q}_\eta^\pi(s,a) - Q^\pi(s,a)\} \le \eta\sum_{(s,a)\in\mathcal{S}\times\mathcal{A}} w(s,a).$$

Using Lemma 12 and $\|\cdot\|_1 \ge \|\cdot\|_\infty$, we further derive

$$\sum_{(s,a)\in\mathcal{S}\times\mathcal{A}} \rho(s,a)\{\tilde{Q}_\eta^\pi(s,a) - Q^\pi(s,a)\} \ge \min_{(s,a)\in\mathcal{S}\times\mathcal{A}}\rho(s,a) \sum_{(s,a)\in\mathcal{S}\times\mathcal{A}} (\tilde{Q}_\eta^\pi(s,a) - Q^\pi(s,a))$$

$$= \min_{(s,a)\in\mathcal{S}\times\mathcal{A}}\rho(s,a)\left\|\tilde{Q}_\eta^\pi - Q^\pi\right\|_1$$

$$\ge \min_{(s,a)\in\mathcal{S}\times\mathcal{A}}\rho(s,a)\left\|\tilde{Q}_\eta^\pi - Q^\pi\right\|_\infty.$$

where the first inequality follows from $\tilde{Q}_\eta^\pi(s,a) - Q^\pi(s,a) > 0$. By dividing both sides by $\min_{(s,a)\in\mathcal{S}\times\mathcal{A}}\rho(s,a) > 0$ leads to

$$\left\|\tilde{Q}_\eta^\pi - Q^\pi\right\|_\infty \le \frac{\eta\sum_{(s,a)\in\mathcal{S}\times\mathcal{A}} w(s,a)}{\min_{(s,a)\in\mathcal{S}\times\mathcal{A}}\rho(s,a)}.$$

The lower bounds come from the second statement of Lemma 8. □

*Proof of the second statement of Theorem 7.* Next, using the reverse triangle inequality, one gets

$$\left\|\tilde{Q}_\eta^\pi - Q^\pi\right\|_\infty = \left\|\tilde{Q}_\eta^\pi - T^\pi\tilde{Q}_\eta^\pi + T^\pi\tilde{Q}_\eta^\pi - Q^\pi\right\|_\infty$$

$$\ge \left\|\tilde{Q}_\eta^\pi - T^\pi\tilde{Q}_\eta^\pi\right\|_\infty - \left\|T^\pi\tilde{Q}_\eta^\pi - T^\pi Q^\pi\right\|_\infty$$

$$\ge \left\|\tilde{Q}_\eta^\pi - T^\pi\tilde{Q}_\eta^\pi\right\|_\infty - \gamma\left\|\tilde{Q}_\eta^\pi - Q^\pi\right\|_\infty,$$

where the last inequality comes from the contraction of the Bellman operator.

Rearranging terms on both sides results in

$$\left\|\tilde{Q}_\eta^\pi - T^\pi\tilde{Q}_\eta^\pi\right\|_\infty \le (1+\gamma)\left\|\tilde{Q}_\eta^\pi - Q^\pi\right\|_\infty \le \frac{(1+\gamma)\eta\sum_{(s,a)\in\mathcal{S}\times\mathcal{A}} w(s,a)}{\min_{(s,a)\in\mathcal{S}\times\mathcal{A}}\rho(s,a)}.$$

The lower bounds come from the first statement of Lemma 12, and this completes the proof. □

## M  APPENDIX: PROPOSITION 1

In Section 6, we introduce a novel DQN algorithm inspired by the idea of standard DQN (Mnih et al., 2015) and consider the following loss function:

$$L(\theta) := \frac{1}{|B|} \sum_{(s,a,r,s') \in B, a' \in \mathcal{A}} [Q_\theta(s,a) + \eta\varphi(r + \gamma Q_\theta(s',a') - Q_\theta(s,a))],$$

where $B$ is a mini-batch uniformly sampled from the experience replay buffer $D$, $|B|$ is the size of the mini-batch, $Q_\theta$ is a deep neural network approximation of Q-function, $\theta \in \mathbb{R}^m$ is the parameter to be determined, and $(s, a, r, s')$ is the transition sample of the state-action-reward-next state. The loss function can be seen as a stochastic approximation of $f_\eta$, where $\rho$ and $w$ can be set to probability distributions corresponding to the replay buffer. However, this stochastic approximation is generally biased because it approximates a function in which the state transition probabilities appear outside the log-barrier function. In fact, by applying Jensen's inequality, we can show that the above loss function is essentially an unbiased stochastic approximation of an upper bounding surrogate function of $f_\eta$. The related result is first introduced below.

**Proposition 1.** *We have*

$$f_\eta(Q) \leq \sum_{(s,a) \in \mathcal{S} \times \mathcal{A}} \rho(s,a)Q(s,a)$$

$$+ \eta \sum_{(s,a,s',a') \in \mathcal{S} \times \mathcal{A} \times \mathcal{S} \times \mathcal{A}} P(s'|s,a)w(s,a,a')\varphi\left(r(s,a,s') + \gamma Q(s',a') - Q(s,a)\right)$$

$$=: g_\eta(Q)$$

*Proof.* It can be proved directly using Jensen's inequality. □

The above result employed Jensen's inequality to derive the upper-bounding surrogate function $g_\eta$. Relating this to the loss function $L$, we observe that the surrogate function can be obtained by taking the expectation of $L$. In other words, under the assumption that the mini-batch samples are i.i.d., the loss function $L$ can be shown to be an unbiased stochastic approximation of the upper surrogate function. Of course, as noted in the main text, if the environment is deterministic, then the original function $f_\eta$ and the upper surrogate function $g_\eta$ are identical. Therefore, $L$ is no longer a stochastic approximation of the upper surrogate function; it is now an exact representation.

## N  APPENDIX: LOG-BARRIER DQN ALGORITHM

Here, we discuss the implementation details of the deep RL variant of the proposed method. Since the loss function $L$

$$L(\theta) := \frac{1}{|B|} \sum_{(s,a,r,s') \in B, a' \in \mathcal{A}} [Q_\theta(s,a) + \eta\varphi(r + \gamma Q_\theta(s',a') - Q_\theta(s,a))],$$

relies on random samples of state–action pairs, applying gradient descent does not update the parameters for all state–action pairs simultaneously. As a result, there is no guarantee that $Q_\theta$ strictly satisfies the inequality constraints of the LP during the update process. In fact, $Q_\theta$ corresponding to state–action pairs that are not sampled may move outside the feasible region defined by the inequality constraints. Such behavior can significantly undermine the stability of learning. To address this issue, in practical implementations, we replace $L$ with the following alternative loss function $\tilde{L}$ as follows:

$$\tilde{L}(\theta) := \frac{1}{|B|} \sum_{(s,a,r,s') \in B} \left[Q_\theta(s,a) + \eta \sum_{a' \in A} h(r + \gamma Q_\theta(s',a') - Q_\theta(s,a))\right]$$

where

$$h(x) := \begin{cases} \varphi(x-\varepsilon) & \text{if} \quad x < 0 \\ \nu x & \text{if} \quad x \geq 0 \end{cases}$$

---

**Algorithm 1** Log-barrier deep Q-learning

1: Initialize replay memory $D$ to capacity $|D|$
2: Initialize the parameter $\theta \in \mathbb{R}^m$ such that $Q_\theta \cong \kappa \mathbf{1}_{|\mathcal{S}||\mathcal{A}|}$ with large enough $\kappa > 0$
3: Set $k = 0$
4: **for** Episode $i \in \{1, 2, \ldots\}$ **do**
5: $\quad$ Observe $s_0$
6: $\quad$ **for** $t \in \{0, 1, \ldots, \tau - 1\}$ **do**
7: $\quad\quad$ Take an action $a_t$ according to

$$a_t = \begin{cases} \arg\max_{a \in \mathcal{A}} Q_\theta(s_t, a) & \text{with probability } 1 - \varepsilon \\ a \sim \text{uniform}(\mathcal{A}) & \text{with probability } \varepsilon \end{cases}$$

8: $\quad\quad$ Observe $r_{t+1}, s_{t+1}$
9: $\quad\quad$ Store the transition $(s_t, a_t, r_{t+1}, s_{t+1})$ in $D$
10: $\quad\quad$ Sample uniformly a random mini-batch $B$ of transitions $(s, a, r, s')$ from $D$
11: $\quad\quad$ Perform few gradient descent steps

$$\theta \leftarrow \theta - \alpha \nabla_\theta \tilde{L}(\theta)$$

12: $\quad\quad$ Set $k \leftarrow k + 1$
13: $\quad$ **end for**
14: **end for**

---

$\varepsilon > 0$ is a small number, e.g. $\varepsilon = 10^{-6}$, added for numerical stability and $\nu > 0$ is a large number, e.g. $\nu = 10^3$, in order to enforce $Q_\theta$ to be feasible when the TD error $r + \gamma Q_\theta(s', a') - Q_\theta(s, a)$ is nonnegative. Based on these details, the overall deep RL version is given in Algorithm 1. In Algorithm 1, $\mathbf{1}_{|\mathcal{S}||\mathcal{A}|} \in \mathbb{R}^{|\mathcal{S}||\mathcal{A}|}$ is the vector with all entries one. The initialization, $Q_\theta \cong \kappa \mathbf{1}_{|\mathcal{S}||\mathcal{A}|}$, is motivated by the observation that a candidate strictly feasible solution to the LP formulation in Equation (1) takes the form $\kappa \mathbf{1}_{|\mathcal{S}||\mathcal{A}|}$ with a large $\kappa > 0$. To implement this initialization within a deep neural network, one may set all weights to zero while retaining only a constant bias term.

## O    APPENDIX: LOG-BARRIER DDPG ALGORITHM

Here, we discuss the implementation details of the deep RL variant of the proposed method for policy evaluation. Since the loss function $L_{\text{critic}}(\theta; \pi_w)$

$$L_{\text{critic}}(\theta; \pi_w) := \frac{1}{|B|} \sum_{(s,a,r,s') \in B} [Q_\theta(s, a) + \eta\varphi(r + \gamma Q_\theta(s', \pi_w(s')) - Q_\theta(s, a))],$$

for policy evaluation relies on random samples of state–action pairs, applying gradient descent does not update the parameters for all state–action pairs simultaneously. Similar to the previous section, in practical implementations, we replace $L$ with the following alternative loss function $\tilde{L}$ as follows:

$$\tilde{L}_{\text{critic}}(\theta; \pi_w) := \frac{1}{|B|} \sum_{(s,a,r,s') \in B} [Q_\theta(s, a) + \eta h(r + \gamma Q_\theta(s', \pi_w(s')) - Q_\theta(s, a))]$$

where

$$h(x) := \begin{cases} \varphi(x - \varepsilon) & \text{if } x < 0 \\ \nu x & \text{if } x \geq 0 \end{cases}$$

$\varepsilon > 0$ is a small number, e.g. $\varepsilon = 10^{-6}$, added for numerical stability and $\nu > 0$ is a large number, e.g. $\nu = 10^3$, in order to enforce $Q_\theta$ to be feasible when the TD error $r + \gamma Q_\theta(s', \pi_w(s')) - Q_\theta(s, a)$ is nonnegative. As usual, the actor loss function (Lillicrap et al., 2015) is given by

$$L_{\text{actor}}(w) := \frac{1}{|B|} \sum_{(s,a,r,s') \in B} (-Q_\theta(s, \pi_w(s))).$$

The overall detailed algorithm is shown in Algorithm 2.

---

**Algorithm 2** Log-barrier DDPG

---

1: Initialize replay memory $D$ to capacity $|D|$
2: Initialize the critic parameter $\theta \in \mathbb{R}^{m_1}$ such that $Q_\theta \cong \kappa \mathbf{1}_{|S||A|}$ with large enough $\kappa > 0$ and the target critic parameter $\theta' = \theta \in \mathbb{R}^{m_1}$
3: Initialize the online actor parameter $w \in \mathbb{R}^{m_2}$ and the target actor parameter $w' = w \in \mathbb{R}^{m_2}$
4: Set $k = 0$
5: **for** Episode $i \in \{1, 2, \ldots\}$ **do**
6:     Observe $s_0$
7:     **for** $t \in \{0, 1, \ldots, T-1\}$ **do**
8:         Take an action $a_t = \pi_w(s_t) + e_t$ where $e_t \sim \mathcal{N}(0, \sigma)$ is the exploration noise and $\mathcal{N}(0, \sigma)$ is the normal distribution.
9:         Observe $r_{t+1}, s_{t+1}$
10:         Store the transition $(s_t, a_t, r_{t+1}, s_{t+1})$ in $D$
11:         Sample uniformly a random mini-batch $B$ of transitions $(s, a, r, s')$ from $D$
12:         Critic update: perform few gradient descent steps

$$\theta \leftarrow \theta - \alpha_{\text{critic}} \nabla_\theta \tilde{L}_{\text{critic}}(\theta; \pi_w)$$

13:         Actor update: perform a gradient descent step

$$w \leftarrow w + \alpha_{\text{actor}} \nabla_w L_{\text{actor}}(w)$$

14:         Soft target network update

$$w' \leftarrow \tau w + (1 - \tau) w'$$
$$\theta' \leftarrow \tau \theta + (1 - \tau) \theta'$$

15:         Set $k \leftarrow k + 1$
16:     **end for**
17: **end for**

---

In Algorithm 2, we adopt settings that are consistent with the original DDPG paper (Lillicrap et al., 2015). In particular, we employ target networks for both actor and critic and apply the soft target updates. Moreover, Algorithm 2 incorporates exploration through Gaussian noise. Our main deviations lie in the structure of the critic loss and the initialization of the critic network. As presented in Section P, we initialize the critic network bias with $\kappa = 100$, set the enforcement parameter to $\nu = 100$, and use a barrier parameter of $\eta = 0.015$ in the critic loss.

## P   APPENDIX: HYPERPARAMETERS

We summarize in the following tables the hyperparameter configurations employed in the experiments for the proposed log-barrier deep Q-learning and log-barrier DDPG. Specifically, Table 1 lists the hyperparameters for the log-barrier deep Q-learning experiments, and Table 2 lists those for the log-barrier DDPG experiments. In the comparative experiments, the parameters of standard DQN and standard DDPG were tuned to achieve optimal performance. For fairness, common hyperparameters were controlled to be identical under the same experimental conditions.

Table 1: Hyperparameter settings of log-barrier deep Q-learning for the discrete control

| Hyperparameter | Value |
| --- | --- |
| Initial network bias ($\kappa$) | 100 |
| Enforce parameter ($\nu$) | 1000 |
| Discount factor ($\gamma$) | 0.99 |
| Batch size | 64 |
| Epsilon ($\epsilon$) | 0.1 |
| Episodes | 200 (CartPole-v1, Acrobot-v1) |
| | 500 (LunarLander-v3, MountainCar-v0, Pendulum-v1) |
| Barrier parameter ($\eta$) | 7 (CartPole-v1), 0.25 (Acrobot-v1) |
| | 2 (LunarLander-v3), 0.4, (MountainCar-v0), 1.5 (Pendulum-v1) |
| Learning rate | 0.0002 (LunarLander-v3), 0.0005 (otherwise) |
| Replay buffer size | 100,000 (CartPole-v1, Acrobot-v1) |
| | 1,000,000 (LunarLander-v3, MountainCar-v0, Pendulum-v1) |

Table 2: Hyperparameter settings for log-barrier DDPG on MuJoCo tasks

| Hyperparameter | Value |
| --- | --- |
| Initial network bias ($\kappa$) | 100 |
| Enforce parameter ($\nu$) | 100 |
| Discount factor ($\gamma$) | 0.99 |
| Batch size | 256 |
| Epsilon ($\epsilon$) | 0.01 |
| Barrier parameter ($\eta$) | 0.015 |
| Exploration noise ($\sigma$) | 0.1 (Gaussian) |
| Total timesteps | 2M steps equivalent (Walker2d, Ant, Hopper, HalfCheetah, Humanoid) |
| Target update rate ($\tau$) | 0.005 |
| Actor learning rate | 0.001 |
| Critic learning rate | 0.001 |
| Replay buffer size | 1,000,000 |
| Hidden units per layer | 256 |
| Number of hidden layers | 2 |

## Q  GENERAL REMARKS

Several discussions that are not included in the main text due to page limits are summarized below.

1. **Analysis under deep neural network function approximation**: When a deep neural network is used, the model becomes a nonlinear function of the parameters $\theta$, and the precise theoretical results derived for the tabular setting no longer apply directly. Nevertheless, the tabular analysis provides useful intuition: it clarifies the role of the log-barrier formulation, the effect of the barrier coefficient eta, and the qualitative behavior one should expect when using function approximation, and thus offers approximate predictions for the deep-RL case.

2. **Sensitivity of learning performance on** $\eta$: We found the proposed methods to be generally highly sensitive to the barrier coefficient $\eta$. Accordingly, we performed careful hyperparameter tuning for the experiments, and we acknowledge that this sensitivity is an important issue that requires further investigation. We can try an annealing scheme that gradually reduces $\eta$ during training, and this often yields improved results. However, annealing itself introduces additional tuning choices: if $\eta$ is decreased via a schedule or learned online, one must still decide the initial value, the decay rate (how quickly to reduce $\eta$, and the final value, and experimental outcomes can vary substantially depending on these choices). Nonetheless, these auxiliary techniques frequently enhance empirical performance at the cost of increase tunning efforts. To free the method from excessive tuning, we plan to develop improved algorithms that are less dependent on hyperparameter selection. For

example, mirror-descent–style updates that operate directly in the space satisfying the inequality constraints could maintain feasibility without heavy reliance on $\eta$. We view these directions as promising topics for future work.

3. **Effect of small** $\eta$: Theoretically, as $\eta$ goes to zero, the log-barrier solution $\tilde{Q}_\eta$ approaches the true $Q^*$. This relationship is established in Theorem 1. In practice, however, taking $\eta$ too small can cause numerical issues. For example, with very small $\eta$, the gradient-based estimate of $\tilde{Q}_\eta$ may easily step outside the domain that satisfies the inequality constraints with relevantly small step-sizes.

4. **Potential applicability to constrainted RL**: The proposed approach can be integrated very naturally into constrained or safe RL by, for example, incorporating a log-barrier term. However, this introduces additional tuning issue, namely the need to carefully set additional additional barrier coefficients. Consequently, we argue that a more principled framework, with stronger theoretical guarantees and improved algorithmic mechanisms for handling such hyperparameters, is necessary.

## R  APPENDIX: ADDITIONAL EXPERIMENTS OF LOG-BARRIER Q-LEARNING APPROACH WITH LINEAR FUNCTION APPROXIMATION

In this appendix section, we present additional experiments comparing standard Q-learning and a modified version of our log-barrier Q-learning in linear function approximation settings.

### R.1  ENVIRONMENT

#### R.1.1  MARKOV DECISION PROCESS SETTING

To make the comparison concrete, we consider a toy Markov decision process (MDP) with $|\mathcal{S}| = 4$ states and $|\mathcal{A}| = 2$ actions. The expected reward $R(s, a)$ used in our simulations is defined as below:

$$R(s, a) = \begin{pmatrix} 0.5 & 1.0 \\ 0.2 & 0.4 \\ 0.8 & 0.3 \\ 1.5 & 0.1 \end{pmatrix},$$

where each row corresponds to a state $s \in \{0, 1, 2, 3\}$ and each column corresponds to an action $a \in \{0, 1\}$. The transition probability $P(s' \mid s, a)$ is described by two state–transition matrices, one for each action. For action $a = 0$, we have

$$P(s' \mid s, a = 0) = \begin{pmatrix} 0.7 & 0.3 & 0.0 & 0.0 \\ 0.0 & 0.6 & 0.0 & 0.4 \\ 0.5 & 0.0 & 0.5 & 0.0 \\ 0.6 & 0.4 & 0.0 & 0.0 \end{pmatrix},$$

and for action $a = 1$, the transition matrix is

$$P(s' \mid s, a = 1) = \begin{pmatrix} 0.0 & 0.4 & 0.6 & 0.0 \\ 0.0 & 0.0 & 0.8 & 0.2 \\ 0.0 & 0.0 & 0.0 & 1.0 \\ 0.0 & 0.0 & 0.0 & 1.0 \end{pmatrix}.$$

Figure 7 provides a clear visual illustration of the environment described above.

#### R.1.2  LINEAR FUNCTION APPROXIMATION

We next construct a linear function approximator with a fixed basis matrix of rank 6. This basis is deliberately designed so that several state–action pairs share identical or nearly collinear feature vectors, which allows us to highlight the well-known instability and under-performance of linear function approximation under off-policy Q-learning updates. The basis matrix $\Phi \in \mathbb{R}^{8 \times 6}$, where

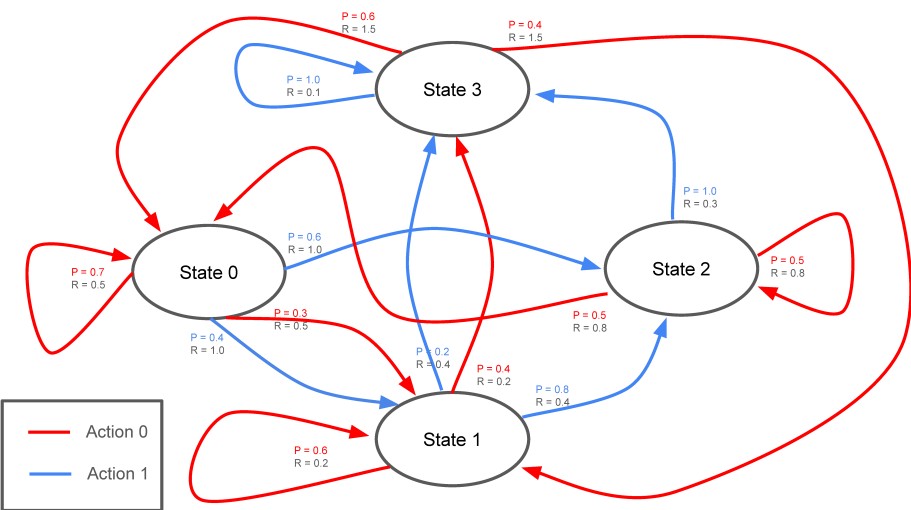

Figure 7: Visualization of the transition structure and reward layout of the toy MDP. Transitions under action 0 are shown as red arrows, while those under action 1 are shown as blue arrows. The corresponding reward values are indicated in black text.

each row corresponds to a state–action pair $(s, a)$, is defined as

$$\Phi = \begin{pmatrix} 1.0 & 0.0 & 0.0 & 0.1 & 0.0 & 0.0 \\ 0.1 & 0.0 & 0.0 & 1.0 & 0.0 & 0.0 \\ 1.0 & 0.0 & 0.0 & 0.1 & 0.0 & 0.0 \\ 0.1 & 0.0 & 0.0 & 1.0 & 0.0 & 0.0 \\ 0.0 & 1.0 & 0.0 & 0.0 & 0.1 & 0.0 \\ 0.0 & 0.1 & 0.0 & 0.0 & 1.0 & 0.0 \\ 0.0 & 0.0 & 1.0 & 0.0 & 0.0 & 0.1 \\ 0.0 & 0.0 & 0.1 & 0.0 & 0.0 & 1.0 \end{pmatrix},$$

where rows of $\Phi$ are ordered as $(s, a) = (0,0), (0,1), (1,0), (1,1), (2,0), (2,1), (3,0), (3,1)$. Given this basis, the linear Q-function is represented as

$$Q_w(s, a) = \phi(s, a)^\top w, \qquad w \in \mathbb{R}^6,$$

where $\phi(s, a)$ denotes the corresponding row of $\Phi$. Because states $s = 0$ and $s = 1$ share identical feature patterns, the resulting approximation is intentionally limited in expressive power, making it a suitable scenario for examining divergence phenomena in both standard Q-learning and our log-barrier-adjusted updates.

To satisfy the theoretical precondition of the log-barrier approach, namely that the initial Q-values are sufficiently large, we initialize the parameter vector $w$ so that all state–action values start from a common high level. Concretely, let $q_{\text{init}} \in \mathbb{R}$ be a prescribed initialization constant and define

$$\mathbf{q}_0 \in \mathbb{R}^{|S||A|}, \qquad \mathbf{q}_0 := q_{\text{init}}\, \mathbf{1},$$

where $\mathbf{1} \in \mathbb{R}^{|\mathcal{S}||\mathcal{A}|}$ denotes the all-ones vector of dimension $|\mathcal{S}||\mathcal{A}|$. We then choose the initial parameter $w_0$ by solving the least-squares problem

$$w_0 = \arg\min_{w \in \mathbb{R}^6} \left\| \Phi w - \mathbf{q}_0 \right\|_2^2.$$

In practice, this corresponds to computing the least-squares solution

$$\Phi w_0 \approx \mathbf{q}_0,$$

so that $Q_{w_0}(s, a) \approx q_{\text{init}}$ holds for all $(s, a) \in \mathcal{S} \times \mathcal{A}$.

### R.1.3 STANDARD Q-LEARNING

The standard Q-learning algorithm serves as a baseline method. Following the classical formulation of semi-gradient Q-learning with linear function approximation (Sutton and Barto, 1998), the update rules are given as follows:

$$\text{Loss:} \quad L(w) = \frac{1}{2}(\text{no\_grad}[r(s,a,s') + \gamma \max_{a'} Q_w(s',a')] - Q_w(s,a))^2$$

$$\text{Gradient:} \quad \nabla_w L(w) = -\delta \, \phi(s,a), \qquad \delta = r(s,a,s') + \gamma \max_{a' \in \mathcal{A}} Q_w(s',a') - Q_w(s,a),$$

$$\text{Update:} \quad w \leftarrow w + \alpha \, \delta \, \phi(s,a). \tag{R.1}$$

where $(s,a,s',a')$ means the current state, current action, next state, next action, and $\text{no\_grad}$ means that the corresponding term is set as a constant.

### R.1.4 LOG-BARRIER Q-LEARNING

Next, we consider our proposed method, the log-barrier Q-learning approach. Although the implementation in Appendix N employs a deep neural network $Q_\theta$ to parameterize the Q-function, here we replace it with a linear function approximation of the form $Q_w = \Phi w$. Under this parametrization, we can derive a gradient-based update rule by directly computing the loss and its corresponding parameter update as:

Loss:

$$L^{LB}(w) = \begin{cases} Q_w(s,a) - \eta \, \log\big(Q_w(s,a) - (TQ_w)(s,a,s') + \varepsilon\big), & \text{if } Q_w(s,a) - (TQ_w)(s,a,s') > 0, \\ Q_w(s,a) + \eta \, \nu \, \big((TQ_w)(s,a,s') - Q_w(s,a)\big), & \text{if } Q_w(s,a) - (TQ_w)(s,a,s') \leq 0, \end{cases}$$

where $(TQ_w)(s,a,s') = r(s,a,s') + \gamma \max_{u \in \mathcal{A}} Q_w(s',u)$.

Gradient:

$$\nabla_w L^{LB}(w) = \begin{cases} \phi(s,a) - \eta \, \dfrac{\phi(s,a) - \gamma \, \phi(s',a^*)}{Q_w(s,a,s') - (TQ_w)(s,a,s') + \varepsilon}, & \text{if } Q_w(s,a) - (TQ_w)(s,a,s') > 0, \\ \phi(s,a) - \eta \, \nu \, \big(\phi(s,a) - \gamma \, \phi(s',a^*)\big), & \text{if } Q_w(s,a) - (TQ_w)(s,a,s') \leq 0, \end{cases}$$

where $a^* := \arg\max_{u \in \mathcal{A}} Q_w(s',u)$.

Update:

$$w \leftarrow w - \alpha \, \nabla_w L^{LB}(w), \tag{R.2}$$

### R.1.5 ALGORITHM

Both the standard Q-learning and the log-barrier Q-learning method share the same training loop and differ only in how the parameter update is computed. The common algorithmic skeleton is summarized in Algorithm 3.

In this experiments, the behavioral policy $\pi_b$ is required only to have *full support* over the action set, i.e.,

$$\pi_b(a \mid s) > 0 \qquad \text{for all } (s,a),$$

so that every action remains reachable under the data-collection process. Any stochastic policy satisfying this condition can be used in the common training loop (Algorithm 3), and no particular structure is assumed beyond full support.

With this flexibility in mind, and as we will later demonstrate, we deliberately consider two contrasting choices of $\pi_b$: the standard $\epsilon$-greedy policy and the $\epsilon$-*reverse*-greedy variant. The former tends to favor the maximizer of the current value estimate, whereas the latter intentionally favors the minimizer. Both policies maintain full support (as long as $\epsilon > 0$), but they induce substantially different degrees of off-policy mismatch. This design allows us to amplify and expose the classical instability mechanisms associated with the deadly triad in linear Q-learning, while at the same time enabling us to observe how the proposed log-barrier approach behaves under such deliberately adverse learning conditions.

---

**Algorithm 3** Common training loop under linear function approximation with $\pi_b$

---

1: **Input:** MDP environment, initial parameter vector $w_0$, basis matrix $\Phi$, behavioral policy $\pi_b$, horizon $T$
2: Initialize state $s_0$ and set $w \leftarrow w_0$
3: **for** $t = 0, 1, \ldots, T-1$ **do**
4:     Sample action $a_t \sim \pi_b(\cdot \mid s_t, w)$                                              $\triangleright$ behavioral policy
5:     Observe transition $(s_{t+1}, r_{t+1})$
6:     Compute gradient:

$$\nabla L_t(w) = \begin{cases} \text{Standard Q-learning: } \nabla_w L(w), & \text{(Eq. R.1)} \\ \text{Log-barrier Q-learning: } \nabla_w L^{\text{LB}}(w), & \text{(Eq. R.2)} \end{cases}$$

7:     Parameter update:
$$w \leftarrow w - \alpha \nabla L_t(w)$$

8: **end for**
9: **return** $w$, $Q = \Phi \cdot w$ .

---

## R.2 EXPERIMENT RESULTS

We consider two behavioral policies, each defined by a simple selection rule. For any state $s$, the standard $\epsilon$-greedy policy selects

$$a = \begin{cases} \text{Uniform}(\mathcal{A}), & \text{with probability } \epsilon, \\ \arg\max_{a' \in \mathcal{A}} Q_w(s, a'), & \text{with probability } 1 - \epsilon, \end{cases}$$

whereas the $\epsilon$-reverse-greedy policy selects

$$a = \begin{cases} \text{Uniform}(\mathcal{A}), & \text{with probability } \epsilon, \\ \arg\min_{a' \in \mathcal{A}} Q_w(s, a'), & \text{with probability } 1 - \epsilon. \end{cases}$$

Both policies maintain full support as long as $\epsilon > 0$, but they differ significantly in the extent to which they induce off-policy sampling.

As described above, we conducted two sets of experiments that differ only in the choice of the behavioral policy. The first set employs the standard $\epsilon$-greedy strategy, while the second adopts the $\epsilon$-reverse-greedy variant. These two policies induce notably different degrees of off-policy mismatch, enabling us to observe how each update rule behaves under progressively more adverse sampling conditions. The corresponding results are presented in Figure 8 and Figure 9, respectively.

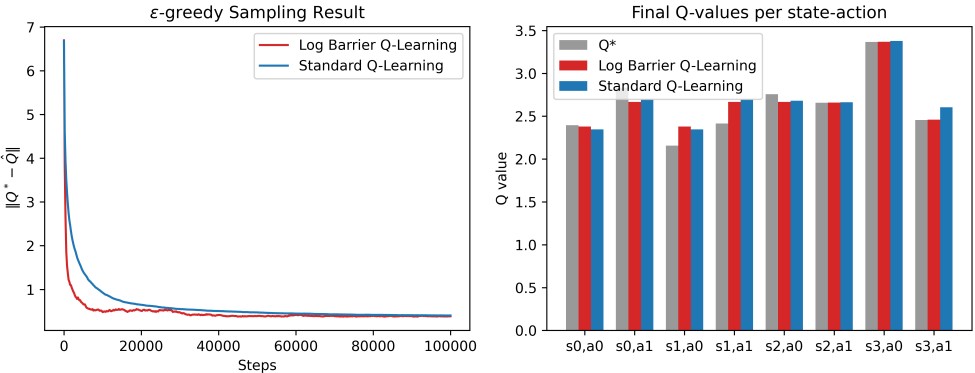

Figure 8: Experiment result under the $\epsilon$-greedy behavioral policy.

Table 3: Hyperparameter settings for the toy MDP experiments under two behavioral policies.

| Parameter | $\epsilon$-greedy | $\epsilon$-reverse-greedy |
|---|---|---|
| **General training settings** | | |
| $\gamma$ (discount factor) | 0.7 | 0.7 |
| $\epsilon$ (exploration rate) | 0.3 | 0.3 |
| $T$ (total training steps) | 100,000 | 100,000 |
| $\alpha$ (learning rate) | 0.2 | 0.2 |
| $Q_{\text{init}}$ (initial Q-value used for LS warm-start) | 5.0 | 5.0 |
| **Log-barrier coefficients** | | |
| $\eta$ (barrier parameter) | $5 \times 10^{-5}$ | $5 \times 10^{-5}$ |
| $\varepsilon$ (feasibility margin) | $10^{-3}$ | $10^{-3}$ |
| $\nu$ (infeasible-region enforce parameter) | $1.2 \times 10^{5}$ | $3.2 \times 10^{5}$ |

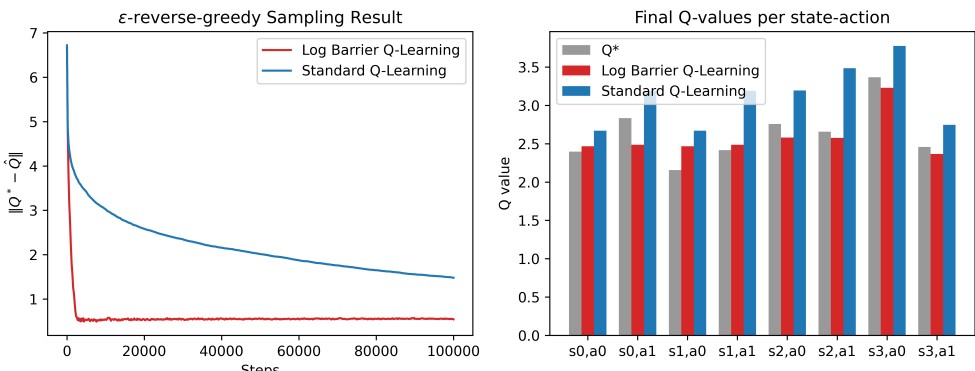

Figure 9: Experiment result under the $\epsilon$-reverse-greedy behavioral policy.

From these results, by comparing Figure 8 and Figure 9, we clearly see that standard Q-learning is highly sensitive to the sampling distribution induced by the behavioral policy. When the distribution of the data deviates from that implied by the optimal policy, the resulting off-policy mismatch amplifies error propagation—a phenomenon well documented in (Munos and Szepesvári, 2008). In our experiments, deliberately modifying the behavioral policy (e.g., through the $\epsilon$-reverse-greedy strategy) creates a more adverse off-policy regime. Under this setting, standard Q-learning (blue) exhibits slower convergence and larger approximation errors, whereas the milder $\epsilon$-greedy regime leads to comparatively more stable learning outcomes, as expected.

In contrast, our log-barrier Q-learning method (red) exhibits a notably different behavior. As long as the behavioral policy maintains full support, changes in its sampling distribution do not substantially degrade performance. The algorithm consistently converges rapidly and produces more accurate value estimates. This can be clearly seen in Figure 9: even under the adverse $\epsilon$-reverse-greedy policy, the final estimated $Q$-values remain close to $Q^*$, and the corresponding error curve $\|Q^* - \hat{Q}\|$ exhibits substantially reduced sensitivity to the off-policy sampling conditions.

However, we note that these results reflect idealized settings obtained through careful hyperparameter tuning, and in practice each experiment required different choices of the barrier coefficient ($\eta$) and the infeasible enforce parameter ($\nu$) to achieve stable performance. The precise coefficients used in each setting are summarized in Table 3. In the next subsection, we discuss what phenomena arise during this tuning process and how these parameters can be adjusted in a controlled and systematic manner to maintain stability across different sampling conditions.

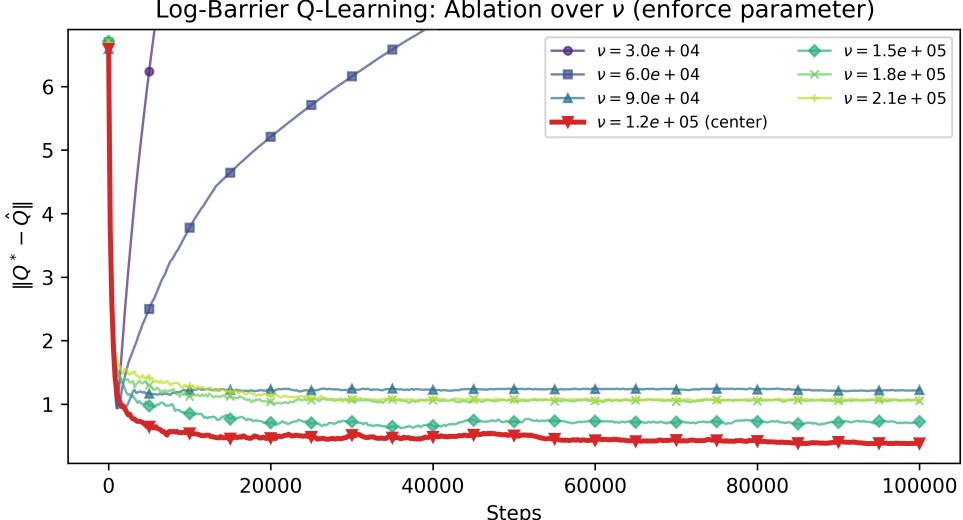

Figure 10: Ablation results of the log-barrier Q-learning method under the $\epsilon$-greedy behavioral policy for various choices of the enforce parameter $\nu$. The $\eta$ value is fixed as $5 \times 10^{-5}$.

### R.3 Practical Considerations for Hyperparameter Selection

While tuning the hyperparameters, we observed several important patterns in how our approach behaves under different settings. The discussion in this subsection therefore provides practical guidance for selecting appropriate hyperparameters. In this subsection, we present an ablation study that varies the enforcement parameter $\nu$ while keeping the barrier coefficient $\eta$ fixed, evaluated under the $\epsilon$-greedy behavioral policy. The key observation is that, for a fixed value of $\eta$, the optimal choice of $\nu$ is tightly coupled with it. As shown clearly in Figure 10, the best-performing hyperparameter pair appears at $\nu = 1.2 \times 10^5$ and $\eta = 5 \times 10^{-5}$ (highlighted by the red curve).

When the enforcement parameter $\nu$ is decreased below this optimal value, the error curve initially descends to a shallow minimum but then increases sharply, indicating a rapid degradation in stability. This phenomenon is not because the alrogithm suffers from overestimation. Rather, the pressure in the minimization step drives the estimated $\hat{Q}$-values down aggressively, leading to a sharp increase in the approximation error. This occurs because reducing $\nu$ weakens the penalty in the infeasible region, allowing the algorithm to accept the violation and push the estimated $\hat{Q}$-values further downward, ultimately destabilizing the error dynamics. This behavior is clearly illustrated in Figure 11.

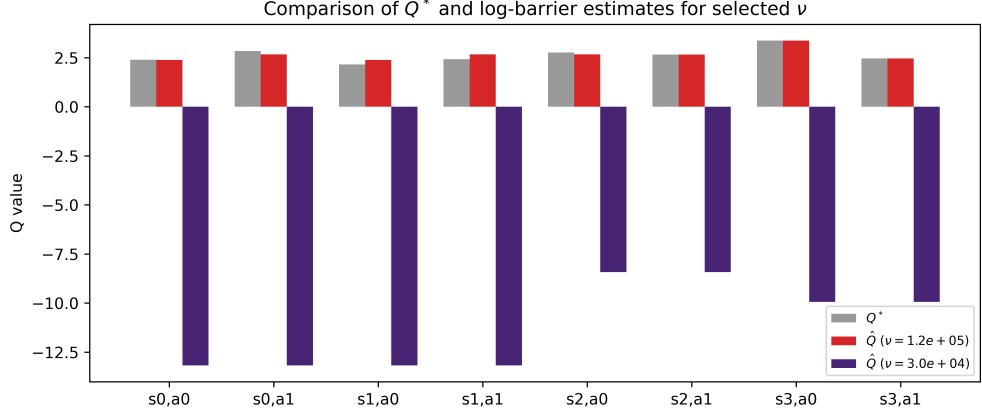

Figure 11: Comparison of $\hat{Q}$-values between the optimal and a smaller enforcement parameter $\nu$ setting.

In contrast, when $\nu$ is set larger than its optimal value, the framework still converges stably but exhibits a noticeable bias, which appears as an overestimation of the $Q$-values. This overestimation does not stem from theoretical defect; rather, it arises because an excessively large enforcement parameter imposes strong penalties whenever the update violates the infeasible region. The influence of this penalty persists even within the feasible region, effectively discouraging the algorithm from approaching the boundary closely and preventing it from accurately approximating the true $Q$-values. This behavior is clearly illustrated in Figure 12.

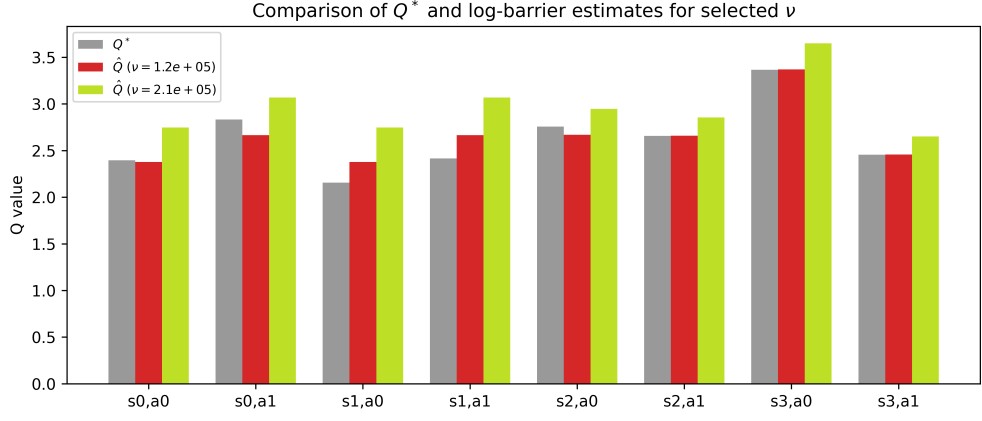

Figure 12: Comparison of $\hat{Q}$-values between the optimal enforcement parameter and a larger $\nu$ setting.

In practice, our empirical observations indicate that when the underlying MDP configuration is altered (e.g., the discount factor $\gamma$, the sampling policy distribution, or the reward function $r$), the optimal values of the hyperparameters $\nu$ and $\eta$ also shift accordingly. Nevertheless, with appropriately tuned hyperparameter pairs, the algorithm consistently achieves estimates that remain close to the true optimal value function $Q^*$. According to our theoretical result in Theorem 1, annealing the barrier coefficient $\eta$ during training may appear to be a reasonable strategy. However, such an approach inherently requires an adaptive adjustment of the enforcement parameter $\nu$, which complicates the optimization process. For this reason, based on our empirical findings, it is generally more stable to fix one parameter (either $\eta$ or $\nu$) and tune the other accordingly.

R.4   TABULAR SETTING UNDER STOCHASTIC SAMPLING

Using the experimental setup described in Appendix R, our analysis can be directly extended to the tabular setting as a special case under the same stochastic sampling conditions. In particular, when the identity feature map is adopted, the function approximation architecture becomes an exact tabular representation of the state–action value function. Under this specialization, the tabular log-barrier Q-learning algorithm corresponds to the instance of Algorithm 3 obtained by setting $\Phi = I$ and selecting log-barrier Q-learning as the learning framework, and its explicit form is presented in Algorithm 4.

---

**Algorithm 4** Log-barrier Q-learning (tabular case)

---

1: Initialize $Q(s, a)$ as a table.
2: Set step size $\alpha > 0$, discount $\gamma \in (0, 1)$, log-barrier parameter $\eta > 0$, enforce parameter $\nu > 0$, and margin $\varepsilon > 0$
3: **for** each episode **do**
4:     $s \leftarrow s_0$
5:     **while** $s$ is not terminal **do**
6:         Choose $a$ at state $s$ by $\pi_b$ w.r.t. $Q(s, \cdot)$
7:         Take action $a$, observe reward $r$ and next state $s'$
8:         **if** $s'$ is terminal **then**
9:             $(TQ)(s, a, s') = r$
10:            $a^*$ undefined
11:        **else**
12:            $a^* = \arg\max_{u \in \mathcal{A}} Q(s', u)$
13:            $(TQ)(s, a, s') = r + \gamma Q(s', a^*)$
14:        **end if**
15:        Define the Bellman gap

$$\Delta(s, a) := Q(s, a) - (TQ)(s, a, s').$$

16:        Compute the gradient at $(s, a)$:

$$\nabla_{Q(s,a)} L(Q) = \begin{cases} 1 - \eta \dfrac{1}{\Delta(s, a) + \varepsilon}, & \text{if } \Delta(s, a) > 0, \\[2mm] 1 - \eta \nu, & \text{else if } \Delta(s, a) \le 0. \end{cases}$$

17:        Compute the gradient at $(s', a^*)$ (full gradient into the target):

$$\nabla_{Q(s',a^*)} L(Q) = \begin{cases} 0, & \text{if } s' \text{ is terminal}, \\[2mm] \eta \gamma \dfrac{1}{\Delta(s, a) + \varepsilon}, & \text{else if } \Delta(s, a) > 0, \\[2mm] \eta \nu \gamma, & \text{else if } \Delta(s, a) \le 0. \end{cases}$$

18:        Update current state–action value:

$$Q(s, a) \leftarrow Q(s, a) - \alpha \nabla_{Q(s,a)} L(Q).$$

19:        Update next greedy state–action value:

$$Q(s', a^*) \leftarrow Q(s', a^*) - \alpha \nabla_{Q(s',a^*)} L(Q).$$

20:        $s \leftarrow s'$
21:    **end while**
22: **end for**

---

From a theoretical perspective, standard Q-learning is known to converge in the tabular case, as established in classical results (Jaakkola et al., 1993; Tsitsiklis, 1994). In contrast, our log-barrier framework, as shown in Theorem 1 and Theorem 4, also converges but exhibits a systematic bias

governed by the barrier coefficient $\eta$. Conducting experiments in the tabular setting therefore provides a clear and practical illustration of these theoretical properties.

### R.4.1 $\epsilon$-GREEDY SAMPLING

Therefore, we conducted an experiment in the $\epsilon$-greedy setting under the tabular case to compare standard Q-learning with log-barrier Q-learning. The results of this comparison are presented in Figure 13.

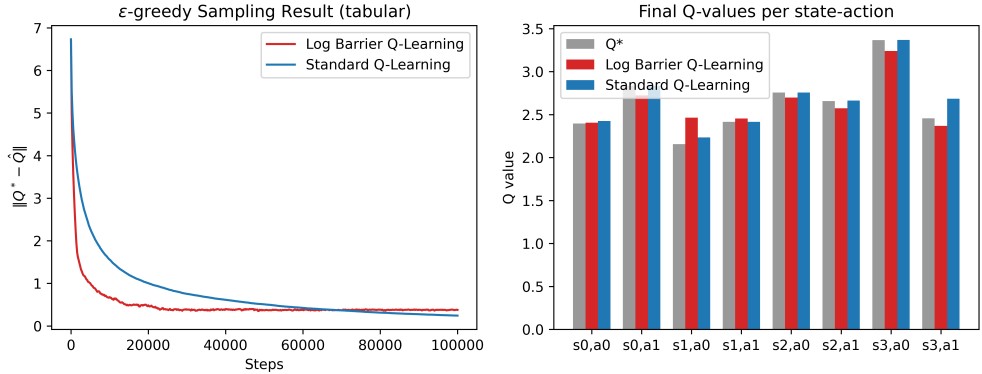

Figure 13: Experiment result under tabular setting using $\epsilon$-greedy sampling.

According to Figure 13, the experimental results align well with our theoretical expectations. While the standard Q-learning algorithm exhibits stable convergence in the tabular setting, our log-barrier variant likewise converges stably but retains a small, systematic bias, consistent with the analysis discussed above. Nevertheless, the log-barrier method offers a clear advantage in terms of sampling efficiency: it converges noticeably faster than standard Q-learning. This highlights a natural trade-off between convergence speed and approximation bias, which may motivate the choice of the log-barrier framework in scenarios where rapid learning is preferable. In this experiment, the $\eta$ and $\nu$ are tuned as $\eta = 5 \times 10^{-5}$ and $\nu = 1.3 \times 10^5$, and any other hyperparameters are same as Table 3.

### R.4.2 $\epsilon$-REVERSE-GREEDY SAMPLING

Similarly, we conducted an $\epsilon$-reverse-greedy sampling experiment to examine the behavior of both algorithms under tabular setting. The corresponding results are shown in Figure 14.

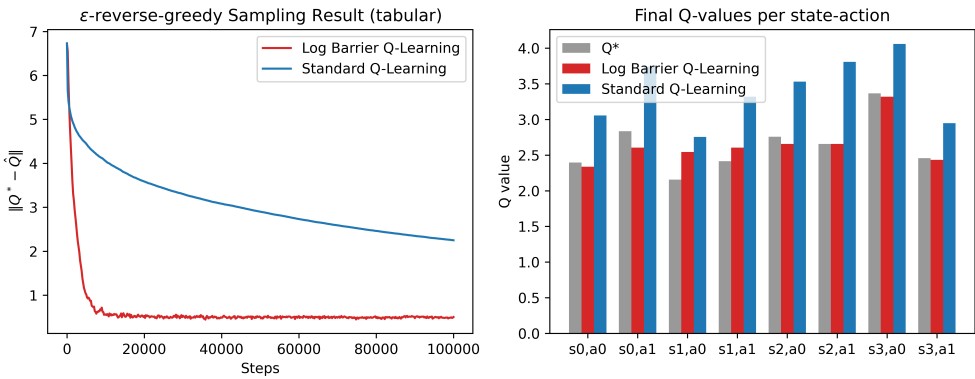

Figure 14: Experiment result under tabular setting using $\epsilon$-greedy sampling.

Under this setting, although standard Q-learning is guaranteed to converge, its convergence is noticeably slow due to the adverse sampling conditions. In contrast, our log-barrier approach exhibits coherent behavior, converging more rapidly while maintaining stable learning dynamics. In this experiment, we set the hyperparameters to $\eta = 5 \times 10^{-5}$ and $\nu = 6.0 \times 10^5$, and any other hyperparameters are same as Table 3.

## S    APPENDIX: COMPARISON OF LOG-BARRIER Q-LEARNING AND PRIMAL-DUAL Q-LEARNING

In this short section, we briefly compare the performance of the proposed log-barrier Q-learning and a conventional stochastic primal–dual Q-learning method in a simple tabular setting. We use the OpenAI Gym `FrozenLake` environment with the `is_slippery` option set to `True`, which induces stochasticity in the environment and makes it challenging to accurately compute the action–value function. The primal–dual Q-learning algorithm follows the approach of Lee and He (2018), which solves the MDP's LP formulation using a first-order primal–dual method; this algorithm is applicable only in the tabular case. Figure 15 below shows the return curves for the two algorithms.

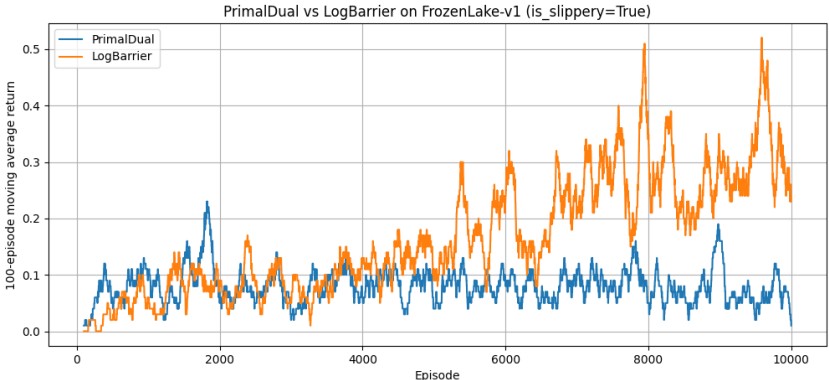

Figure 15: Return vs episodes plot of the proposed log-barrier Q-learning and stochastic primal-dual Q-learning (Lee and He, 2018)

From the figure, we observe that log-barrier Q-learning achieves superior learning performance compared with the first-order primal–dual method. Note that the hyperparameters for each method were carefully tuned to give the best possible performance for that method.

## T    APPENDIX: ABLATION AND ADDITIONAL EXPERIMENTS OF DEEP LEARNING VARIANTS

As discussed in Section Q, we found the proposed methods to be highly sensitive to the barrier coefficient $\eta$. In this section, we present an ablation study over different choices of $\eta$ and analyze the sensitivity of learning performance to this coefficient. In addition, we provide additional DDPG experiments that were omitted from the main text due to page limitations.

### T.1    ABLATION STUDY ON DEEP LEARNING VARIANTS

To investigate the sensitivity of the barrier coefficient ($\eta$) in the log-barrier DQN and DDPG algorithms, we conducted a series of ablation studies by varying the value of $\eta$. For each Gymnasium environment used for log-barrier DQN, we performed a grid search over $\eta$ using task-specific ranges and step sizes rather than a single fixed grid across all tasks. For MuJoCo environments used for log-barrier DDPG, we test the coefficient $\eta$ ranged from 0.005 to 0.05 in increments of 0.005. As

presented in Figure 16 and Figure 17, we found the barrier coefficient ($\eta$) to be a highly sensitive hyperparameter, particularly in MuJoCo environments.

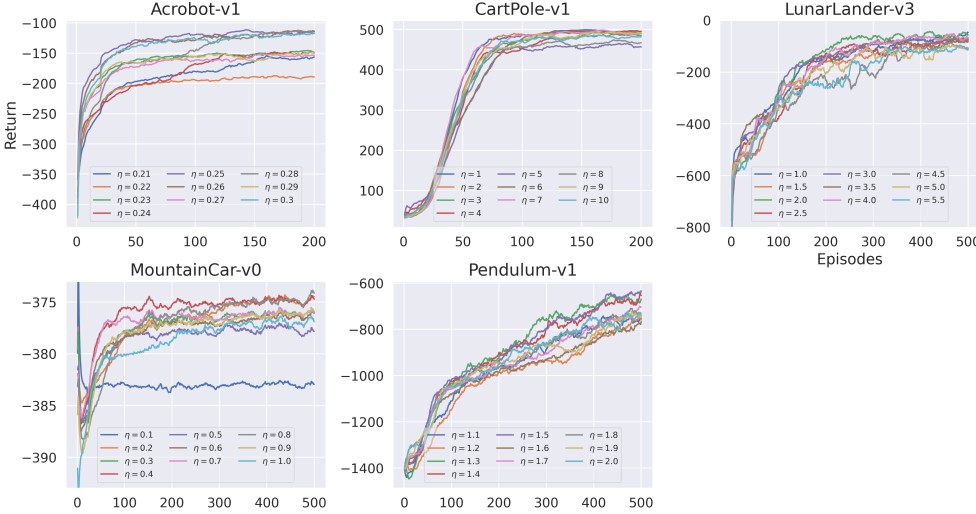

Figure 16: Ablation study of log-barrier DQN over the log-barrier coefficient $\eta$ across different environments.

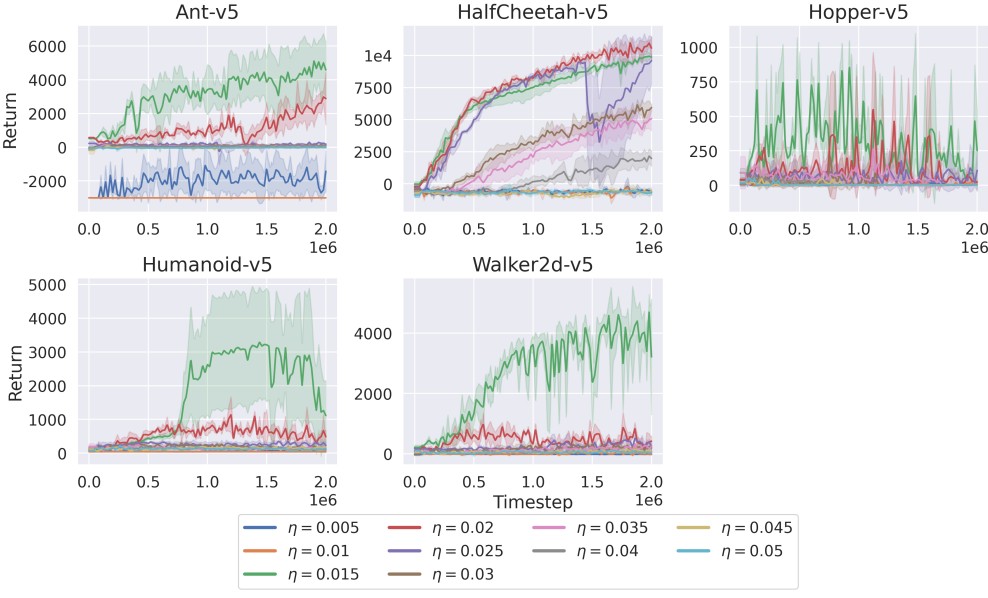

Figure 17: Ablation study of log-barrier DDPG over the log-barrier coefficient $\eta$ across different environments.

## T.2 ADDITIONAL EXPERIMENTS OF LOG-BARRIER DDPG

To further demonstrate the performance of log-barrier DDPG, we conducted additional experiments beyond those reported in Section 7 and present the extended results below. We evaluate and compare the original DDPG with our proposed log-barrier DDPG across several Gymnasium-Robotics environments, using five different random seeds for each algorithm. The resulting performance curves are shown in Figure 18, and the list of environments along with their final returns is summarized

in Table 4. Overall, log-barrier DDPG performs slightly better or at least comparably to the original DDPG in most environments.

From the results Figure 18 and Table 4, the fact that `FetchPickAndPlaceDense`, `FetchPushDense`, and `FetchSlideDense` exhibit nearly identical returns for both algorithms may initially appear unusual. However, this behavior is expected once we consider that these environments compute rewards solely based on the distance between the movable object ("puck") and the goal position. Since both algorithms completely fail to manipulate the object in these challenging tasks, the puck remains near its initial position, resulting in identical returns for both DDPG and log-barrier DDPG.

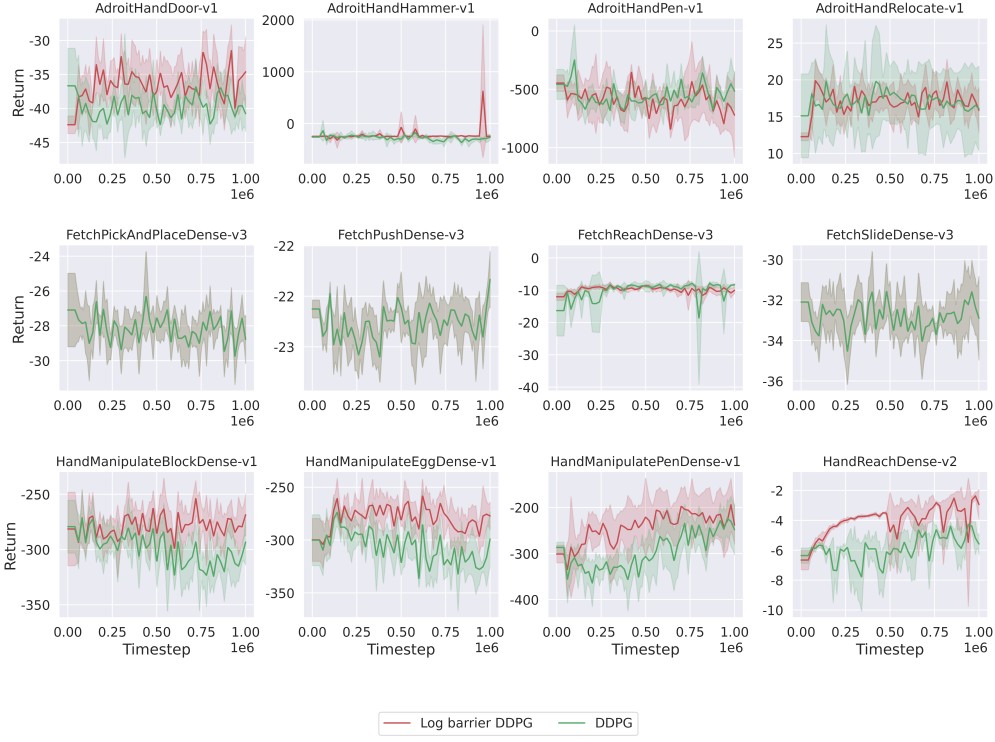

Figure 18: Additional experiments conducted in various Gymnasium-robotics environments.

| Environment | DDPG | Log-barrier DDPG |
|---|---|---|
| AdroitHandDoor-v1 | -39.3 ± 1.3 | **-36.0 ± 2.0** |
| AdroitHandHammer-v1 | -303.1 ± 22.0 | **-154.8 ± 129.4** |
| AdroitHandPen-v1 | **-519.1 ± 56.1** | -635.9 ± 125.0 |
| AdroitHandRelocate-v1 | 16.5 ± 5.0 | **17.3 ± 0.8** |
| FetchPickAndPlaceDense-v3 | **-28.3 ± 0.4** | **-28.3 ± 0.4** |
| FetchPushDense-v3 | **-22.7 ± 0.1** | **-22.7 ± 0.1** |
| FetchReachDense-v3 | **-9.3 ± 1.0** | -10.2 ± 0.7 |
| FetchSlideDense-v3 | **-32.5 ± 0.2** | **-32.5 ± 0.2** |
| HandManipulateBlockDense-v1 | -308.7 ± 14.1 | **-278.8 ± 5.4** |
| HandManipulateEggDense-v1 | -318.2 ± 14.4 | **-285.2 ± 5.5** |
| HandManipulatePenDense-v1 | -248.8 ± 21.5 | **-223.1 ± 51.2** |
| HandReachDense-v2 | -5.1 ± 0.5 | **-3.2 ± 0.6** |

Table 4: DDPG vs. Log-barrier DDPG Performance comparison across different environments.

## U    APPENDIX: EXPERIMENTS OF MAXIMIZATION BIAS

In this section, we demonstrate that our log-barrier approach can effectively avoid overestimation, a phenomenon from which iteration-based algorithms such as Q-learning often suffer. To illustrate this, we employ the well-known maximization bias example introduced in Sutton and Barto (1998) where there exist two non-terminal states A and B at which two actions, left and right, are admitted as illustrated in Figure 19. There also are two terminal states denoted with grey boxes and A is the initial state. Executing the action right leads to the termination of an episode with no rewards, while the action left leads to state B with no rewards. There exist many actions at state B, which will all immediately terminate the episode with a reward drawn from a normal distribution $\mathcal{N}(\mu, 1)$. In our experiments, we consider $r \sim \mathcal{N}(\mu, 1)$ with $\mu = -0.1$ .

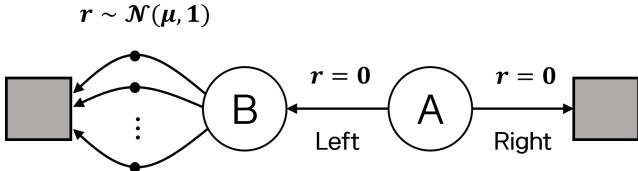

Figure 19: Sutton's example for demonstrating overestimation bias

The implementations of Q-learning and double Q-learning follow the descriptions in Sutton and Barto (1998), both using an $\epsilon$-greedy exploration strategy with $\epsilon = 0.1$. In this experiments, we adopt Algorithm 4 for the tabular log-barrier Q-learning method and configure its $\epsilon$-greedy behavior policy to match the same $\epsilon = 0.1$, ensuring a consistent exploration scheme across all methods. For this experiment, we carefully tuned the hyperparameters of log-barrier Q-learning to $\eta = 0.0005$, $\nu = 2000$, and $\varepsilon = 0.01$. We emphasize that the parameter $\varepsilon$ follows the definition given in Section N and is strictly distinct from the exploration rate $\epsilon$ used in the behavior policy.

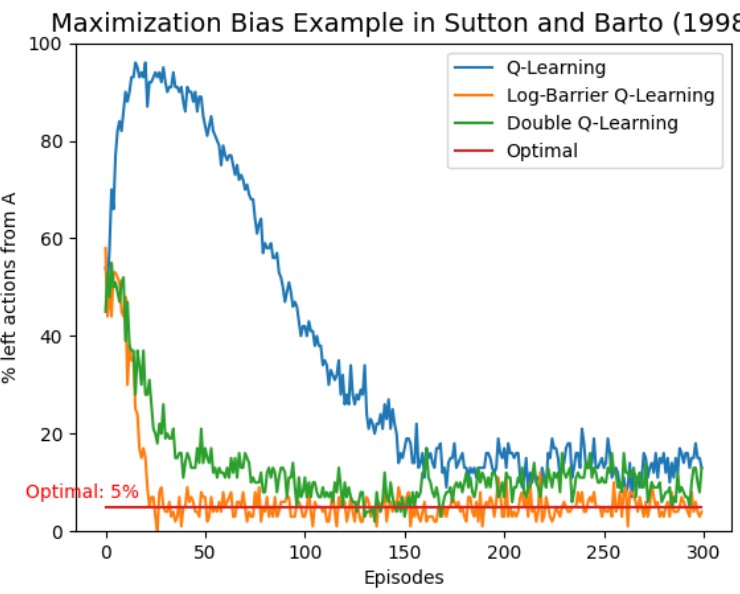

Figure 20: Comparison of maximization bias among Q-learning, log-barrier Q-learning, and double Q-learning. The implementations of Q-learning and double Q-learning strictly follow the descriptions in Sutton and Barto (1998).

From Figure 20, we observe that our proposed log-barrier Q-learning effectively mitigates the overestimation issue. In contrast, original Q-learning exhibits substantial overestimation, as is well

known, and double Q-learning, which was specifically developed to address this problem, nevertheless retains a small but non-negligible positive bias above the 5% line even after convergence. By comparison, log-barrier Q-learning converges to the optimal line more rapidly and maintains unbiased action selection throughout training. Moreover, whereas double Q-learning requires two separate Q-tables, our approach is able to accurately estimate the optimal action using only a single Q-table.