# OpenReview forum: "Analysis of approximate linear programming solution to Markov decision problem with log barrier function"
_ICLR.cc/2026/Conference — ICLR 2026 Poster_

### Official Review · Reviewer_fCTj · 2025-10-17

**Soundness:** 3
**Presentation:** 3
**Contribution:** 3
**Rating:** 8
**Confidence:** 3

**Summary:**

This paper proposes a novel framework for solving Markov Decision Problems (MDPs) through a linear programming (LP) formulation augmented with a log-barrier function.
The authors show that this reformulation converts the constrained LP into an unconstrained optimization problem, which can then be efficiently solved using gradient descent.
Theoretical results include upper and lower error bounds on the approximate Q-function and policy performance, both shown to scale linearly with the barrier parameter.
The framework is extended to deep reinforcement learning, yielding new variants of DQN and DDPG that replace the standard Bellman loss with a log-barrier–based objective.
Experimental results demonstrate competitive or superior performance in several OpenAI Gym and MuJoCo tasks.

**Strengths:**

The application of log-barrier methods to the LP formulation of MDPs is original and fills a gap in reinforcement learning literature, connecting classical optimization theory to RL in a principled way.

Extending the framework to DQN and DDPG variants demonstrates the practical potential of the theoretical findings.

The empirical results show tangible benefits in stability and performance, particularly in complex continuous-control tasks.

**Weaknesses:**

The experiments are confined to a few benchmark environments.
Broader testing (e.g., across more complex or stochastic settings) would strengthen the empirical claims.

There is little analysis isolating the contribution of the log-barrier term relative to standard LP or Bellman-based objectives.

**Questions:**

How sensitive is the convergence behavior to the choice of the barrier parameter $\eta$ and step-size in practice?
Could adaptive or annealing strategies improve stability?

Can the proposed approach be generalized to constrained RL or safe RL problems where inequality constraints are inherent to the objective?

Have the authors considered how the proposed loss compares, in practice, to other convex formulations such as convex Q-learning or logistic Q-learning?

---

> ### Author Response · Authors · 2025-11-21
>
> Answer to weakness 1.
>
> We agree with the reviewer’s comments. Following the reviewer's recommendations, we have strengthened the manuscript by conducting additional experiments across a broader set of environments (new Appendix R, Appendix S, Appendix T,and Appendix U). In particular, we have extended the experimental section as follows. For the deep-RL experiments, we have conducted additional comparisons across a broader set of environments and added an ablation study that systematically varies the barrier weight $\eta$ (Appendix T). We also included convergence-performance comparisons between the proposed log-barrier method and standard Q-learning in both the tabular setting and under linear function approximation (Appendix R). These results demonstrate that the log-barrier approach outperforms standard Q-learning in the tabular case and when using linear function approximation. In addition, we have also compared the convergence of the proposed log-barrier Q-learning with a conventional primal–dual Q-learning via simulations in the tabular setting (Appendix S).
>
>
> Answer to weakness 2.
>
> Thank you for the insightful comment. The results stated in Theorem 1 and Theorem 2 establish error bounds between the minimizer of our proposed objective and the true optimal Q-function. By contrast, standard LP and Bellman-based objectives typically admit the exact optimal Q as their minimizer. Our log-barrier formulation, however, yields an approximate optimal Q and the approximation error is explicitly controlled by the barrier weight $\eta$: as eta decreases the error becomes smaller. We hope this addresses the reviewer’s concern, and we welcome any further questions if parts remain unclear.
>
> Answer to question 1.
>
> We found the method to be generally highly sensitive to the barrier coefficient $\eta$. Accordingly, we performed careful hyperparameter tuning for the experiments, and we acknowledge that this sensitivity is an important issue that requires further investigation. As the reviewer suggested, we applied an annealing scheme that gradually reduces $\eta$ during training, and this often yields improved results. However, annealing itself introduces additional tuning choices: if $\eta$ is decreased via a schedule or learned online, one must still decide the initial value, the decay rate (how quickly to reduce $\eta$, and the final value, and experimental outcomes can vary substantially depending on these choices). Nonetheless, these auxiliary techniques frequently enhance empirical performance at the cost of increase tunning efforts. To free the method from excessive tuning, we plan to develop improved algorithms that are less dependent on hyperparameter selection. For example, mirror-descent–style updates that operate directly in the space satisfying the inequality constraints could maintain feasibility without heavy reliance on $\eta$. We view these directions as promising topics for future work. In response to the reviewer's comment, we have newly included a brief clarification of the above discussion (Appendix Q).
>
> Answer to question 2.
>
> We believe the proposed approach can be integrated very naturally into constrained or safe RL by, for example, incorporating a log-barrier term. However, this introduces additional tuning issue, namely the need to carefully set additional additional barrier coefficients. Consequently, we argue that a more principled framework, with stronger theoretical guarantees and improved algorithmic mechanisms for handling such hyperparameters, is necessary.
>
> Answer to question 3.
>
> Thank you for this important question. These methods are indeed close in spirit, and they replace the nonconvex Bellman objective with convex surrogates to obtain more stable optimization but they differ from our approach in several important respects (objective formulation, how inequality constraints are handled, and the role of a barrier/weighting term).
> For example, convex Q-learning is typically developed for MDPs with continuous state and action spaces, often modeled as dynamical systems. Therefore, this approach is not directly applicable to MDPs with discrete state and action spaces.
> Moreover, logistic Q replaces the hard max/raw-TD residuals with a smooth convex surrogate (e.g. logistic or log-sum-exp), trading some approximation bias for greater numerical stability and easier optimization in practice.
> Log-Barrier Q, by contrast, starts from the MDP’s LP formulation and enforces inequality constraints via a log-barrier term: its approximation error is explicitly controlled by the barrier weight $\eta$ and admits rigorous tabular guarantees.
> In short, logistic Q is a practical smoothing trick to stabilize learning; log-Barrier Q is a principled, constraint-aware LP-based approach with $\eta$-controlled approximation bounds (but requires careful tuning and is theoretically strongest in the tabular setting).

---

### Official Review · Reviewer_csmF · 2025-10-30

**Soundness:** 3
**Presentation:** 3
**Contribution:** 3
**Rating:** 8
**Confidence:** 4

**Summary:**

This paper introduces a log-barrier-based reformulation of the LP formulation for MDPs, replacing the inequality constraints with a single-objective function fη. The authors provide a thorough theoretical analysis of the approximation error, convergence and policy performance, and extend the framework to deep reinforcement learning (RL) via novel DQN and DDPG variants with empirical validation on Gymnasium and MuJoCo tasks.

**Strengths:**

1. The use of log-barrier functions to transform the LP formulation of MDPs into an unconstrained problem is innovative and a comprehensive theoretical interpretation has not been widely explored in the RL literature.

2. The paper provides a comprehensive theoretical foundation, including error bounds, convergence analysis and policy performance guarantees. The proofs are detailed and well-structured in the appendix.

3. The paper is well-organized with a clear statement of formulation and a clear presentation of analysis.

4. The authors evaluate the proposed method on both discrete and continuous control real-world tasks, demonstrating the competitive or improved performance over standard DQN and DDPG.

**Weaknesses:**

In general, broader evaluation on more real-world datasets is needed to strengthen the claims.

**Questions:**

1. How does the computational cost of the log-barrier approach scale with the size of the state and action spaces? Are there any extreme settings where the method becomes computational costly?

2. What is the performance of log-barrier function method compared to other LP-based or primal-dual RL methods in terms of sample efficiency and computation time?

---

> ### Author Response · Authors · 2025-11-21
>
> Answer to weakness 1.
>
> Following the reviewer's recommendation, we have strengthened the manuscript by adding experiments across a wider and more diverse set of environments (through newly added Appendix R, Appendix S, Appendix T, Appendix U). Evaluation in truly practical, real-world settings will likely require longer-term effort beyond the current revision window. Nevertheless, we will make every effort to perform experiments and provide additional evaluations in more practical environments before resubmission.
>
> Answer to question 1.
>
> Thank you for the question. (1) To address it, we have added the following iteration complexity to the revision: to achieve ||Q_k - \tilde Q_eta||infty <= epsilon, we need at least the following number of iterations:
> O( ln( ( sqrt(|S x A|) / epsilon ) * { ( eta * sum{(s,a,a') in S x A x A} w(s,a,a') ) / ( min_{(s,a) in S x A} rho(s,a) ) + r_max / (1 - gamma) } ) )
> The result is for the deterministic gradient descent case.
> We note that if w is a probability distribution, then the sum in the numerator becomes one. On the other hand, the term min_{(s,a) in S x A} rho(s,a) depends on the initial state distribution rho, and this term can be made arbitrarily small. For instance, when the initial state distribution is uniform, i.e., rho(s,a) = 1 / (|S||A|) and w is a probability distribution, then
> O( ln( eta * |S x A|^{3/2} / epsilon + ( |S x A|^{1/2} / epsilon ) * ( r_max / (1 - gamma) ) ) )
>
> For cost of the gradient computation, in tabular log-barrier method, each gradient update scales with the number of state–action pairs (i.e., linear in |S x A|) because the objective and its gradient must be evaluated over the relevant domain. Following the reviewer's comment, the above discussions have been newly included in Appendix H and Appendix J. Besides, the convergence analysis for the stochastic RL version requires significantly more research and is difficult to discuss now, considering the remaining page limit and time constraints. However, as the reviewer mentioned, this is an important area that warrants independent future study.
>
> (2) For the last question, we did not identify any extreme settings in which the proposed method becomes unusually costly compared with standard alternatives. We appreciate the reviewer’s insightful question.

---

> ### Author Response · Authors · 2025-11-21
>
> Answer to question 2.
>
> Thank you for the insightful question.  In the tabular setting, a single gradient update for the log-barrier objective costs $O(|{\cal S}||{\cal A}|)$, since the objective and its gradient are evaluated over all state–action pairs. This per-step order is therefore comparable to that of LP/primal–dual tabular solvers.
>
> In practice, however, two important differences make the log-barrier approach computationally favorable. First, primal–dual LP methods often require many more outer iterations and sometimes inner linear-system or dual solves (or costly projection steps), which substantially increases wall-clock time even when the asymptotic per-iteration order is similar. Second, the log-barrier method uses simple gradient-based updates that (i) avoid inner linear solves or complicated projection routines, (ii) are straightforward to implement and tune, and (iii) are readily parallelizable and accelerator-friendly. These factors lead to lower runtime in practice despite similar per-step complexity.
>
> Moreover, first-order primal–dual schemes applied to LP formulations typically work with objectives that lack strong (or strict) convexity, so their best available guarantees are often sublinear convergence rates. By contrast, our log-barrier objective $f_\eta$ is shown to be locally strongly convex (and locally Lipschitz) around the approximate solution. This local strong convexity implies that simple gradient updates attain linear (exponential) convergence once the iterates enter the strongly convex neighborhood, whereas primal–dual updates are generally limited to sublinear progress. In practical terms, local strong convexity translates into fewer iterations to reach high accuracy and substantially lower wall-clock time (especially because the log-barrier method also avoids costly inner linear solves or projection steps common in primal–dual solvers).
>
> Finally, we note that a full theoretical treatment (for example, sample-complexity and runtime guarantees) therefore requires substantial additional analysis and would exceed the scope and page budget of the present submission, since our primary goal here is to provide a rigorous solution analysis and $\eta$-dependent error bounds, material that already occupies the bulk of the manuscript.
> Accordingly, in the revised paper, we have instead added an empirical comparison of convergence speed between our log-barrier–based Q-learning and a first-order primal–dual RL baseline in the tabular and linear function approximation settings in Appendix R, Appendix S, Appendix T,and Appendix U. A more rigorous theoretical convergence rate analysis of the proposed RL (including sample complexity and runtime) remains an important direction for future work. We appreciate the reviewer’s insightful question.

---

### Official Review · Reviewer_Khvy · 2025-10-31

**Soundness:** 2
**Presentation:** 3
**Contribution:** 2
**Rating:** 4
**Confidence:** 3

**Summary:**

The paper proposes a new approach to solving the Bellman optimality equation in tabular Markov decision processes.
Instead of addressing the problem iteratively via dynamic programming, the authors formulate it as an LP.
They then reformulate the LP—with its inequality constraints—into an unconstrained optimization problem using a log-barrier, and obtain a solution by gradient descent.
Leveraging the convexity of the log-barrier function, the paper shows that the resulting objective is strongly convex (and hence strictly convex), which in turn justifies the effectiveness of gradient-descent–based optimization.
Moreover, by establishing the relationship between the approximation error of the proposed solution and the barrier parameter, the paper demonstrates convergence toward the optimal solution.
Empirically, the authors validate the approach by modifying the DQN objective with a log-barrier term and evaluating performance across several environments.

**Strengths:**

1. In terms of originality, this work investigates an LP-based route that has been relatively underexplored for solving the Bellman optimality equation. In particular, it introduces a new method that applies log-barrier techniques from convex optimization and develops a corresponding theoretical foundation.

2. While I have not thoroughly verified every proof in the appendix, the paper provides convergence guarantees for the proposed method. This foundation appears promising for subsequent research in the RL community on LP-based planning and related topics.

3. The manuscript is written clearly and is relatively easy to follow. The experimental section supports the claimed advantages of the method.

**Weaknesses:**

1. Motivation of the proposed approach should be supplemented. In one sentence, the proposed approach can be viewed as: cast the Bellman optimality equation as a constrained LP, convert it to an unconstrained problem with a log-barrier, and solve it by gradient descent. Although the theoretical foundation is established, the motivation for preferring this route over classical DP is not fully developed. A discussion comparing convergence speed and computational complexity to DP-based methods would strengthen the case for the approach.

2. More direct empirical validation should be added. Additional experiments could more directly isolate the contribution of the log-barrier loss. As the authors note, the log-barrier DQN differs from standard DQN not only in its loss but also, for example, in not using a target network. Hence, it is not entirely clear that the observed results are driven by the barrier term itself. A tabular setting directly comparing the proposed method to DP would provide more definitive evidence of its utility.

**Questions:**

1. On line 128, $\pi^\ast$ is defined as the greedy policy with respect to the optimal value, whereas on line 182 it is described via a probabilistic policy induced by the dual optimal variables. If $\pi^\ast$ is deterministic, can it still be represented in the form given on line 182 through $\lambda^*$?

2. As I understand it, $w$ functions like a hyperparameter, similar to $\eta$. Beyond the uniform choice (e.g., $w = 1/(|S||A|^2)$), what alternatives are reasonable, and what implementation differences would they entail? How is this parameter chosen in the actual experiments?

3. I do not fully understand the claim that standard DQN struggles with error propagation at sharp decision boundaries, while the log-barrier LP approach avoids this issue (line 396-401). Could the authors provide experimental evidence that directly supports this explanation?

4. Intuitively, the log-barrier imposes a strong penalty for violating Bellman constraints, controlled by $\eta$. Would using other penalties such as hinge or softplus lead to different empirical behavior or performance?

5. Why does the proposed method underperform (or perform less favorably) on the Hopper task? What characteristics of Hopper differentiate it from the other tasks in ways that affect the method?

6. Figure 3 suggests that smaller $\eta$ yields tighter bounds. In practice, should $\eta$ therefore be set as small as possible? What criteria were used to select the barrier parameter in Tables 1 and 2?

[Minor]

1. On line 209 the log-barrier function is denoted by $\phi$, whereas in Eq. (4) it is written as $\varphi$.

**Details Of Ethics Concerns:**

N.A.

---

> ### Author Response · Authors · 2025-11-21
>
> Answer to weakness 1.
>
> (1) For the first point, the LP formulation of MDPs has recently received renewed interest, especially in the offline-RL literature, because it offers several advantages over Bellman-equation–based approaches. The log-barrier approach proposed in this paper is intended as an effective alternative for solving the LP formulation of MDPs. RL methods that operate on the LP form typically rely on primal–dual schemes, which are known to exhibit relatively slow convergence in practice and can be less efficient than standard Q-learning [R1-R4]. By introducing a log-barrier term, our method enables a faster and more stable solution of the MDP LP. For these reasons, we believe there is a clear need to develop effective alternatives for solving the LP form, and our log-barrier method contributes toward that goal. In response to the reviewer's comment, we have revised the Introduction to present the above logical thread more clearly and cohesively, explicitly emphasizing the motivation for the log-barrier formulation, the main theoretical guarantees (Theorems 1–2), and the key empirical findings so that the paper’s rationale and contributions are immediately transparent to the reader.
>
> (2) For the second point, the reviewer is correct that a careful theoretical comparison of convergence speeds is an important and valuable direction. However, a complete theoretical treatment (e.g., sample complexity, runtimes, and convergence under alternative assumptions) would require substantial additional analysis and thus exceed the scope and page budget of this submission. Our primary objective is a rigorous analysis of the log-barrier solution and its error bounds, and that material already occupies the majority of the manuscript. Instead, we have empirically compared the convergence speed of the proposed gradient method with that of DP and newly added these results in the revised version (Appendix I). We have also newly performed an ablation study over different step-size choices for the proposed gradient method and reported the resulting analysis (Appendix I). We plan to pursue this topic in a dedicated follow-up study in the future that will investigate convergence rates, sample complexity, and practical runtime trade-offs both empirically and theoretically. We appreciate the reviewer’s suggestion.
>
> [R1] Mengdi Wang and Yichen Chen. An online primal-dual method for discounted markov
>  decision processes. In 55th IEEE Conference on Decision and Control (CDC), pages 4516–4521, 2016.
>
> [R2] Yichen Chen and Mengdi Wang. Stochastic primal-dual methods and sample complexity of reinforcement learning. arXiv preprint arXiv:1612.02516, 2016.
>
> [R3] Lee,D. and He, N. (2018). Stochastic primal-dual Q-learning. arXivpreprintarXiv:1810.08298.
>
> [R4] Lee,D.andHe,N. (2019). Stochastic primal-dual Q-learning algorithm for discounted MDPs. In 2019 American Ccontrol conference (Acc),pages4897–4902.
>
>
> Answer to weakness 2.
>
> Thank you for the helpful suggestion. (1) For the first point, note that the log-barrier DQN variant in our implementations did not use a target network, whereas the log-barrier DDPG variant employed a standard target network as in vanilla DDPG. Despite using target networks only in the DDPG variant, the log-barrier DDPG achieved superior performance to standard DDPG across many environments. We believe these results provide additional empirical evidence for the effectiveness of the proposed log-barrier approach.
>
> (2) Moreover, in response to the reviewer's suggestion, we have strengthened the experimental section in the revised version.
> In particular, we have empirically compared the convergence rates of the proposed gradient-based method against classical DP in tabular settings in the revised version (Appendix I and Appendix R). Moreover, following the reviewer's comment, we have also newly added an ablation study that systematically examines the effect of different step-size values $\alpha$ on convergence behavior. We have added an experiment in Appendix R.4 that directly and fairly compares standard Q-learning and log-barrier Q-learning in the tabular setting, where we explain how the two algorithms behave differently, thereby clarifying the effect and role of the barrier and the underlying optimization scheme.

---

> ### Author Response · Authors · 2025-11-21
>
> Answer to question 1.
>
> We thank the reviewer for pointing out these issues. As the reviewer suggested, when the optimal policy $\pi^*$ is deterministic, the stochastic policy described in line 182 can be shown theoretically to collapse to a one-hot vector, i.e., it reduces to the deterministic policy [R1, R2]. We have newly clarified this point in the revised version.
>
> [R1] Mengdi Wang and Yichen Chen. An online primal-dual method for discounted markov decision processes. In 55th IEEE Conference on Decision and Control (CDC), pages 4516–4521, 2016.
>
> [R2] Yichen Chen and Mengdi Wang. Stochastic primal-dual methods and sample complexity of reinforcement learning. arXiv preprint arXiv:1612.02516, 2016.
>
> Answer to question 2.
>
> Thank you for the good question. In the deterministic setting, a natural choice is $w(s,a)=1$; the effective weighting is then controlled by the barrier coefficient $\eta$. In the deep-RL variant, $w$ should be interpreted as the empirical distribution of state–action pairs induced by the replay buffer or by mini-batch sampling. Consequently, there is no need to choose $w$ explicitly in deep RL implementations, and standard replay/minibatch sampling already implements this weighting. In response, we have newly added a short clarification in the revised manuscript.
>
>
> Answer to question 3.
>
> To be more precise, standard DQN contains non-smoothness introduced by the max operator. This non-smoothness is concentrated on decision surfaces where the argmax action changes, and it can produce abrupt shifts in the Bellman target used by an MSE loss. Such sudden target changes are known to increase variance during training and to undermine stability. Moreover, the max operator is also associated with overestimation bias in MSE-based Bellman residual methods, which is well documented and is known to adversely affect the convergence behavior of Q-learning. By contrast, our proposed log-barrier Q formulation does not use a max operator and is therefore not subject to these particular sources of non-smoothness, abrupt target shifts, or the max-induced overestimation bias.
>
> Such a claim is generally difficult to demonstrate directly by experiment, but it can be supported indirectly through improved empirical performance. We have shown these performance gains in our experiments; in particular, when the method is applied to DDPG it produces substantially larger improvements than the conventional Bellman-residual (MSE) loss.
> Moreover, in response to the reviewer's suggestion, we have added several additional experiments (Appendix R and Appendix U) in the revision to more clearly demonstrate and quantify these improvements. In particular, we have newly added a linear function approximation experiment in Appendix R, allowing a direct comparison between standard Q-learning and our log-barrier Q-learning algorithm with linear function approximation, where we explain how the proposed method behaves differently and mitigates overestimation compared to the standard counterpart. Additionally, we experimentally demonstrate in Appendix U, using Sutton’s illustrative example, that the proposed log-barrier Q-learning reduces overestimation bias. Although the evidence is partial (since it is limited to the linear setting), we believe it provides clear and empirical support that our method can effectively avoid such instability. Finally, thank you for the helpful comment.
>
> Answer to question 4.
>
> Thank you for the good point. We have tested several alternative choices (e.g., SoftPlus) and found that, while these alternatives often yield reasonable performance, our proposed method typically performed better in our experiments. As a result, we omitted those alternatives from the final experimental set. Exploring and rigorously evaluating these alternative functions is, however, a challenging and interesting direction, and in future work we plan to study them more systematically. Following the reviewer's comment, we have newly included a brief clarification of the above discussion in the revised version.

---

> ### Author Response · Authors · 2025-11-21
>
> Answer to question 5.
>
> We thank the reviewer for the question. In general, deep-RL algorithms that rely on highly nonlinear neural networks exhibit convergence and performance that are difficult to predict, and algorithmic behavior often depends sensitively on environment dynamics, reward structure, and exploration properties. Therefore, it is not easy to identify a characteristic feature of Hopper that would logically explain the observed performance discrepancy relative to the other environments. One possible explanation is an \emph{underestimation bias}. As presented in Appendix R, our approach effectively suppresses the overestimation of the Q-function, while slightly underestimating the Q-values for some actions. Prior work has reported that methods explicitly designed to mitigate maximization bias—such as Double Q-learning and its clipped or maxmin variants—often trade overestimation for underestimation, depending on the size of the action space and the amount of function-approximation noise [1–4]. Although our method does not rely on TD targets or the standard $\max_{a'} Q(s',a')$ backup, its optimization objective has a similarly conservative effect in that it systematically discourages overly large Q-values. It is therefore plausible that, in environments with a small discrete action space—where the intrinsic maximization bias of standard Q-learning is already weak—our estimator behaves slightly pessimistically and underestimates the value of some actions. The Gymnasium environments used for our DQN experiments have small discrete action spaces, and among the MuJoCo control tasks we consider, Hopper has one of the lowest action dimensions. We therefore hypothesize that the proposed method performs less favorably on these environments precisely because their action spaces are relatively small. A more precise characterization of this bias, and its dependence on the action-space size, is left for future work.
>
> Moreover, Hopper involves highly nonlinear, contact-rich locomotion dynamics and is known to be sensitive to small changes in policy updates, reward scaling, and exploration. Such sensitivity can amplify small differences in optimization dynamics introduced by the log-barrier term. We will investigate this phenomenon further and, where appropriate, discuss these caveats and additional experimental results in the revised manuscript.
>
> [1] Z. Zhang, Z. Pan, and M. J. Kochenderfer, “Weighted Double Q-learning,” in Proceedings of the 26th International Joint Conference on Artificial Intelligence (IJCAI-17), pp. 3455–3461, Melbourne, Australia, 2017.
>
> [2] Q. Lan, Y. Pan, A. Fyshe, and M. White, “Maxmin Q-learning: Controlling the Estimation Bias of Q-learning,” in Proceedings of the 8th International Conference on Learning Representations (ICLR 2020), Addis Ababa, Ethiopia, 2020.
>
> [3] H. Jiang, J. Xie, and J. Yang, “Action Candidate Based Clipped Double Q-learning for Discrete and Continuous Action Tasks,” in Proceedings of the Thirty-Fifth AAAI Conference on Artificial Intelligence (AAAI-21), pp. 7979–7986, 2021.
>
> [4] W. Wei, Y. Zhang, J. Liang, L. Li, and Y. Li, “Controlling Underestimation Bias in Reinforcement Learning via Quasi-median Operation,” in Proceedings of the AAAI Conference on Artificial Intelligence, vol. 36, no. 8, pp. 8621–8628, 2022.

---

> ### Author Response · Authors · 2025-11-21
>
> Answer to question 6.
>
> Thank you for the question. Theoretically, as $\eta$ goes to zero, the log-barrier solution $\tilde Q_{\eta}$ approaches the true $Q^*$. This relationship is established in Theorem 1. In practice, however, taking $\eta$ too small can cause numerical issues. For example, with very small $\eta$, the gradient-based estimate of $\tilde Q_{\eta}$ may easily step outside the domain that satisfies the inequality constraints with relevantly small step-sizes due to finite-precision and optimization error.
>
> Moreover, a small $\eta$ can affect the numerical stability related to $\nu$ which is the infeasible-region penalty. To explain this phenomenon more clearly, we added an Appendix R.3.
> As we gradually decrease $\nu$, there comes a point at which the convex minimization framework prefers to reduce the Q-values even at the cost of incurring the penalty imposed by enforcement parameter $\nu$. Once this occurs, the optimizer pushes the solution into the infeasible region, after which the Q-values continue to decrease without bounds. This behavior corresponds precisely to the instability observed in Appendix R.3: when $\nu$ is smaller than $1.2 \times 10^5$, Figure 10 shows unstable optimization dynamics, and Figure 11 illustrates that the resulting Q-values fall significantly below their ideal levels. These empirical results clearly demonstrate the effect described above.
> Since the infeasible-region penalty is given by the product $\nu \times \eta$, smaller $\eta$ yields this phenomenon. Therefore, although using a smaller or gradually decreasing $\eta$ may appear theoretically reasonable, in practice it can destabilize the optimization and should be applied with caution.
>
> The $\eta$ values reported in Tables 1 and 2 were obtained by empirical tuning; in our experiments, we typically used moderate values of $\eta$ (see Tables 1–2 for the exact settings). At present, a principled criterion for selecting $\eta$ has not been established, and we view the development of such a selection rule as important future work. In response to the reviewer's comment, we have included a brief clarification of the above discussion in Appendix in the revised version.
>
> Answer to question 7.
>
> We thank the reviewer for carefully reading the manuscript. The typographical errors  will be corrected in the revised version.

---

### Official Review · Reviewer_UmFB · 2025-10-31

**Soundness:** 2
**Presentation:** 2
**Contribution:** 2
**Rating:** 4
**Confidence:** 3

**Summary:**

The paper reformulates the Q-LP for discounted MDPs using a log-barrier to combine all Bellman inequality constraints into a single unconstrained objective. For this unconstrained optimization problem, the paper studies its theoretical results and proves that both the upper and lower bounds scale linearly with the barrier parameter. They also propose barrier-style DQN/DDPG variants and report empirical gains on several MuJoCo tasks.

**Strengths:**

1. Theoretical soundness: Provides a clean log-barrier reformulation of the Q-LP, with proofs of convexity/strong-convexity on sublevel sets, Lipschitz gradients, exponential gradient-descent convergence, and linear-in-$\eta$ upper/lower bounds.

2. Conceptual clarity: The problem is well specified and the pipeline—Q-LP → barrier objective → first-order conditions → approximate duals/policies—is presented coherently and is easy to follow.

3. Practical algorithmic instantiation: The barrier idea is instantiated with minimal changes to standard DQN/DDPG, making it straightforward to implement.

**Weaknesses:**

At present, I do not find the contributions sufficient for ICLR. As a theory-focused paper, the motivation and insights feel limited. That said, I may be missing something, and I am happy to adjust my score if the concerns below are addressed.

In particular, my main concerns are as follows:

1. Motivation and positioning. The motivation for introducing a log-barrier is not fully convincing. I appreciate that the LP formulation of MDPs is less explored, but it remains unclear what concrete advantages the log-barrier brings over standard alternatives. In Section 4, the paper asserts that Lagrangian or primal–dual methods can be slow or lack convergence guarantees, but no references are provided and there are no experimental comparisons. I suggest to expand this discussion with specific contrasts in performance metrics such as convergence speed, required assumptions, dependence on $|\mathcal{S}|$, $|\mathcal{A}|$, computational cost, and practical stability.

2. Empirical support and ablations. The advantages over standard DQN/DDPG in Section 7 are stated but not theoretically grounded, and some claims read as conjectural. The paper also lacks ablation studies. It would help to include controlled synthetic simulations/experiments where the barrier variants have a clear, interpretable edge, compared to standard benchmark methods.

3. The theoretical results do not appear to extend to the deep RL setting in Section 6, as the Q function is not guaranteed to be convex as a function of $\theta$. I suggest to add some discussions for the neural settings.

**Questions:**

1. The main theoretical results show that the upper and lower bounds scale linearly with $\eta$, but the proposed algorithm only uses a fixed $\eta$. Are there any potential concerns on using a decreasing sequence of $\eta$ in the practical algorithm?

2. If possible, would you list the technical contributions of the paper, such as new frameworks or proof techniques?

---

> ### Author Response · Authors · 2025-11-21
>
> Answer to weakness 1.
>
> (1) Thank you for the excellent points. For the first point, the LP formulation of MDPs has recently received renewed interest, especially in the offline-RL literature, because it offers several advantages over Bellman-equation–based approaches. Unfortunately, RL methods that operate on the LP form typically rely on primal–dual schemes [R1-R4], which are known to exhibit relatively slow convergence in practice and can be less efficient than standard Q-learning as discussed in [R1-R4]. The log-barrier approach proposed in this paper is intended as an effective alternative for solving the LP formulation of MDPs. By introducing a log-barrier term, our method enables a faster and more stable solution of the MDP LP. For these reasons, we believe there is a clear need to develop effective alternatives for solving the LP form, and our log-barrier method contributes toward that goal. Moreover, as shown in our experiments, the proposed log-barrier method yields superior results in the policy-evaluation phase of DDPG compared with the conventional mean-squared Bellman-error (MSE) loss. Therefore, we view these findings as evidence of the promise of the log-barrier approach. In response to the reviewer's comment, we have revised the Introduction to present the above logical thread more clearly and cohesively, explicitly emphasizing the motivation for the log-barrier formulation, the main theoretical guarantees (Theorems 1–2), and the key empirical findings so that the paper’s rationale and contributions are immediately transparent to the reader.
>
> (2) For the second point, as suggested by the reviewer, we have added additional references on applying first-order primal–dual methods to LP [R5] in the revised version. For LP-based RLs, prior work [R1-R4] in this vein primarily provides convergence analysis in the tabular setting.
>
> (3) The primary objective of this paper is to provide a rigorous theoretical analysis of solutions of the log-barrier objective function, and to assess the approach’s applicability to deep RL. Because a large portion of the current manuscript is already devoted to these theoretical developments and to demonstrating empirical promise in function-approximation settings, a full treatment of tabular RL convergence rates and sample-complexity would require extensive additional material and would, in practice, exceed the scope of the present submission considering the limited timelines and pages. We agree with the reviewer that such analysis is valuable and worth pursuing. Accordingly, we would like to address convergence and sample-complexity for the tabular setting in a separate follow-up work in the future. Instead, we have empirically compared the convergence speed of the proposed log-barrier method with that of primal–dual RL in the tabular setting, and include these results in the revised manuscript in Appendix S. Although this comparison is empirical rather than theoretical, it will provide evidence of the efficiency of the proposed log-barrier method and offer useful practical insights.
> Finally, we appreciate the reviewer’s constructive suggestion and ask for their understanding of our decision to keep the current paper focused on the log-barrier objective function analysis and empirical evaluation of its deep RL versions.
>
> [R1] Mengdi Wang and Yichen Chen. An online primal-dual method for discounted markov
>  decision processes. In 55th IEEE Conference on Decision and Control (CDC), pages 4516–4521, 2016.
>
> [R2] Yichen Chen and Mengdi Wang. Stochastic primal-dual methods and sample complexity of reinforcement learning. arXiv preprint arXiv:1612.02516, 2016.
>
> [R3] Lee,D. and He, N. (2018). Stochastic primal-dual Q-learning. arXiv preprint arXiv:1810.08298.
>
> [R4] Lee,D.andHe,N. (2019). Stochastic primal-dual Q-learning algorithm for discounted MDPs. In 2019 American Ccontrol conference (Acc),pages4897–4902.
>
> [R5] Applegate, D., D´ıaz, M., Hinder, O., Lu, H., Lubin, M., O’Donoghue, B., and Schudy, W. (2021). Practical large-scale linear programming using primal-dual hybrid gradient. Advances in Neural Information Processing Systems, 34:20243–20257.

---

> ### Author Response · Authors · 2025-11-21
>
> Answer to weakness 2.
>
> Thank you for the helpful comment. (1) Extending the theoretical analysis results to deep RL is substantially more difficult or even impossible because the use of deep neural networks makes the model non-linear in the parameters $\theta$; this same difficulty arises in popular deep-RL methods such as DQN and DDPG. As a result, theoretical analysis in the deep-RL regime is largely intractable and we must primarily rely on empirical evaluation and comparisons. In the previous sections, we provide rigorous theoretical results for the tabular setting; while these results do not transfer directly to deep function approximation, and they nevertheless offer useful intuition and insight into how the proposed objective and algorithms behave when implemented with deep networks. In response to the reviewer’s first suggestion, we have incorporated the discussions described above into the revised manuscript. (2) Moreover, in response to the reviewer’s second suggestion, we have included ablation studies in the revised manuscript (Appendix T), where we will perform a systematic sweep of the barrier coefficient $\eta$ and report how changes in $\eta$ affect learning dynamics and final performance of the deep RL versions. In addition to the deep RL experiments, we have also implemented simulations in a linear function approximation setting (Appendix~R). Although the results in this section are still far from deep neural network setting, they nonetheless provide empirical support for our earlier conjecture that the log-barrier approach suppresses overestimation. Furthermore, under this setting, we also discuss how the log-barrier approach behaves compare to standard Q-learning algorithm, and clarify the interpretable relationship between the barrier variant and the enforcement parameter. We thank the reviewer for this constructive recommendation.
>
> Answer to weakness 3.
>
> Thank you for the helpful comments. When a deep neural network is used, the model becomes a nonlinear function of the parameters $\theta$, and the precise theoretical results derived for the tabular setting no longer apply directly. Nevertheless, the tabular analysis provides useful intuition: it clarifies the role of the log-barrier formulation, the effect of the barrier coefficient $\eta$, and the qualitative behavior one should expect when using function approximation, and thus offers approximate predictions for the deep-RL case. Following the reviewer’s suggestion, we have added a discussion to the revised manuscript explaining these limitations and the insights that carry over from the tabular setting. Thank you again for the constructive advice.
>
> Answer to question 1.
>
> Thank you for this helpful comment. As the reviewer suggested, we tried an annealing scheme that gradually reduces $\eta$ during training, and this often yields improved results. However, annealing itself introduces additional tuning choices: if $\eta$ is decreased via a schedule or learned online, one must still decide the initial value, the decay rate (how quickly to reduce $\eta$, and the final value), and experimental outcomes can vary substantially depending on these choices. Nonetheless, these auxiliary techniques frequently enhance empirical performance at the cost of increased tunning efforts.
> Moreover, we conjecture that the stability of the framework depends on a coupled relationship between the barrier coefficient $\eta$ and the enforcement parameter $\nu$. This is due to the practical implementation of the framework and is clarified in Appendix R.3, which was added to provide a clear explanation of this interaction in the revised version. According to our analysis, changing $\eta$ inevitably requires retuning $\nu$, which makes the situation more difficult to find optimal hyperparameter.
>
> To free the method from excessive tuning, we plan to develop improved algorithms that are less dependent on hyperparameter selection. For example, mirror-descent–style updates that operate directly in the space satisfying the inequality constraints could maintain feasibility without heavy reliance on $\eta$. We view these directions as promising topics for future work. In response to the reviewer's comment, we will briefly incorporate the discussions described above into the revised manuscript (Appendix Q).

---

> ### Author Response · Authors · 2025-11-21
>
> Answer to question 2.
>
> Thank you for the question. The main contributions of this paper can be summarized as follows:
>
> 1. A novel log-barrier LP formulation for MDPs and rigorous error bounds. To the best of our knowledge, we are the first to introduce a log-barrier approach that corresponds to the LP formulation of MDP, and to derive rigorous error bounds for the resulting approximate solution. In particular, we establish a clear quantitative relationship between the barrier weight $\eta$ and the approximation error.
>
> 2. Analytic properties of the log-barrier objective and convergence guarantees. We prove several important properties of the proposed objective, including convexity, local strong convexity, local Lipschitz continuity, and convexity of the feasible domain. Under the tabular setting we further show exponential convergence of deterministic gradient descent to an approximate solution.
>
> 3.	Preliminary investigation of applicability to deep RL. We develop a deep-RL variant of the proposed method and provide empirical results demonstrating that the approach is practical: the deep version trains stably and, in several environments, outperforms standard deep-RL baselines. These experiments illustrate the potential of the log-barrier formulation when combined with function approximation.
>
> In the revised manuscript, we have updated the paper based on the points above so that the stated contributions are presented more clearly (please see Introduction).

---

### Meta-Review · Area_Chair_UiKu · 2026-01-06

**Summary:**

This paper analyzes algorithms based on log-barriers for solving the LP formulation of MDPs.

The major concerns of the reviewers with negative scores include: studying log barriers-based algorithms for solving MDPs is not well-motivated, and the experiments did not fully decouple the effect of log-barrier methods and other design choices.

**Reviewer Concerns:**

I believe the authors did a job good addressing the above concerns. In particular, the authors explained in their response that log-barrier based methods could be a promising approach for designing new methods for solving the LP formulation of MDPs. Moreover, to address the concerns on experiments, the authors explained the design choices and provide more experiment results.

**Reviewer Scores:**

I think at least one of the two reviewers with a score of 4 would have increased the score to 6 given a full discussion.

---

### Decision · Program_Chairs · 2026-01-26

Accept (Poster)